# On the Stability and Generalization of Meta-Learning

**Yunjuan Wang**
Department of Computer Science
Johns Hopkins University
Baltimore, MD, 21218
ywang509@jhu.edu

**Raman Arora**
Department of Computer Science
Johns Hopkins University
Baltimore, MD, 21218
arora@cs.jhu.edu

## Abstract

We focus on developing a theoretical understanding of meta-learning. Given multiple tasks drawn i.i.d. from some (unknown) task distribution, the goal is to find a good pre-trained model that can be adapted to a new, previously unseen, task with little computational and statistical overhead. We introduce a novel notion of stability for meta-learning algorithms, namely *uniform meta-stability*. We instantiate two uniformly meta-stable learning algorithms based on regularized empirical risk minimization and gradient descent and give explicit generalization bounds for convex learning problems with smooth losses and for weakly convex learning problems with non-smooth losses. Finally, we extend our results to stochastic and adversarially robust variants of our meta-learning algorithm.

## 1 Introduction

Traditional machine learning algorithms excel at generalizing, but they often require extensive training data and assume that both training and test data come from the same distribution or task. In real-world scenarios, large sets of training data from a single task are often lacking. Instead, training data may stem from diverse tasks with shared similarities, while test data come from entirely new tasks. The challenge is to rapidly adapt to these unseen tasks without the need to train from scratch.

To address this challenge, meta-learning, also referred to as learning-to-learn, has emerged as an effective approach. Meta-learning has gained significant attention recently [Hospedales et al., 2021], with applications spanning across various domains including computer vision [Nichol et al., 2018] and robotics [Al-Shedivat et al., 2017], ranging from few-shot classification [Snell et al., 2017], hyperparameter optimization [Franceschi et al., 2018], to personalized recommendation systems [Wang et al., 2022].

As the name suggests, meta-learning operates on two levels of abstraction to enhance learning over time. On an intra-task level, the learner needs to find models that perform well on individual tasks. On a meta-level, the learner needs to figure out useful meta-information, perhaps a prior over tasks, that relates different tasks and allows transferring and adaptation of knowledge to new unseen tasks efficiently (both in terms of statistical as well computational overhead). It is typical to represent such meta-information in the form of a pre-trained model, which we can represent using certain meta-parameters. Distinct from a standard setting, meta-learning involves training on a diverse set of tasks. At test time, we evaluate the performance of the pre-trained model on new unseen tasks while allowing it to adapt using a small sample on the test task.

An increasing body of empirical research is dedicated to advancing meta-learning algorithms, among which model-agnostic meta-learning (MAML) [Finn et al., 2017] stands out as a prominent approach. MAML is designed to find a good meta-parameter w which facilitates the learning of task-specific parameters through a single step of gradient descent. In particular, given a set of $m$ tasks denoted as $\{\mathcal{D}_j\}_{j=1}^m$, MAML estimates the meta-parameter as $\mathrm{w} = \mathrm{argmin}_{\mathrm{w}} \frac{1}{m} \sum_{j=1}^m L(\mathrm{u}_j, \mathcal{D}_j)$, where task-specific parameters are computed as $\mathrm{u}_j = \mathrm{w} - \eta \nabla L(\mathrm{w}, \mathcal{D}_j)$.

38th Conference on Neural Information Processing Systems (NeurIPS 2024).

However, a notable limitation of MAML is that it requires computing second-order derivatives, which is computationally demanding for deep neural networks in practical applications. This computational complexity also poses a challenge for a theoretical understanding of MAML, an aspect that remains largely under-explored. To mitigate this challenge, several MAML variants have been proposed, including first-order MAML [Finn et al., 2017], Reptile [Nichol et al., 2018], and iMAML [Rajeswaran et al., 2019]. Owing to its success, MAML has been used for robust adversarial meta-learning [Yin et al., 2018, Goldblum et al., 2020, Wang et al., 2021, Collins et al., 2020], differential private meta-learning [Li et al., 2019], and personalized federated learning [Chen et al., 2018, Fallah et al., 2020].

Another popular framework for meta-learning is based on a "proximal" update, wherein the task-specific parameter are iteratively learned by minimizing the empirical loss and an $\ell_2$ regularizer [Denevi et al., 2018, Zhou et al., 2019, Denevi et al., 2019a, 2020, Jiang et al., 2021]. Given a task $\mathcal{D}$ and a meta-parameter w, the task-specific parameter u are defined as u $=$ $\mathrm{argmin}_{\mathrm{u}} L(\mathrm{u}; \mathcal{D}) + \frac{\lambda}{2} \| \mathrm{u} - \mathrm{w} \|^2$. This regularization strategy ensures that the task-specific parameter remains close to the meta-parameter. A similar strategy has been explored in other contexts. For example, Kuzborskij and Orabona [2017] study the problem of hypothesis transfer learning and show a fast rate on the generalization error of a task-specific parameter u returned by regularized empirical risk minimization conditioned on a good meta-parameter w. Yet, it remains unclear how to ensure finding such a good meta-parameter, *provably*. Relatedly, Denevi et al. [2019b] study stochastic gradient descent with biased regularization for linear model and incrementally update the bias (meta-parameter). Concurrently, Zhou et al. [2019] proposed the Meta-Prox algorithm as a generic stochastic meta-learning approach. Specifically, given a set of meta-training tasks $\mathcal{D}_1, \ldots, \mathcal{D}_m$, the meta-parameter w is estimated by solving $\min_{\mathrm{w}} \sum_{j=1}^{m} \min_{\mathrm{u}} L(\mathrm{u}, \mathcal{D}_j) + \frac{\lambda}{2} \| \mathrm{u} - \mathrm{w} \|^2$. Zhou et al. [2019] argue that Meta-Prox is a generalization of MAML since the gradient descent update in MAML can be viewed as taking the first-order Taylor expansion of the objective, [Zhou et al., 2019, Section 3.1].

In this work, we adopt the framework of Zhou et al. [2019] to study meta-learning from a theoretical perspective. Given $m$ tasks drawn i.i.d. from some (unknown) task distribution $\mu$, our goal is to find a good pre-trained model (the meta-parameter) which can be adapted to a new unseen task, drawn i.i.d. from $\mu$, at test time, using gradient descent. Our key contributions are as follows.

1. We introduce a novel notion of stability for meta-learning algorithms, namely uniform meta-stability. For $\bar{\beta}$ uniformly meta-stable algorithm, we bound the generalization gap by $\mathcal{O}(\bar{\beta} \log{(mn/\delta)} + \sqrt{\log{(1/\delta)}/(mn)})$.

2. We consider two variants of task-specific learning – based on regularized empirical risk minimization (RERM) and gradient descent (GD) – within our meta-learning framework. We apply our stability-based analysis to these variants to learning problems with convex, smooth losses and weakly convex, non-smooth losses. Our results are summarized in Table 1.

| Algorithm | Loss | Conditions | Uniform meta-stability $\bar{\beta}$ |
|---|---|---|---|
| Algo. 1 with RERM | convex, $G$-Lipschitz | $\gamma \leq \frac{1}{\lambda}$ | $\frac{G^2}{\lambda m} + \frac{G^2}{\lambda n}$ |
| Algo. 1 with RERM | convex, $H$-smooth, $M$-bounded | $\gamma \leq \frac{1}{\lambda}, \lambda \geq H$ | $\frac{HM}{\lambda(2n-1)} + \frac{HM}{\lambda(m+1)}$ |
| Algo. 1 with GD | convex, $G$-Lipschitz, $H$-smooth | $\eta \leq \frac{2}{H+2\lambda}, \gamma \leq \frac{1}{\lambda T}$ | $\frac{G^2}{\lambda m} + \frac{G^2}{\lambda n}$ |
| Algo. 1 with GD | $\rho$-weakly convex, $G$-Lipschitz | $\eta \leq \frac{1}{\lambda}, \gamma \leq \frac{1}{\lambda T}, \lambda \geq 2\rho$ | $G^2 \sqrt{\frac{\eta}{\lambda}} + \frac{G^2}{\lambda m} + \frac{G^2}{\lambda n}$ |
| Algo. 3 with GD | $\rho$-weakly convex, $G$-Lipschitz | $\eta \leq \frac{1}{\lambda}, \gamma \leq \frac{1}{\lambda T}, \lambda \geq 2\rho$ | $G^2 \sqrt{\frac{\eta}{\lambda}} + \frac{G^2}{\lambda m} + \frac{G^2}{\lambda n}$, w.h.p. |

Table 1: Bounds on uniform meta-stability $\bar{\beta}$ for different families of learning problems. Here, $\eta$ is the step-size for GD for task-specific learning, $\gamma$ is the step-size for GD for meta-parameter learning, $m$ is the number of tasks during training, $n$ is the number of training data for the task at test time.

3. We extend our results to stochastic and adversarially robust variants of our meta-learning algorithm.

## 1.1 Related Work

**Algorithmic Stability Analysis.** In many machine learning problems, standard learning theoretic tools, such as uniform convergence, do not apply since the associated complexity measures are unbounded or undefined (e.g., nearest neighbor classification), or yield guarantees that are not meaningful. Stability-based analysis is an alternative approach for obtaining generalization bounds

in such settings, introduced by Bousquet and Elisseeff [2002] and further developed in a long line of influential works [Elisseeff et al., 2005, Mukherjee et al., 2006, Shalev-Shwartz et al., 2010, Liu et al., 2017]. More recently, there have been significant breakthroughs in this field, with the work of Feldman and Vondrak [2018, 2019], Bousquet et al. [2020], Klochkov and Zhivotovskiy [2021], thereby improving the high probability bounds for uniformly stable learning algorithms beyond those established by Bousquet and Elisseeff [2002]. These results are complemented by Hardt et al. [2016], who provide the generalization bounds via algorithmic stability analysis of stochastic gradient for stochastic convex optimization with smooth loss functions. Subsequent work by Bassily et al. [2020] improves upon these results by removing the smoothness assumption, while Zhou et al. [2022], Lei [2023] advance the state-of-the-art by relaxing the convexity assumption.

**Theoretical Guarantees for Meta-Learning.** There has been significant progress in understanding the theoretical aspects of meta-learning, both in terms of convergence guarantees [Fallah et al., 2019, Ji et al., 2020, Mishchenko et al., 2023] and the generalization guarantees. The first generalization analysis can be traced back to Baxter [2000], who assumed that all tasks are sampled i.i.d. from the same task distribution. Subsequent works have enriched the guarantees through various learning theoretic constructs, including VC theory [Ben-David and Schuller, 2003, Maurer, 2009, Maurer et al., 2016], information-theoretic tools [Chen et al., 2021, Jose and Simeone, 2021, Jose et al., 2021, Rezazadeh et al., 2021, Hellström and Durisi, 2022], PAC-Bayes framework [Pentina and Lampert, 2014, Amit and Meir, 2018, Rothfuss et al., 2021, Farid and Majumdar, 2021, Liu et al., 2021, Ding et al., 2021, Rezazadeh, 2022, Riou et al., 2023, Zakerinia et al., 2024], etc. Other works that do not rely on the task distribution assumption instead choose to get a handle on the bound by defining certain metrics to measure either the task similarity [Du et al., 2020, Tripuraneni et al., 2020, Guan and Lu, 2021] or the divergence between the new tasks and the training sample for the training tasks [Fallah et al., 2021]. Finally, several works focus on the online meta-learning setting, also referred to as the lifelong learning [Pentina and Lampert, 2014, Balcan et al., 2019, Denevi et al., 2019a,b, Meunier and Alquier, 2021].

A prominent line of work, starting with that of Maurer [2005], focuses on giving theoretical guarantees for meta-learning via algorithmic stability analysis. More recently, Chen et al. [2020] establish connections between single-task learning with support/query (episodic) meta-learning algorithms, providing generalization gap of $\mathcal{O}(1/\sqrt{m})$ (where $m$ is the number of tasks) for smooth functions that is independent of the sample size $n$ – this was shown to be nearly optimal in Guan et al. [2022]. Subsequently, Fallah et al. [2021] show a bound of $\mathcal{O}(1/mn)$ for strongly convex functions and by leveraging a new notion of stability. Al-Shedivat et al. [2021] extend the result of Maurer [2005] to practical meta-learning algorithms for Lipschitz and smooth losses. Farid and Majumdar [2021] derive a PAC-Bayes bound to address the qualitatively different challenges of generalization within the task compared to that at the meta-level. Other relevant work includes analyzing the stability of bilevel optimization [Bao et al., 2021] and federated learning [Sun et al., 2024] for smooth functions.

## 2 Problem Setup and Preliminaries

**Notation.** Throughout the paper, we denote scalars and vectors with lowercase italics and lowercase bold Roman letters, respectively; e.g., $u$, $\mathbf{u}$. We work in a Euclidean space and use $\|\cdot\|$ and $\|\cdot\|_2$ to denote the $\ell_2$ norm. We use $[n]$ to represent the set $\{1, 2, \ldots, n\}$, and define $\mathrm{U}[n]$ to be the uniform distribution over $[n]$. Let $\Pi_{\mathcal{W}}$ be the Euclidean projection onto $\mathcal{W}$. We adopt the standard O-notation and use $\lesssim$ and $\mathcal{O}$ interchangeably. We use $\tilde{\mathcal{O}}$ to hide poly-logarithmic dependence on the parameters.

Let $\mathcal{X}, \mathcal{Y}$ denote the input and output spaces, respectively. Consider a supervised learning setting where each data point is denoted by $\mathbf{z} = (\mathbf{x}, y)$ drawn from some unknown distribution $\mathcal{D}$ over $\mathcal{Z} = \mathcal{X} \times \mathcal{Y}$. We consider a hypothesis space $\mathcal{H}$ (maps from $\mathcal{X} \to \mathcal{Y}$) parameterized by $\mathbf{w} \in \mathcal{W}$, where $\mathcal{W} \subseteq \mathbb{R}^d$ is a closed set with radius $D$. Let $\ell : \mathbb{R}^d \times \mathcal{Z} \to \mathbb{R}^+$ denote the loss function. We say that a loss function $\ell$ is $\underline{M\text{-bounded}}$ if $\forall \mathbf{w} \in \mathcal{W}, \forall \mathbf{z} \in \mathcal{D}, \ell(\mathbf{w}, \mathbf{z}) \leq M$; $\ell$ is $\underline{\mu\text{-strongly convex}}$ if $\forall \mathbf{w}_1, \mathbf{w}_2 \in \mathcal{W}, \forall \mathbf{z} \in \mathcal{D}, \ell(\mathbf{w}_1, \mathbf{z}) \geq \ell(\mathbf{w}_2, \mathbf{z}) + \langle \nabla \ell(\mathbf{w}_2, \mathbf{z}), \mathbf{w}_1 - \mathbf{w}_2 \rangle + \frac{\mu}{2} \|\mathbf{w}_1 - \mathbf{w}_2\|_2^2$; if $\mu = 0$, we say $\ell(\cdot, \mathbf{z})$ is convex. We say $\ell$ is $\underline{G\text{-Lipschitz continuous}}$ if $\forall \mathbf{w}_1, \mathbf{w}_2 \in \mathcal{W}, \forall \mathbf{z} \in \mathcal{D}, \|\ell(\mathbf{w}_1, \mathbf{z}) - \ell(\mathbf{w}_2, \mathbf{z})\|_2 \leq G \|\mathbf{w}_1 - \mathbf{w}_2\|_2$; $\ell$ is $\underline{H\text{-smooth}}$ if $, \forall \mathbf{w}_1, \mathbf{w}_2 \in \mathcal{W}, \forall \mathbf{z} \in \mathcal{D}, \|\nabla \ell(\mathbf{w}_1, \mathbf{z}) - \nabla \ell(\mathbf{w}_2, \mathbf{z})\|_2 \leq H \|\mathbf{w}_1 - \mathbf{w}_2\|_2$.

In a standard (single-task) learning setup, given a model $\mathbf{w}$, the expected loss on task $\mathcal{D}$ and the empirical loss on a training sample $\mathcal{S}$ drawn i.i.d. from $\mathcal{D}$, are defined, respectively, as follows.

$$L(\mathbf{w}, \mathcal{D}) = \mathbb{E}_{\mathbf{z} \sim \mathcal{D}} \left[ \ell(\mathbf{w}, \mathbf{z}) \right]; \quad L(\mathbf{w}, \mathcal{S}) = \frac{1}{n} \sum_{\mathbf{z} \in \mathcal{S}} \ell(\mathbf{w}, \mathbf{z}).$$

In a meta-learning framework, we consider distributions $\{\mathcal{D}_j\}_{j=1}^m$ associated with $m$ different tasks that are drawn from some (unknown) task distribution $\mu$. For each task $j$, we assume that the learner has access to $n$ training examples drawn i.i.d. from $\mathcal{D}_j$, i.e., $\mathcal{S}_j = \{\mathbf{z}_j^i\}_{i=1}^n \sim \mathcal{D}_j^n$. We denote the cumulative training data as $\mathbf{S} = \{\mathcal{S}_j\}_{j=1}^m$, and refer to it as the underline{meta-sample}.

A underline{meta-learning algorithm} $\mathcal{A}$ takes the meta-sample $\mathbf{S}$ as input and outputs an algorithm $\mathcal{A}(\mathbf{S})$ : $(\mathcal{X} \times \mathcal{Y})^n \to \mathcal{H}$. The performance of the meta-algorithm $\mathcal{A}$ is measured in terms of its ability to generalize w.r.t. loss $\ell(\cdot)$ to a new (previously unseen) task from the task distribution $\mu$; we also refer to it as the *transfer risk*:

$$L(\mathcal{A}(\mathbf{S}), \mu) = \mathbb{E}_{\mathcal{D} \sim \mu} \mathbb{E}_{\mathcal{S} \sim \mathcal{D}^n} L(\mathcal{A}(\mathbf{S})(\mathcal{S}), \mathcal{D}).$$

The goal of meta-learning is to learn a useful prior over tasks to help with rapid adaptation to new tasks. Formally, we pose the problem as learning a underline{meta-model}, parameterized by what we will refer to as underline{meta-parameter w}, that performs well on a variety of tasks. The hope is that the meta-parameter w can be adapted easily to a new task $\mathcal{D} \sim \mu$; in particular, that a task-specific model u can be quickly learned from a task-specific training set $\mathcal{S} \sim \mathcal{D}^n$ of size $n$ using the following proximal update:

$$\mathbf{u} = \operatorname{argmin}_{\mathbf{u} \in \mathcal{W}} L(\mathbf{u}, \mathcal{S}) + \frac{\lambda}{2} \|\mathbf{u} - \mathbf{w}\|^2 ,$$

where $\lambda > 0$ is a regularization parameter.

| **Algorithm 1** Prox Meta-Learning Algorithm $\mathcal{A}$ | **Algorithm 2** Task-specific Algorithm $\mathcal{A}_{\text{task}}$ |
|---|---|
| **Input:** Meta-sample $\mathbf{S} = \{\mathcal{S}_j\}_{j=1}^m$, epochs $T, K$, step sizes $\gamma, \eta$, regularization parameter $\lambda$ | **Input:** Pretrained model w, training data $\mathcal{S}$, #epochs $K$, step size $\eta$, reg. parameter $\lambda$ |
| 1: $\mathbf{w}_1 = 0$. | 1: Option 1 (RERM): |
| 2: **for** $t = 1, 2, \ldots, T$ **do** | 2: $\mathbf{u}(\mathbf{w}, \mathcal{S}) = \operatorname{argmin}_{\mathbf{u} \in \mathcal{W}} L(\mathbf{u}, \mathcal{S}) + \frac{\lambda}{2} \|\mathbf{u} - \mathbf{w}\|^2$. |
| 3:     **for** $j = 1, \ldots, m$ **do** | |
| 4:        $\mathbf{u}(\mathbf{w}_t, \mathcal{S}_j) = \mathcal{A}_{\text{task}}(\mathbf{w}_t, \mathcal{S}_j, K, \eta, \lambda)$ | 3: Option 2 (GD): Set $\mathbf{u}^{(1)}(\mathbf{w}, \mathcal{S}) = \mathbf{w}$ |
|        % Using Algorithm 2 | 4: **for** $t = 1, 2, \ldots, K - 1$ **do** |
| 5:     **end for** | 5:     $\mathbf{u}^{(k+1)}(\mathbf{w}, \mathcal{S}) = \mathbf{u}^{(k)}(\mathbf{w}, \mathcal{S})$ |
| 6:     Calculate the gradient, $\forall j \in [m]$, |        $- \eta(\nabla L(\mathbf{u}^{(k)}(\mathbf{w}, \mathcal{S}), \mathcal{S})$ |
|     $\nabla F_{\mathcal{S}_j}(\mathbf{u}(\mathbf{w}_t, \mathcal{S}_j), \mathbf{w}_t) = -\lambda(\mathbf{u}(\mathbf{w}_t, \mathcal{S}_j) - \mathbf{w}_t)$. |          $+ \lambda(\mathbf{u}^{(k)}(\mathbf{w}, \mathcal{S}) - \mathbf{w}))$ |
| 7:     Update $\mathbf{w}_{t+1} = \mathbf{w}_t - \frac{\gamma}{m} \sum_{j=1}^m \nabla F_{\mathcal{S}_j}(\mathbf{u}(\mathbf{w}_t, \mathcal{S}_j), \mathbf{w}_t)$ | 6:     $\mathbf{u}^{(k+1)}(\mathbf{w}, \mathcal{S}) = \Pi_{\mathcal{W}}(\mathbf{u}^{(k+1)}(\mathbf{w}, \mathcal{S}))$ |
| 8:     $\mathbf{w}_{t+1} = \Pi_{\mathcal{W}}(\mathbf{w}_{t+1})$ | 7: **end for** |
| 9: **end for** | 8: **return** Option 1 (RERM): $\mathbf{u}(\mathbf{w}, \mathcal{S})$ |
| 10: **return** $@\mathcal{A}_{\text{task}}(\mathbf{w}_{T+1}, \cdot, K, \eta, \lambda)$ |       Option 2 (GD): $\frac{1}{K} \sum_{k=1}^K \mathbf{u}^{(k)}(\mathbf{w}, \mathcal{S})$ |

The meta-parameter w itself is learned on the given meta-sample $\mathbf{S}$ by minimizing a regularized empirical loss averaged over tasks, where the regularization term penalizes the task-specific models in proportion to the $\ell_2$ distance from the meta-parameter [Zhou et al., 2019]:

$$\widehat{\mathbf{w}} = \operatorname*{argmin}_{\mathbf{w} \in \mathcal{W}} \frac{1}{m} \sum_{j=1}^m \min_{\mathbf{u} \in \mathcal{W}} F_{\mathcal{S}_j}(\mathbf{u}, \mathbf{w}) := \operatorname*{argmin}_{\mathbf{w} \in \mathcal{W}} \frac{1}{m} \sum_{j=1}^m \min_{\mathbf{u} \in \mathcal{W}} \left[ L(\mathbf{u}, \mathcal{S}_j) + \frac{\lambda}{2} \|\mathbf{u} - \mathbf{w}\|^2 \right]. \quad (1)$$

The formulation above involves a bi-level optimization problem. The upper-level optimization involves finding the meta-parameter w which requires solving the lower-level optimization problem of finding task-specific model parameters u. We consider both Gradient Descent (GD) as well Regularized Empirical Risk Minimization (RERM) for task-specific learning (see Algorithm 2 for more details); for meta-learning we employ a gradient descent method (see Algorithm 1).

We would like to bound the transfer risk in terms of the *empirical multi-task risk*:

$$L(\mathcal{A}(\mathbf{S}), \mathbf{S}) = \frac{1}{m} \sum_{j=1}^m L(\mathcal{A}(\mathbf{S})(\mathcal{S}_j), \mathcal{S}_j).$$

To do so, we rely on the stability of the meta-learning algorithm.

**Stability of Meta-Learning Algorithm.** Given a meta-sample $\mathbf{S} = \{\mathcal{S}_j\}_{j=1}^m$, define $\mathbf{S}^{(j)}$ to be the meta-sample obtained by replacing the training samples $\mathcal{S}_j$ for the $j$-th task, in $\mathbf{S}$, by another i.i.d. sample $\mathcal{S}_j' \sim \mathcal{D}_j^n$. We refer to $\mathbf{S}, \mathbf{S}^{(j)}$ as underline{neighboring meta-samples}. For a task-specific training sample $\mathcal{S} = \{\mathbf{z}^i\}_{i=1}^n$, let $\mathcal{S}^{(i)}$ denote the training data obtained by replacing the $i$-th example $\mathbf{z}^i \in \mathcal{S}$ by another example $\mathbf{z}' \sim \mathcal{D}$ drawn independently; we refer to $\mathcal{S}, \mathcal{S}^{(i)}$ as underline{neighboring samples}.

**Theorem 2.1** (Maurer [2005])**.** Suppose the meta-algorithm $\mathcal{A}$ satisfies:

1. (Uniform Stability of Single-Task Learning) For any meta-sample $\mathbf{S}$ and any $\mathcal{S}, \mathcal{S}^{(i)}$,
$$\left| \ell(\mathcal{A}(\mathbf{S})(\mathcal{S}), \mathbf{z}) - \ell(\mathcal{A}(\mathbf{S})(\mathcal{S}^{(i)}), \mathbf{z}) \right| \leq \beta.$$

2. (Uniform Stability of Meta-Learning) For any $\mathbf{S}, \mathbf{S}^{(j)}$ and any given training set $\mathcal{S} \sim \mathcal{D}$,
$$\left| L(\mathcal{A}(\mathbf{S})(\mathcal{S}), \mathcal{S}) - L(\mathcal{A}(\mathbf{S}^{(j)})(\mathcal{S}), \mathcal{S}) \right| \leq \beta'.$$

Then, for $M$-bounded loss $\ell$, with probability at least $1 - \delta$, we have that
$$L(\mathcal{A}(\mathbf{S}), \mu) \lesssim L(\mathcal{A}(\mathbf{S}), \mathbf{S}) + (m\beta' + M)\sqrt{\log(1/\delta)/m} + \beta.$$

Theorem 2.1 follows using a simple extension of arguments in Bousquet and Elisseeff [2002]. By utilizing sharper bounds tailored for uniformly stable algorithms [Bousquet et al., 2020], a tighter bound can be achieved, as demonstrated in Theorem 2.2 below. A similar result was shown in Guan et al. [2022] for episodic training algorithms (except there is no $\beta$).

**Theorem 2.2.** Suppose the meta-algorithm $\mathcal{A}$ satisfies the same conditions as shown in Theorem 2.1. Then for $M$-bounded loss $\ell$, with probability at least $1 - \delta$, we have that
$$L(\mathcal{A}(\mathbf{S}), \mu) \lesssim L(\mathcal{A}(\mathbf{S}), \mathbf{S}) + \beta' \log(m) \log(1/\delta) + M\sqrt{\log(1/\delta)/m} + \beta.$$

## 3  Uniform Meta-Stability

Motivated by prior work (i.e., Theorem 2.1 and the definitions therein), we introduce a new notion of stability which measures the sensitivity of the learning algorithm as we replace both a task in the meta-sample as well as a single training example available for the task at test time.

**Definition** (Uniform Meta-Stability). We say that a meta-learning algorithm $\mathcal{A}$ is $\bar{\beta}$-uniformly meta-stable if for any neighbouring meta-samples $\mathbf{S}, \mathbf{S}^{(j)}$, and neighboring samples $\mathcal{S}, \mathcal{S}^{(i)}$, for any task $\mathcal{D} \sim \mu$ and any $\mathbf{z} \sim \mathcal{D}$, we have that
$$\left| \ell(\mathcal{A}(\mathbf{S})(\mathcal{S}), \mathbf{z}) - \ell(\mathcal{A}(\mathbf{S}^{(j)})(\mathcal{S}^{(i)}), \mathbf{z}) \right| \leq \bar{\beta}.$$

The definition above is rather natural. Intuitively, for a meta-learning algorithm to transfer well, we require that the learning algorithms, i.e., $\mathcal{A}(\mathbf{S})$ and $\mathcal{A}(\mathbf{S}')$, returned on two neighboring meta-samples, when trained on two neighboring samples return models that predict similarly. Our first result bounds the generalization gap in terms of the uniform meta-stability parameter.

**Theorem 3.1.** Consider a meta-learning problem for some $M$-bounded loss function $\ell$ and task distribution $\mu$. Let $\mathbf{S}$ be a meta-sample consisting of training samples on $m$ tasks each of size $n$, and let $\mathcal{S} \sim \mathcal{D}$ be a sample of size $n$ on a previously unseen task $\mathcal{D} \sim \mu$. Then, for any $\beta$-uniformly meta-stable learning algorithm $\mathcal{A}$, we have that with probability $1 - \delta$,
$$L(\mathcal{A}(\mathbf{S}), \mu) \lesssim L(\mathcal{A}(\mathbf{S}), \mathbf{S}) + \bar{\beta} \log(mn) \log(1/\delta) + M\sqrt{\log(1/\delta)/(mn)}.$$

The result above is a direct analogue of Theorem 2.1 with stability parameters $\beta, \beta'$ both subsumed into a single meta-stability parameter. We do obtain a faster rate of convergence – as we instantiate concrete algorithms and specialize our results to specific problems in Section 4.1, we will see a notable improvement in rates from $1/\sqrt{m}$ to $1/m$, for $n > m$.

We conclude the section by presenting an alternate notion of algorithmic meta-stability and a basic result that directly bounds the generalization gap for the meta-learning problem.

**Definition** (On-Average Meta-Stability). Let $\mu$ be an (unknown) underlying task distribution. We say that a meta-learning algorithm $\mathcal{A}$ is $\bar{\beta}$-on-average-replace-one-meta-stable if
$$\mathbb{E}_{\mathbf{S} \sim \{\mathcal{D}_j^n\}_{j=1}^m, (\mathcal{S}_j', \mathbf{z}_j') \sim \mathcal{D}_j^{n+1}, \{\mathcal{D}_j\}_{j=1}^m \sim \mu^m, j \sim \mathrm{U}[m], i \sim \mathrm{U}[n]} \left| \ell(\mathcal{A}(\mathbf{S})(\mathcal{S}_j), \mathbf{z}_j^i) - \ell(\mathcal{A}(\mathbf{S}^{(j)})(\mathcal{S}_j^{(i)}), \mathbf{z}_j^i) \right| \leq \bar{\beta}.$$

**Theorem 3.2.** Let $\mu$ be an underlying task distribution. Given a meta-sample $\mathbf{S}$, test task $\mathcal{D} \sim \mu$, and $\mathcal{S} \sim \mathcal{D}^n$, for any $\bar{\beta}$-on-average-replace-one-meta-stable meta-learning algorithm $\mathcal{A}$, we have that
$$\mathbb{E}_{\mathbf{S} \sim \{\mathcal{D}_j^n\}_{j=1}^m, \{\mathcal{D}_j\}_{j=1}^m \sim \mu^m} [L(\mathcal{A}(\mathbf{S}), \mu) - L(\mathcal{A}(\mathbf{S}), \mathbf{S})] \leq \bar{\beta}.$$

## 4  Bounding Transfer Risk

In this section, we consider a concrete meta-learning algorithm given in Algorithm 1.

## 4.1 Convex and Smooth Losses

We begin with meta-learning problems with convex, Lipschitz (and potentially smooth) losses.

**Lemma 4.1.** Assume that the loss function $\ell$ is convex and $G$-Lipschitz loss. Let $\mathbf{S}$, $\mathbf{S}^{(j)}$ denote neighboring meta-samples and $\mathcal{S}$, $\mathcal{S}^{(i)}$ the neighboring samples on a test task. Then, the following holds for Algorithm 1 with RERM for task-specific learning (i.e., Option 1 for Algorithm 2) $\forall T \geq 1$,

$$\sup_{\mathbf{S},\mathcal{S},j\in[m],i\in[n]} \left\| \mathcal{A}(\mathbf{S})(\mathcal{S}) - \mathcal{A}(\mathbf{S}^{(j)})(\mathcal{S}^{(i)}) \right\| \leq \frac{G}{\lambda m} + \frac{2G}{\lambda n}.$$

Further, if $\ell$ is convex, $M$-bounded and $H$-smooth, then setting $\lambda \geq H$, $\gamma \leq \frac{1}{\lambda}$, we have $\forall T \geq 1$,

$$\sup_{\mathbf{S},\mathcal{S},j\in[m],i\in[n]} \left\| \mathcal{A}(\mathbf{S})(\mathcal{S}) - \mathcal{A}(\mathbf{S}^{(j)})(\mathcal{S}^{(i)}) \right\| \leq \frac{2\sqrt{2HM}}{2\lambda n - H} + \frac{n}{2\lambda n - H}\frac{4\sqrt{2HM}}{(m+1)}.$$

We can now use the result above with Theorem 3.1 to get the following bound on the transfer risk.

**Theorem 4.2.** The following holds for Algorithm 1 with step-size $\gamma \leq \frac{1}{\lambda}$ on a given meta-sample $\mathbf{S}$, and RERM for task-specific learning (i.e., Option 1 for Algorithm 2), for all $T \geq 1$:

1. For convex, $M$-bounded, and $G$-Lipschitz loss functions, with probability at least $1 - \delta$

$$L(\mathcal{A}(\mathbf{S}),\mu) \lesssim L(\mathcal{A}(\mathbf{S}),\mathbf{S}) + \left( \frac{G^2}{\lambda n} + \frac{G^2}{\lambda m} \right) \log(mn) \log(1/\delta) + \frac{M\sqrt{\log(1/\delta)}}{\sqrt{mn}}.$$

2. For convex, $M$-bounded, and $H$-smooth loss functions ($H \leq \lambda$), with probability at least $1 - \delta$

$$L(\mathcal{A}(\mathbf{S}),\mu) \lesssim L(\mathcal{A}(\mathbf{S}),\mathbf{S}) + \left( \frac{HM}{(2n-1)\lambda} + \frac{HM}{(m+1)\lambda} \right) \log(mn) \log(1/\delta) + \frac{M\sqrt{\log(1/\delta)}}{\sqrt{mn}}.$$

Next, we give analogous results for GD for task-specific learning (i.e., Option 2 for Algorithm 2), albeit for smooth loss functions. Lemma 4.3 bounds the output sensitivity of the meta-learning algorithm. We use it with Theorem 3.1 to give the generalization guarantee in Theorem 4.4.

**Lemma 4.3.** Assume that the loss function is convex, $G$-Lipschitz and $H$-smooth. Let $\mathbf{S}$, $\mathbf{S}^{(j)}$ denote neighboring meta-samples and $\mathcal{S}$, $\mathcal{S}^{(i)}$ the neighboring samples on a test task. Then the following holds for Algorithm 1 with GD for task-specific learning (i.e., Option 2 for Algorithm 2) with $\eta \leq \frac{2}{H+2\lambda}$, for all $T \geq 1$ as long as we set $\gamma \leq \frac{1}{\lambda T}$,

$$\sup_{\mathbf{S},\mathcal{S},j\in[m],i\in[n]} \left\| \mathcal{A}(\mathbf{S})(\mathcal{S}) - \mathcal{A}(\mathbf{S}^{(j)})(\mathcal{S}^{(i)}) \right\| \leq \frac{4eG}{\lambda m} + \frac{2G}{\lambda n}.$$

**Theorem 4.4.** Assume that the loss function is convex, $M$-bounded, $G$-Lipschitz and $H$-smooth. Suppose we run Algorithm 1 for $T$ iterations with $\gamma \leq \frac{1}{\lambda T}$ on a given meta-sample $\mathbf{S}$, and GD for task-specific learning (Option 2, Algorithm 2) with $\eta \leq \frac{2}{H+2\lambda}$. Then, with probability at least $1 - \delta$,

$$L(\mathcal{A}(\mathbf{S}),\mu) \lesssim L(\mathcal{A}(\mathbf{S}),\mathbf{S}) + \left( \frac{G^2}{\lambda m} + \frac{G^2}{\lambda n} \right) \log(mn) \log(1/\delta) + \frac{M\sqrt{\log(1/\delta)}}{\sqrt{mn}}.$$

The results above show that meta-stable learning algorithms do not overfit. The bound on the generalization gap of $\tilde{\mathcal{O}}(\frac{1}{m} + \frac{1}{n} + \frac{1}{\sqrt{mn}})$ is tighter than what we would obtain using prior work. Indeed, we show that Theorem 2.2 yields a rate of $\tilde{\mathcal{O}}(\frac{1}{m} + \frac{1}{n} + \frac{1}{\sqrt{m}})$ (see Theorems C.2 and C.3 in Appendix), which is worse for all $m \leq n^2$. Notably, the bounds on the generalization gap are independent of the number of iterations of the meta learning Algorithm 1 and the number of iterations of GD for Algorithm 2. This holds since the objective we are minimizing is strongly convex (given the strongly convex regularizer), which ensures that the output sensitivity (in Lemmas 4.3 and 4.1 are independent of $T$ and $K$. In itself, this should not be surprising since we only bound the generalization error in terms of the empirical error – the latter may not be small unless the algorithms have converged. To get a better handle on the generalization error we focus on excess (transfer) risk bounds in Section 4.3. But first we give a similar development for another important problem class.

## 4.2 Weakly Convex and Non-smooth Losses

Here, we focus on a more practical setting of learning problems with loss functions that are weakly convex and non-smooth. The notion of weak convexity is often used in non-convex optimization literature in a variety of problems including robust phase retrieval [Davis et al., 2020] and dictionary learning [Davis and Drusvyatskiy, 2019]; see Drusvyatskiy [2017] for an extended discussion.

**Definition.** A function $f(\mathrm{w})$ is $\rho$-*weakly convex* w.r.t. $\|\cdot\|$ if $f(\mathrm{w}) + \frac{\rho}{2}\|\mathrm{w}\|^2$ is convex in w.

The class of weakly convex functions is contained within the larger class of non-smooth functions and semi-smooth functions [Mifflin, 1977]. It includes convex functions and smooth functions with Lipschitz continuous gradient as special cases; $\rho < 0$ implies that the function is strongly convex. An important example from a practical perspective is that of training over-parameterized two-layer neural networks with smooth activation functions using a smooth loss [Richards and Rabbat, 2021]. We first bound the sensitivity of Algorithm 1 for weakly convex and non-smooth losses.

**Lemma 4.5.** Assume that the loss function is $\rho$-weakly convex and $G$-Lipschitz. Let $\mathbf{S}$, $\mathbf{S}^{(j)}$ denote neighboring meta-samples and $\mathcal{S}$, $\mathcal{S}^{(i)}$ the neighboring samples on a test task. Then the following holds for Algorithm 1 with $\lambda \geq 2\rho$, and GD for task-specific learning (i.e., Option 2 for Algorithm 2) with $\eta \leq \frac{1}{\lambda}$, for all $T \geq 1$ as long as we set $\gamma \leq \frac{1}{\lambda T}$,

$$\sup_{\mathbf{S},\mathcal{S},j\in[m],i\in[n]} \left\| \mathcal{A}(\mathbf{S})(\mathcal{S}) - \mathcal{A}(\mathbf{S}^{(j)})(\mathcal{S}^{(i)}) \right\| \leq (8eG+2G)\sqrt{\frac{\eta}{\lambda}} + \frac{8eG}{\lambda m} + \frac{8G}{\lambda n}.$$

Using the result above in conjunction with Thm 3.1 gives the following bound on the transfer risk.

**Theorem 4.6.** Assume that the loss function is $\rho$-weakly convex, $M$-bounded, and $G$-Lipschitz. Suppose we run Algorithm 1 for $T$ iterations with $\gamma \leq \frac{1}{\lambda T}, \lambda \geq 2\rho$ on a meta-sample $\mathbf{S}$, and GD for task-specific learning (Option 2, Algorithm 2) with $\eta \leq \frac{1}{\lambda}$, Then, with probability at least $1 - \delta$,

$$L(\mathcal{A}(\mathbf{S}), \mu) \lesssim L(\mathcal{A}(\mathbf{S}), \mathbf{S}) + \left( G^2\sqrt{\frac{\eta}{\lambda}} + \frac{G^2}{\lambda m} + \frac{G^2}{\lambda n} \right) \log(mn)\log(1/\delta) + \frac{M\sqrt{\log(1/\delta)}}{\sqrt{mn}}.$$

Proof of Theorem 4.6 follows from Lemma 4.5 and Theorem 3.1. A few remarks are in order.

For learning rate $\gamma \leq \frac{1}{\lambda T}$, Theorem 4.6 gives a rate of $\tilde{\mathcal{O}}(\sqrt{\eta} + \frac{1}{m} + \frac{1}{n} + \frac{1}{\sqrt{mn}})$ on the generalization gap. This naturally suggests setting $\eta = \frac{1}{\lambda K}$, where $K \geq \min\{m,n\}$ is the number of iterations of GD in task-specific learning. Then, similar to the discussion in Section 4.1, Theorem 4.6 gives a tighter bound, when $n > m$, than those derived using prior work (Theorem 2.2); we refer the reader to Theorem D.4 in the appendix for further details.

Our proof technique shares similarities with Bassily et al. [2020]. However, our result is not a straightforward application of theirs as we deal with a bi-level optimization problem and focus on weakly convex functions. It is worth noting that our results for weakly convex non-smooth losses require regularization parameter $\lambda \geq 2\rho$, which can be chosen in practice using cross-validation.

The work most related to ours is that of Guan et al. [2022]. However, our results are fundamentally different from theirs in several aspects. Firstly, the algorithms we study are different. Guan et al. [2022] focus on support/query (S/Q) training strategies (aka episodic training) where each task $\mathcal{S}_j$ is split into two non-overlapping parts – the support set $\mathcal{S}_j^{tr}$ for training the task-specific parameter and the query set $\mathcal{S}_j^{ts}$ for measuring the algorithm's performance [Vinyals et al., 2016]. The meta-parameter is learned by minimizing the loss computed over the query set. Such S/Q training strategy is popular for modern gradient-based meta-learning algorithm such as MAML for few-shot learning [Finn et al., 2017], where the optimization objective can be written as $\min_{\mathrm{w}} \frac{1}{m} \sum_{j=1}^{m} L(\mathrm{w} - \nabla L(\mathrm{w}, \mathcal{S}_j^{tr}), \mathcal{S}_j^{ts})$. One notable limitation is that Guan et al. [2022] assume that the loss function on the task level, e.g., $R(\mathrm{w}, \mathcal{S}_j) = L(\mathrm{w} - \nabla L(\mathrm{w}, \mathcal{S}_j^{tr}), \mathcal{S}_j^{ts})$, is convex or (Hölder) smooth. Such an assumption is highly impractical, as demonstrated by [Mishchenko et al., 2023, Theorem 1, Theorem 2], which provides several counterexamples where $L$ is convex and smooth but $R$ is neither convex nor smooth. In contrast, we directly deal with $L$ being weakly convex and nonsmooth. Our approach requires a more involved proof that deals with stability of bi-level optimization. This is in stark contrast with Guan et al. [2022] who directly reduce the meta-learning problem to a single-task learning problem without considering the bi-level structure of the problem.

The work of Fallah et al. [2021] proposed a notion of stability similar to ours. The difference is that they consider S/Q training and define the stability by changing a mini-batch of samples in $\mathcal{S}_j^{tr}$ as

well as a single sample in $\mathcal{S}_j^{ts}$. Moreover, their focus is primarily on strongly convex losses. They discuss generalization to training tasks and unseen tasks separately, as they do not assume all tasks are sampled from the same task distribution. Another related work of Guan and Lu [2021] present a generalization bound of $\mathcal{O}(\sqrt{C/mn})$ under a task relatedness assumption, where $C$ captures the logarithm of the covering number of hypothesis class that possibly depends on the dimension $d$. More recently, Riou et al. [2023] provide generalization bounds with a fast rate of $\mathcal{O}(\frac{1}{m} + \frac{1}{n})$, albeit under an additional extended Bernstein's condition.

## 4.3 Excess Transfer Risk

In the previous sections, we focused on establishing that meta-stable rules do not overfit to the meta-sample. In this Section, we focus on the question of whether meta-learning Algorithm 1 can achieve a small generalization error, i.e., are they guaranteed to transfer well on unseen tasks? We show that by focusing on the computational aspects, i.e., by bounding the optimization error in terms of the number of iterations. Furthermore, we give bounds on excess risk, wherein the benchmark is the performance of the best possible in-class predictor.

Let $\mathbf{u}_* = \operatorname{argmin}_{\mathbf{u} \in \mathcal{W}} L(\mathbf{u}, \mathcal{D})$, $\mathbf{u}_j^* = \operatorname{argmin}_{\mathbf{u} \in \mathcal{W}} L(\mathbf{u}, \mathcal{S}_j)$, $\forall j \in [m]$ be the optimal task-specific hypotheses for the unseen task and the given training tasks, respectively. Given a meta-algorithm $\mathcal{A}$, the excess transfer risk can be decomposed as follows:

$$\underbrace{L(\mathcal{A}(\mathbf{S})(\mathcal{S}),\mathcal{D}) - L(\mathbf{u}_*,\mathcal{D})}_{\text{Excess Transfer Risk } \mathcal{E}_{\text{risk}}(\mathcal{A})} = \underbrace{L(\mathcal{A}(\mathbf{S})(\mathcal{S}),\mathcal{D}) - \frac{1}{m}\sum_{j=1}^{m} L(\mathcal{A}(\mathbf{S})(\mathcal{S}_j),\mathcal{S}_j)}_{\text{Generalization Gap } \mathcal{E}_{\text{gen}}(\mathcal{A})} + \underbrace{\frac{1}{m}\sum_{j=1}^{m}\big[L(\mathcal{A}(\mathbf{S})(\mathcal{S}_j),\mathcal{S}_j) - L(\mathbf{u}_j^*,\mathcal{S}_j)\big]}_{\text{Optimization and Approximation Error } \mathcal{E}_{\text{opt+app}}(\mathcal{A})}$$

$$+ \underbrace{\frac{1}{m}\sum_{j=1}^{m}\big[L(\mathbf{u}_j^*,\mathcal{S}_j) - L(\mathbf{u}_*,\mathcal{S}_j)\big]}_{\leq 0} + \underbrace{\frac{1}{m}\sum_{j=1}^{m}\big[L(\mathbf{u}_*,\mathcal{S}_j) - L(\mathbf{u}_*,\mathcal{D})\big]}_{\mathbb{E}_{\forall j \in [m], \mathcal{S}_j \sim \mathcal{D}_j^n, \mathcal{D}_j \sim \mu, \mathcal{D} \sim \mu} = 0}.$$

To control excess risk, we need to bound $\mathcal{E}_{\text{gen}}(\mathcal{A})$ and $\mathcal{E}_{\text{opt+app}}(\mathcal{A})$ simultaneously. The bounds on the first term are presented in the previous section. Here, we focus on analyzing the second term.

**Theorem 4.7.** Assume that the loss $\ell$ is convex and $G$-Lipschitz. Define $\mathbf{u}_j^* = \operatorname{argmin}_{\mathbf{u}} L(\mathbf{u},\mathcal{S}_j), \forall j \in [m]$. Suppose we run Algorithm 1 for $T$ iterations with step-size $\gamma = \frac{1}{\lambda T}$, and using GD for task-specific learning (i.e., Option 2 for Algorithm 2), to find an algorithm $\mathcal{A}(\mathbf{S}) = \mathcal{A}_{\text{task}}(\mathbf{w}_{T+1}, \cdot)$ which is then run on $\mathcal{S}_j$ for $K$ iterations with step-size $\eta \leq \frac{1}{2\lambda}$. Then, we have that

$$L(\mathcal{A}(\mathbf{S})(\mathcal{S}_j),\mathcal{S}_j) - \inf_{\mathbf{u}} L(\mathbf{u},\mathcal{S}_j) \lesssim \frac{D^2}{\eta K} + G^2\eta + GD\eta\lambda + \lambda \|\mathbf{w}_{T+1} - \widehat{\mathbf{w}}\|^2 + \lambda\sigma^2$$

where $\widehat{\mathbf{w}}$ is defined in Equation (1). Here $\sigma^2 := \frac{1}{m}\sum_{j=1}^{m} \|\widehat{\mathbf{w}} - \mathbf{u}_j^*\|^2$ is the approximation error, and $\|\mathbf{w}_{T+1} - \widehat{\mathbf{w}}\|^2 \lesssim \frac{1}{T}(D^2 + \frac{D^2}{\lambda\eta K} + \frac{\eta(G+2\lambda D)^2}{\lambda})$ is the optimization error.

Finally, to bound the excess transfer risk for convex and non-smooth losses, we use Theorem 4.6 with Theorem 4.7 to get that in expectation over the sampling of data (meta-sample $\mathbf{S}$ and sample $\mathcal{S}$)

$$\mathbb{E}[\mathcal{E}_{\text{risk}}(\mathcal{A})] \leq \mathbb{E}[\mathcal{E}_{\text{gen}}(\mathcal{A})] + \mathbb{E}[\mathcal{E}_{\text{opt+app}}(\mathcal{A})] \lesssim G^2\sqrt{\frac{\eta}{\lambda}} + \frac{G^2}{\lambda m} + \frac{G^2}{\lambda n} + \frac{D^2}{\eta K} + G^2\eta + GD\eta\lambda + \frac{\lambda D^2}{T} + \eta(G+2\lambda D)^2 + \lambda\sigma^2.$$

By properly choosing step size $\eta = \mathcal{O}\left(\frac{1}{\lambda K^{2/3}}\right)$, we obtain that the expected excess transfer risk decays at a rate of $\mathcal{O}(\frac{1}{\lambda K^{1/3}} + \frac{1}{\lambda m} + \frac{1}{\lambda n} + \frac{\lambda}{T} + \lambda\sigma^2)$. Similarly, for convex, Lipschitz and smooth losses, applying Theorem 4.4 with Theorem 4.7 and selecting $\eta = \mathcal{O}(\frac{1}{\lambda\sqrt{K}})$ results in an expected excess transfer risk of $\mathcal{O}(\frac{1}{\lambda\sqrt{K}} + \frac{1}{\lambda m} + \frac{1}{\lambda n} + \frac{\lambda}{T} + \lambda\sigma^2)$. Therefore, as $K, T, m, n$ tend to infinity, the excess risk converges to $\sigma^2$. As $\sigma$ represents the average distance between the optimal task-specific parameters $\mathbf{u}_j$'s and the optimal estimated meta-parameter $\widehat{\mathbf{w}}$, the excess risk is small when $\sigma$ is small. It is also typical to set the regularization parameter $\lambda$ inversely proportional to the sample size $n$ (e.g., $\lambda = \mathcal{O}(1/\sqrt{n})$).

Denevi et al. [2019a] study the same algorithm as ours except in the online setting. However, the function classes they consider are limited to compositions of linear hypothesis classes with convex and closed losses. In contrast, our work considers a broader range of functions, encompassing not only convex, Lipschitz, and smooth functions but also weakly-convex and non-smooth functions. The

bound on expected excess risk shown in Denevi et al. [2019a] takes the form $\mathcal{O}(\frac{\text{Var}_m}{\sqrt{n}} + \frac{1}{\sqrt{m}})$, where $\text{Var}_m$ captures the relatedness among the tasks sampled from the task environment. Unfortunately, this bound relies on a specific choice of $\lambda = \mathcal{O}\left(\frac{1}{\text{Var}_m}\sqrt{\frac{\log(n)}{n}}\right)$, which depends on $\text{Var}_m$ – a quantity that is often not known a priori in practice. To compare with our work, set $K = n$, $T = m$, $\eta = \mathcal{O}(1/\sqrt{n})$, and $\lambda = \mathcal{O}(1/\sqrt{n})$. Then, applying Theorem 4.4 with Theorem 4.7, we obtain that $\mathbb{E}[\mathcal{E}_{\text{risk}}(\mathcal{A})] \lesssim \frac{\sqrt{n}}{m} + \frac{\max(1,\sigma^2)}{\sqrt{n}}$. Considering both $\text{Var}_m$ and $\sigma$ as constants, the bound on expected excess risk based on our analysis is tighter than that of Denevi et al. [2019a] when $n \lesssim m$, a common setting studied in meta-learning framework.

We also conduct a simple experiment to empirically verify the tightness of our generalization bounds, which we defer to Appendix A due to space limitations.

# 5 Implications of the Generalization Bounds

Next, we present stochastic and adversarially robust variants of the meta-learning Algorithm 1.

## 5.1 Proximal Meta-Learning with Stochastic Optimization

We adapt Algorithm 1 to utilize sampling-with-replacement where at each iteration we process the training set of a single task; see Algorithm 3 for more details. We show that with high probability the sensitivity of this stochastic meta-learning algorithm is bounded.

**Lemma 5.1.** Assume that the loss function is $\rho$-weakly convex and $G$-Lipschitz. Let $\mathbf{S}$, $\mathbf{S}^{(j)}$ denote neighboring meta-samples and $\mathcal{S}$, $\mathcal{S}^{(i)}$ the neighboring samples on a test task. Then, with probability at least $1 - \exp\left(-T^2 e^2/m^2\right)$, the following holds for Algorithm 3 with $\lambda \geq 2\rho$, and GD for task-specific learning (i.e., Option 2 for Algorithm 2) with $\eta \leq \frac{1}{\lambda}$, for all $T \geq 1$ as long as we set $\gamma \leq \frac{1}{\lambda T}$,

$$\sup_{\mathbf{S},\mathcal{S},i\in[n],j\in[m]} \left\|\mathcal{A}(\mathbf{S})(\mathcal{S}) - \mathcal{A}(\mathbf{S}^{(j)})(\mathcal{S}^{(i)})\right\| \leq (8eG + 2G)\sqrt{\frac{\eta}{\lambda}} + \frac{8eG}{\lambda m} + \frac{8G}{\lambda n}.$$

## 5.2 Robust Adversarial Proximal Meta-Learning

We consider inference-time adversarial attacks with a general threat model $\mathcal{B} : \mathcal{X} \to 2^{\mathcal{X}}$. Specifically, given an input example $\mathbf{x} \in \mathcal{X}, \mathcal{B}(\mathbf{x}) \subseteq \mathbb{R}^d$ represents the set of all possible perturbations of $\mathbf{x}$ that an adversary can choose from. This includes the typical examples such as the $L_p$ threat models that are often considered in practice, or a discrete set of designed transformations.

Given a model parameter $\mathbf{w}$, let $\tilde{\ell}(\mathbf{w}, \mathbf{z}) = \max_{\tilde{\mathbf{z}} \in \mathcal{B}(\mathbf{z})} \ell(\mathbf{w}, \tilde{\mathbf{z}})$ denote the adversarial loss. We adapt the standard meta-learning framework simply by considering the robust variant, $\tilde{\ell}$, of the standard loss $\ell$. We denote the robust transfer risk and empirical robust multi-task risk as $L_{\text{rob}}(\mathcal{A}(\mathbf{S}), \mu)$ and

---

**Algorithm 3** Stochastic Prox Meta-Learning

**Input:** Meta-sample $\mathbf{S} = \{\mathcal{S}_j\}_{j=1}^m$, epochs $T, K$, step size $\gamma, \eta$, regularization parameter $\lambda$.
1: $\mathbf{w}_1 = 0$.
2: **for** $t = 1, 2, \ldots, T$ **do**
3:      Sample $j_t \sim \text{U}[m]$.
4:      $\mathbf{u}(\mathbf{w}_t, \mathcal{S}_{j_t}) = \mathcal{A}_{\text{task}}(\mathbf{w}_t, \mathcal{S}_{j_t}, K, \eta, \lambda)$.
     % Using Algorithm 2
5:      Calculate the gradient
     $\nabla F_{\mathcal{S}_{j_t}}(\mathbf{u}(\mathbf{w}_t, \mathcal{S}_{j_t}), \mathbf{w}_t) = -\lambda(\mathbf{u}(\mathbf{w}_t, \mathcal{S}_{j_t}) - \mathbf{w}_t)$.
6:      Update $\mathbf{w}_{t+1} = \mathbf{w}_t - \gamma \nabla F_{\mathcal{S}_{j_t}}(\mathbf{u}(\mathbf{w}_t, \mathcal{S}_{j_t}), \mathbf{w}_t)$.
7:      $\mathbf{w}_{t+1} = \Pi_{\mathcal{W}}(\mathbf{w}_{t+1})$.
8: **end for**
9: **return** $@\mathcal{A}_{\text{task}}(\mathbf{w}_{T+1}, \cdot, K, \eta, \lambda)$

---

$L_{\text{rob}}(\mathcal{A}(\mathbf{S}), \mathbf{S})$. Now, given meta-sample $\mathbf{S}$, the goal is to learn a robust prior (e.g., a pre-trained model) for rapid adaptation to and robust generalization on new tasks. We adopt the framework presented in Section 2 except we use robust loss for task-specific training; indeed, using GD (Option 2) on robust loss in Algorithm 2 yields adversarial training. We use Algorithm 1 for meta-learning. We now relate a loss function with its adversarially robust counterpart.

**Proposition 5.2.** Given a loss function $\ell(\cdot, \mathbf{z})$ and its adversarial counterpart $\tilde{\ell}(\cdot, \mathbf{z})$, the following holds: (1) If $\ell$ is $G$-Lipschitz (in its first argument), then $\tilde{\ell}$ is $G$-Lipschitz. (2) $\tilde{\ell}$ is **not** $H$-smooth even if $\ell$ is $H$-smooth. (3) If $\ell$ is $H$-smooth in $\mathbf{w}$, then $\tilde{\ell}$ is $H$-weakly convex in $\mathbf{w}$.

Using the result above with Theorem 3.1 yields the following bound on robust (transfer) risk.

**Corollary 5.3.** Assume that the loss $\ell$ is $M$-bounded and $H$-smooth. Suppose we run Algorithm 1 for $T$ iterations with $\gamma \leq \frac{1}{\lambda T}, \eta \leq \frac{1}{\lambda}, \lambda > 2H$, and wherein task-specific learning Algorithm 2 (GD) is invoked with robust loss $\tilde{\ell}$, we have that with probability at least $1 - \delta$,

$$L_{\text{rob}}(\mathcal{A}(\mathbf{S}), \mu) \lesssim L_{\text{rob}}(\mathcal{A}(\mathbf{S}), \mathbf{S}) + \left( G^2 \sqrt{\frac{\eta}{\lambda}} + \frac{G^2}{\lambda m} + \frac{G^2}{\lambda n} \right) \log(mn) \log(1/\delta) + \frac{M \sqrt{\log(1/\delta)}}{\sqrt{mn}}.$$

Note that prior work on robust adversarial meta-learning [Yin et al., 2018, Goldblum et al., 2020, Wang et al., 2021] focuses on empirical study of the problem; we present first theoretical guarantees.

## 6   Conclusion

In this paper, we introduce a novel notion of stability for meta-learning algorithms, namely uniform meta-stability, and offer a tighter bound on the generalization gap for the meta-learning problem compared to existing literature. We instantiate uniformly meta-stable learning algorithms and give generalization guarantees for both convex, smooth losses as well as weakly convex and non-smooth losses. Several avenues for further exciting research remain. For instance, it remains to be seen if our bounds are tight. Can we show lower bounds on the generalization error for meta-learning? Additionally, understanding how meta-learning relates to federated learning may offer insights on how to extend the theory to broader applications and inform the design of new algorithms. Finally, motivated by data privacy considerations, it would be interesting to extend our setup to privacy-preserving meta-learning, similar in spirit to the recent work of Zhou and Bassily [2022].

## Acknowledgments and Disclosure of Funding

This research was supported, in part, by the DARPA GARD award HR00112020004, NSF CAREER award IIS-1943251, funding from the Institute for Assured Autonomy (IAA) at JHU, and the Spring'22 workshop on "Learning and Games" at the Simons Institute for the Theory of Computing. YW acknowledges the support of Amazon Fellowship.

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

# Supplementary Material

## A  Experiments

In this section, we conduct a simple experiment to empirically verify our generalization bounds.

**Setting.** Following the experimental setting in Nichol et al. [2018] and Zhou et al. [2019], we consider a synthetic one-dimensional sine wave regression problem. The goal is to approximate the distribution of parameters of function $f(x; \alpha, \beta) = \alpha \sin(x + \beta)$. The task environment $\mu$ is a joint distribution $\mathcal{D}(\alpha, \beta)$ of the parameters $\alpha$ and $\beta$. We take $\mathcal{D}(\alpha, \beta)$ to be a product distribution of $\mathcal{D}(\alpha) = \mathrm{U}([-5, 5]), \mathcal{D}(\beta) = \mathrm{U}([0, \pi])$. We generate the meta-sample by first sampling $m$ training tasks, i.e., $m$ pairs of $(\alpha, \beta)$ sampled independently from $\mathcal{D}(\alpha, \beta)$. For each of these $m$ tasks, we sample $n = 10$ points, $x_1, \ldots, x_{10}$ uniformly on $[-5, 5]$ and label them as $y_i = f(x_i; \alpha, \beta)$. Similarly, at test time we generate a new task from the task distribution and generate a training sample of size $n$ (by sampling $x$'s uniformly on the interval $[-5, 5]$ and labeling them using $f(x; \alpha, \beta)$. We sample 1000 new tasks at test time. For each of the test task, we also generate an evaluation set of size 200, and use it to estimate the mean-squared error between the predictions of the learned model and the true labels. Our hypothesis class is a two layer network of width 40 and $tanh(\cdot)$ activation function. We run Algorithm 1 for $T = 100$ iterations with a step size of $\gamma = 0.1$ and regularization parameter $\lambda = 0.5$. Algorithm 2 (GD) is run for $K = 15$ iterations with step size $\eta = 0.02$. The experiment is conducted on a T4 GPU.

**Results.** We report the transfer risk, the average empirical risk (over tasks), and the generalization gap for different values of $m$ and $n$ in Figure 1. In the plot on the left, we fix $n = 10$, and vary the number of tasks $m$ from 10 to 5000. In the plot in the middle, we fix $m = 1000$, and change the number of samples $n$ from 5 to 1000. In the plot on the right, we choose $m = n$, and scale $m$ and $n$ simultaneously from 10 to 1000. We observe that in all of these three scenarios, as $m$ (and/or $n$) increase, both the generalization gap as well as the transfer risk decrease. Moreover, the generalization gap decreases at rates approximately $\mathcal{O}(1/m + 1/n + 1/\sqrt{mn})$ for these three scenarios as suggested by our theoretical result.

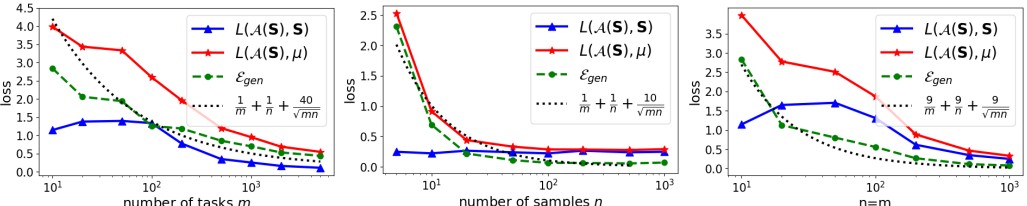

Figure 1: Error plots as a function of: (**left**) the number of tasks $m$ for fixed $n = 10$; (**middle**) the number of training samples $n$ on a test task for fixed $m = 1\mathrm{K}$; (**right**) both $m$ and $n$.

## B  Missing Proofs of Section 3

The following Lemmas and Theorems are used for proving Theorem 3.1.

**Lemma B.1** (Bounded differences/McDiarmid's inequality). Consider a function $f$ of independent random variables $z_1, \ldots, z_n$ that take their value in $\mathcal{Z}$. Suppose that $f$ satisfies the bounded differences property, namely, for any $i = 1, \ldots, n$ and any $z_1, \ldots, z_n, z_i' \in \mathcal{Z}$, it holds that

$$f(z_1, \ldots, z_{\mathrm{n}}) - f(z_1, \ldots, z_{i-1}, z_i', z_{i+1}, \ldots, z_n) \le \beta.$$

Then we have for any $p \ge 2$,

$$\|f(z_1, \ldots, z_{\mathrm{n}}) - \mathbb{E}f(z_1, \ldots, z_{\mathrm{n}})\|_p \le 2\sqrt{np}\beta.$$

**Theorem B.2** (Marcinkiewicz-Zygmund's inequality Ren and Liang [2001]). Let $x_1, \ldots, x_n$ be independent centered random variables with a finite $p$-th moment for $p \ge 2$. Then

$$\left\| \sum_{i=1}^{n} x_i \right\|_p \le 3\sqrt{2np} \left( \frac{1}{n} \sum_{i=1}^{n} \|x_i\|_p^p \right)^{\frac{1}{p}}$$

**Theorem B.3.** Let $Z = (z_1, \ldots, z_n)$ be a vector of independent random variables each taking values in $\mathcal{Z}$. Let $\mathbf{Z} = (Z_1, \ldots, Z_m)$ be a vector of independent random vectors each taking values in $\mathcal{Z}^n$. Let $g_{j,i} : (\mathcal{Z}^n)^m \times \mathcal{Z}^n \to \mathbb{R}$ be some functions such that the following holds for any $i \in [n], j \in [m]$:

(1). $|\mathbb{E}[g_{j,i}(\mathbf{Z}, Z)|Z_j, z_i]| \leq M$ a.s.,

(2). $\mathbb{E}\left[g_{j,i}(\mathbf{Z}, Z)|Z_{[m]\setminus\{j\}}, z_{[n]\setminus\{i\}}\right] = 0$ a.s.,

(3). $g_{j,i}$ has a bounded difference $\bar{\beta}$ w.r.t. all variables except the $(j, i)$-th variable.

Then we have

$$\left\|\sum_{j=1}^{m}\sum_{i=1}^{n} g_{j,i}(\mathbf{Z}, Z)\right\| \lesssim mn\bar{\beta}\log(mn) + M\sqrt{mn}.$$

*Proof of Theorem B.3.* The proof is an extension of [Bousquet et al., 2020, Theorem 4].

Without loss of generality, we suppose that $n = 2^k, m = 2^r$. Otherwise, we can add extra functions that equal to zero. Consider a sequence of partitions $\mathcal{C}_0, \ldots, \mathcal{C}_k$ with $\mathcal{C}_0 = \{\{i\} : i \in [n]\}$, $\mathcal{C}_k = \{[n]\}$, and to get $\mathcal{C}_l$ from $\mathcal{C}_{l+1}$ we split each subset into $\mathcal{C}_{l+1}$ into two equal parts. We have

$$\mathcal{C}_0 = \left\{\{1\}, \ldots, \{2^k\}\right\}, \mathcal{C}_1 = \left\{\{1, 2\}, \{3, 4\}, \ldots, \{2^k - 1, 2^k\}\right\}, \mathcal{C}_k = \left\{\{1, \ldots, 2^k\}\right\}.$$

By construction, we have $|\mathcal{C}_l| = 2^{k-l}$ and $|C| = 2^l$ for each $C \in \mathcal{C}_l$. For each $i \in [n]$ and $l = 0, \ldots, k$, denote by $C^l(i) \in \mathcal{C}_l$ the only set from $\mathcal{C}_l$ that contains $i$. In particular, $C^0(i) = \{i\}$ and $C^k(i) = [n]$.

Similarly, we consider a sequence of partitions $\mathcal{E}_0, \ldots, \mathcal{E}_r$ with $\mathcal{E}_0 = \{\{j\} : j \in [m]\}$, $\mathcal{E}_r = \{[m]\}$, and to get $\mathcal{E}_q$ from $\mathcal{E}_{q+1}$ we split each subset in $\mathcal{E}_{q+1}$ into two equal parts. We have

$$\mathcal{E}_0 = \{\{1\}, \ldots, \{2^r\}\}, \mathcal{E}_1 = \{\{1, 2\}, \{3, 4\}, \ldots, \{2^r - 1, 2^r\}\}, \mathcal{E}_r = \{\{1, \ldots, 2^r\}\}.$$

By construction, we have $|\mathcal{E}_q| = 2^{r-q}$ and $|E| = 2^q$ for each $E \in \mathcal{E}_q$. For each $j \in [m]$ and $q = 0, \ldots, r$, denote by $E^q(j) \in \mathcal{E}_q$ the only set from $\mathcal{E}_q$ that contains $j$. In particular, $E^0(j) = \{j\}$ and $E^r(j) = [m]$.

For each $i \in [n], j \in [m]$ and every $l = 0, \ldots, k, q = 0, \ldots, r$, consider the random variables

$$g_{j,i}^{q,l} = g_{j,i}^{q,l}(Z_j, Z_{[m]\setminus E^q(j)}, z_i, z_{[n]\setminus C^l(i)}),$$

i.e., conditioned on $Z_j, z_i$ and all the vectors that are not in the same set as $Z_j$ in the partition $\mathcal{E}_q$ and all the variables that are not in the same set as $z_i$ in the partition $\mathcal{C}_l$. In particular, $g_{j,i}^{0,0} = g_{j,i}$, $g_{j,i}^{r,k} = \mathbb{E}[g_{j,i}|Z_j, z_i]$. We can write a telescopic sum as follows:

$$g_{j,i} - \mathbb{E}[g_{j,i}|Z_j, z_i] = \sum_{q=0}^{r-1} g_{j,i}^{q,0} - g_{j,i}^{q+1,0} + \sum_{l=0}^{k-1} g_{j,i}^{r,l} - g_{j,i}^{r,l+1},$$

and the total sum of interest satisfies by the triangle inequality

$$\left\|\sum_{j=1}^{m}\sum_{i=1}^{n} g_{j,i}\right\| \leq \left\|\sum_{j=1}^{m}\sum_{i=1}^{n}\mathbb{E}[g_{j,i}|Z_j, z_i]\right\| + \sum_{q=0}^{r-1}\left\|\sum_{j=1}^{m}\sum_{i=1}^{n} g_{j,i}^{q,0} - g_{j,i}^{q+1,0}\right\| + \sum_{l=0}^{k-1}\left\|\sum_{j=1}^{m}\sum_{i=1}^{n} g_{j,i}^{r,l} - g_{j,i}^{r,l+1}\right\|.$$

Since $|\mathbb{E}[g_{j,i}|Z_j, z_i]| \leq M$ and $\mathbb{E}(\mathbb{E}[g_{j,i}|Z_j, z_i]) = 0$, by applying McDiarmid inequality in Lemma B.1, we have

$$\left\|\sum_{j=1}^{m}\sum_{i=1}^{n}\mathbb{E}[g_{j,i}|Z_j, z_i]\right\| \leq 4\sqrt{2mn}M. \tag{2}$$

We observe that

$$g_{j,i}^{q+1,l+1}\left(\mathbf{Z}_j, \mathbf{Z}_{[m]\setminus E^{q+1}(j)}, \mathbf{z}_i, \mathbf{z}_{[n]\setminus C^{l+1}(i)}\right)$$

$$= \mathbb{E}\left[g_{j,i}^{q+1,l}\left(\mathbf{Z}_j, \mathbf{Z}_{[m]\setminus E^{q+1}(j)}, \mathbf{z}_i, \mathbf{z}_{[n]\setminus C^l(i)}\right)|\mathbf{z}_i, \mathbf{z}_{[n]\setminus C^{l+1}(i)}\right]$$

$$\text{(The expectation is take w.r.t. the variable } z_s, s \in C^{l+1}(i)\setminus C^l(i))$$

$$= \mathbb{E}\left[g_{j,i}^{q,l+1}\left(\mathbf{Z}_j, \mathbf{Z}_{[m]\setminus E^q(j)}, \mathbf{z}_i, \mathbf{z}_{[n]\setminus C^{l+1}(i)}\right)|\mathbf{Z}_j, \mathbf{Z}_{[m]\setminus E^{q+1}(j)}\right]$$

$$\text{(The expectation is take w.r.t. the variable } Z_s, s \in E^{q+1}(j)\setminus E^q(j))$$

As function $g_{j,i}^{q,l}$ preserves the bounded differences property, if we apply McDiarmid's inequality conditioned on $\mathbf{Z}_j, \mathbf{Z}_{[m]\setminus E^{q+1}(j)}, \mathbf{z}_i, \mathbf{z}_{[n]\setminus C^{l+1}(i)}$, we obtain a uniform bound

$$\left\|g_{j,i}^{q,0} - g_{j,i}^{q+1,0}\right\|\left(\mathbf{Z}_j, \mathbf{Z}_{[m]\setminus E^{q+1}(j)}, \mathbf{z}_i, \mathbf{z}_{[n]\setminus C^0(i)}\right) \leq 2\sqrt{2^{q+1}}\bar{\beta}$$

$$\left\|g_{j,i}^{r,l} - g_{j,i}^{r,l+1}\right\|\left(\mathbf{Z}_j, \mathbf{Z}_{[m]\setminus E^r(j)}, \mathbf{z}_i, \mathbf{z}_{[n]\setminus C^{l+1}(i)}\right) \leq 2\sqrt{2^{l+1}}\bar{\beta}$$

as there are $2^l$ indices in $C^{l+1}(i)\setminus C^l(i)$ and $2^q$ indices in $E^{q+1}(j)\setminus E^q(j)$.

Now we focus on $\sum_{j\in E^q}\sum_{i\in C^0} g_{j,i}^{q,0} - g_{j,i}^{q+1,0}$ for $E^q \in \mathcal{E}_q$ and $\sum_{j\in E^r}\sum_{i\in C^l} g_{j,i}^{r,l} - g_{j,i}^{r,l+1}$ for $C^l \in \mathcal{C}_l$, respectively. Since $g_{j,i}^{q,0} - g_{j,i}^{q+1,0}$ for $j \in E^q, i \in C^0$ depends on $\mathbf{Z}_j, \mathbf{Z}_{[m]\setminus E^q(j)}, \mathbf{z}_i, \mathbf{z}_{[n]\setminus C^0(i)}$, the terms are independent and zero mean conditioned on $\mathbf{Z}_{[m]\setminus E^q(j)}$. Applying Theorem B.2, we have

$$\left\|\sum_{j\in E^q}\sum_{i\in C^0} g_{j,i}^{q,0} - g_{j,i}^{q+1,0}\right\|^2 \left(\mathbf{Z}_{[m]\setminus E^q}\right)$$

$$\leq 36 \cdot 2^q \frac{1}{2^q}\sum_{j\in E^q}\sum_{i\in C^0}\left\|g_{j,i}^{q,0} - g_{j,i}^{q+1,0}\right\|^2\left(\mathbf{Z}_{[m]\setminus E^q}\right)$$

Integrating with respect to $(\mathbf{Z}_{[m]\setminus E^q})$ and using $\left\|g_{j,i}^{q,0} - g_{j,i}^{q+1,0}\right\| \leq 2\sqrt{2^{q+1}}\bar{\beta}$, we have

$$\left\|\sum_{j\in E^q}\sum_{i\in C^0} g_{j,i}^{q,0} - g_{j,i}^{q+1,0}\right\| \leq 6\sqrt{2^q} \times 2\sqrt{2^{q+1}}\bar{\beta} = 12\sqrt{2} \cdot 2^q\bar{\beta}.$$

Applying triangle inequality over all sets $C^0 \in \mathcal{C}_0, E^q \in \mathcal{E}_q$ gives us that

$$\left\|\sum_{j\in[m]}\sum_{i\in[n]} g_{j,i}^{q,0} - g_{j,i}^{q+1,0}\right\| \leq \sum_{E^q\in\mathcal{E}_q, C^0\in\mathcal{C}_0}\left\|\sum_{j\in E^q, i\in C^0} g_{j,i}^{q,0} - g_{j,i}^{q+1,0}\right\|$$

$$\leq 2^{r+k-q} \times 12\sqrt{2} \cdot 2^q\bar{\beta}$$

$$= 12\sqrt{2} \cdot 2^{r+k}\bar{\beta}.$$

Similarly, $g_{j,i}^{r,l} - g_{j,i}^{r,l+1}$ for $j \in E^r, i \in C^l$ depends on $\mathbf{z}_i, \mathbf{z}_{[n]\setminus C^{l+1}(i)}$, the terms are independent and zero mean conditioned on $\mathbf{z}_{[n]\setminus C^{l+1}(i)}$. Applying Theorem B.2, we have

$$\left\|\sum_{j\in E^r}\sum_{i\in C^l} g_{j,i}^{r,l} - g_{j,i}^{r,l+1}\right\|^2\left(\mathbf{z}_{[n]\setminus C^l}\right)$$

$$\leq 36 \cdot 2^{l+r}\frac{1}{2^{l+r}}\sum_{j\in E^r}\sum_{i\in C^l}\left\|g_{j,i}^{r,l} - g_{j,i}^{r,l+1}\right\|^2\left(\mathbf{z}_{[n]\setminus C^l}\right)$$

Integrating with respect to $(\mathbf{z}_{[n]\setminus C^l})$ and using $\left\|g_{j,i}^{r,l} - g_{j,i}^{r,l+1}\right\| \leq 2\sqrt{2^{l+1}}\bar{\beta}$, we have

$$\left\|\sum_{j\in E^r}\sum_{i\in C^l} g_{j,i}^{r,l} - g_{j,i}^{r,l+1}\right\| \leq 6\sqrt{2^{l+r}} \times 2\sqrt{2^{l+1}}\bar{\beta} = 12\sqrt{2} \cdot 2^{l+0.5r}\bar{\beta}.$$

Applying triangle inequality over all sets $C^l \in \mathcal{C}_l, E^r \in \mathcal{E}_r$ gives us that

$$\left\| \sum_{j \in [m]} \sum_{i \in [n]} g_{j,i}^{r,l} - g_{j,i}^{r,l+1} \right\| \leq \sum_{E^r \in \mathcal{E}_r, C^l \in \mathcal{C}_l} \left\| \sum_{j \in E^r, i \in C^l} g_{j,i}^{r,l} - g_{j,i}^{r,l+1} \right\|$$

$$\leq 2^{k-l} \times 12\sqrt{2} \cdot 2^{l+0.5r} \bar{\beta}$$

$$\leq 12\sqrt{2} \cdot 2^{r+k} \bar{\beta}.$$

Recall that $2^k < 2n, 2^r < 2m$ due to the possible extension of the sample. Therefore we have

$$\sum_{q=0}^{r-1} \left\| \sum_{j=1}^{m} \sum_{i=1}^{n} g_{j,i}^{q,0} - g_{j,i}^{q+1,0} \right\| + \sum_{l=0}^{k-1} \left\| \sum_{j=1}^{m} \sum_{i=1}^{n} g_{j,i}^{r,l} - g_{j,i}^{r,l+1} \right\| \leq 48\sqrt{2} mn\bar{\beta}(\lceil \log(m) \rceil + \lceil \log(n) \rceil)$$

$$\lesssim mn\bar{\beta} \log(mn)$$

Combined with Equation (2) get the required bound.

$\square$

We now restate and prove Theorem 3.1.

**Theorem 3.1.** Consider a meta-learning problem for some $M$-bounded loss function $\ell$ and task distribution $\mu$. Let $\mathbf{S}$ be a meta-sample consisting of training samples on $m$ tasks each of size $n$, and let $\mathcal{S} \sim \mathcal{D}$ be a sample of size $n$ on a previously unseen task $\mathcal{D} \sim \mu$. Then, for any $\beta$-uniformly meta-stable learning algorithm $\mathcal{A}$, we have that with probability $1 - \delta$,

$$L(\mathcal{A}(\mathbf{S}), \mu) \lesssim L(\mathcal{A}(\mathbf{S}), \mathbf{S}) + \bar{\beta} \log(mn) \log(1/\delta) + M\sqrt{\log(1/\delta)/(mn)}.$$

*Proof of Theorem 3.1.* In order to make use of Theorem B.3, we consider the following functions:

$$g_{j,i} = g_{j,i}(\mathbf{Z}, \mathbf{Z}) = \mathbb{E}_{(\mathcal{S}'_j, z'_j) \sim \mathcal{D}_j^{n+1}, \mathcal{D}_j \sim \mu} \mathbb{E}_{(\mathcal{S}, z) \sim \mathcal{D}^{n+1}, \mathcal{D} \sim \mu} \ell(\mathcal{A}(\mathbf{S}^{(j)})(\mathcal{S}), z) - \ell(\mathcal{A}(\mathbf{S}^{(j)})(\mathcal{S}_j^{(i)}), z_j^i)$$

By the definition of uniform meta stability, we can write the following decomposition:

$$|mn(L(\mathcal{A}(\mathbf{S}), \mu) - L(\mathcal{A}(\mathbf{S}), \mathcal{S}))|$$

$$= \left| \sum_{j=1}^{m} \sum_{i=1}^{n} \mathbb{E}_{(\mathcal{S}, z) \sim \mathcal{D}^{n+1}, \mathcal{D} \sim \mu} \ell(\mathcal{A}(\mathbf{S})(\mathcal{S}), z) - \ell(\mathcal{A}(\mathbf{S})(\mathcal{S}_j), z_j^i) \right|$$

$$= \left| \sum_{j=1}^{m} \sum_{i=1}^{n} \mathbb{E}_{\substack{(\mathcal{S}, z) \sim \mathcal{D}^{n+1} \\ \mathcal{D} \sim \mu}} \mathbb{E}_{\substack{(\mathcal{S}'_j, z'_j) \sim \mathcal{D}_j^{n+1} \\ \mathcal{D}_j \sim \mu}} \left( \ell(\mathcal{A}(\mathbf{S})(\mathcal{S}), z) - \ell(\mathcal{A}(\mathbf{S}^{(j)})(\mathcal{S}_j^{(i)}), z_j^i) \right. \right.$$

$$\left. \left. + \ell(\mathcal{A}(\mathbf{S}^{(j)})(\mathcal{S}_j^{(i)}), z_j^i) - \ell(\mathcal{A}(\mathbf{S})(\mathcal{S}_j), z_j^i) \right) \right|$$

$$\leq \left\| \sum_{j=1}^{m} \sum_{i=1}^{n} \mathbb{E}_{\substack{(\mathcal{S}, z) \sim \mathcal{D}^{n+1} \\ \mathcal{D} \sim \mu}} \mathbb{E}_{\substack{(\mathcal{S}'_j, z'_j) \sim \mathcal{D}_j^{n+1} \\ \mathcal{D}_j \sim \mu}} \left( \ell(\mathcal{A}(\mathbf{S})(\mathcal{S}), z) - \ell(\mathcal{A}(\mathbf{S}^{(j)})(\mathcal{S}_j^{(i)}), z_j^i) \right) \right\| + mn\bar{\beta}$$

$$= \left\| \sum_{j=1}^{m} \sum_{i=1}^{n} g_{j,i} \right\| + mn\bar{\beta}$$

Moreover, we have $\mathbb{E}[g_{j,i}|\mathcal{S}_1, \ldots, \mathcal{S}_{j-1}, \mathcal{S}_{j+1}, \ldots, \mathcal{S}_m, z_1, \ldots, z_{i-1}, z_{i+1}, \ldots, z_n] = 0$ and $|g_{j,i}| \leq 2M$ a.s. for $i \in [n], j \in [m]$. Applying Theorem B.3 as well as [Bousquet and Elisseeff, 2002, Lemma 1] achieves the results.

$\square$

Theorem 3.2 can be directly proved by the definition of uniform meta-stability.

**Theorem 3.2.** Let $\mu$ be an underlying task distribution. Given a meta-sample $\mathbf{S}$, test task $\mathcal{D} \sim \mu$, and $\mathcal{S} \sim \mathcal{D}^n$, for any $\bar{\beta}$-on-average-replace-one-meta-stable meta-learning algorithm $\mathcal{A}$, we have that

$$\mathbb{E}_{\mathbf{S} \sim \{\mathcal{D}_j^n\}_{j=1}^m, \{\mathcal{D}_j\}_{j=1}^m \sim \mu^m}[L(\mathcal{A}(\mathbf{S}), \mu) - L(\mathcal{A}(\mathbf{S}), \mathbf{S})] \leq \bar{\beta}.$$

*Proof of Theorem 3.2.* Since $\mathcal{S}$ and $z'$ are both drawn i.i.d. from $\mathcal{D}$, and $\mathbf{S}$ and $\mathcal{S}'_j$ are both drawn i.i.d. from $\mu$, we have

$$\mathbb{E}_{\mathbf{S} \sim \{\mathcal{D}_j^n\}_{j=1}^m, \{\mathcal{D}_j\}_{j=1}^m \sim \mu^m} \mathbb{E}_{\mathcal{S} \sim \mathcal{D}^n, \mathcal{D} \sim \mu} L(\mathcal{A}(\mathbf{S})(\mathcal{S}), \mathcal{D})$$

$$= \mathbb{E}_{\mathbf{S} \sim \{\mathcal{D}_j^n\}_{j=1}^m, (\mathcal{S}'_j) \sim \mathcal{D}_j^n, \{\mathcal{D}_j\}_{j=1}^m \sim \mu^m, j \sim U[m]} L(\mathcal{A}(\mathbf{S}^{(j)})(\mathcal{S}_j), \mathcal{D}_j)$$

$$= \mathbb{E}_{\mathbf{S} \sim \{\mathcal{D}_j^n\}_{j=1}^m, (\mathcal{S}'_j, z'_j) \sim \mathcal{D}_j^{n+1}, \{\mathcal{D}_j\}_{j=1}^m \sim \mu^m, j \sim U[m], i \sim U[n]} \ell(\mathcal{A}(\mathbf{S}^{(j)})(\mathcal{S}_j^{(i)}), z_j^i)$$

as well as

$$\mathbb{E}_{\mathbf{S} \sim \{\mathcal{D}_j^n\}_{j=1}^m, \{\mathcal{D}_j\}_{j=1}^m \sim \mu^m} \left[ \frac{1}{m} \sum_{j=1}^m L(\mathcal{A}(\mathbf{S})(\mathcal{S}_j), \mathcal{S}_j) \right]$$

$$= \mathbb{E}_{\mathbf{S} \sim \{\mathcal{D}_j^n\}_{j=1}^m, \{\mathcal{D}_j\}_{j=1}^m \sim \mu^m, j \sim U[m]} [L(\mathcal{A}(\mathbf{S})(\mathcal{S}_j), \mathcal{S}_j)]$$

$$= \mathbb{E}_{\mathbf{S} \sim \{\mathcal{D}_j^n\}_{j=1}^m, (\mathcal{S}'_j, z'_j) \sim \mathcal{D}_j^{n+1}, \{\mathcal{D}_j\}_{j=1}^m \sim \mu^m, j \sim U[m], i \sim U[n]} \mathbb{E}_{\mathcal{S} \sim \mathcal{D}^n} \ell(\mathcal{A}(\mathbf{S})(\mathcal{S}_j), z_j^i)$$

As a result, we have

$$\mathbb{E}_{\mathbf{S} \sim \{\mathcal{D}_j^n\}_{j=1}^m, \{\mathcal{D}_j\}_{j=1}^m \sim \mu^m}[L(\mathcal{A}(\mathbf{S}), \mu) - L(\mathcal{A}(\mathbf{S}), \mathbf{S})]$$

$$= \mathbb{E}_{\mathbf{S} \sim \{\mathcal{D}_j^n\}_{j=1}^m, (\mathcal{S}'_j, z'_j) \sim \mathcal{D}_j^{n+1}, \{\mathcal{D}_j\}_{j=1}^m \sim \mu^m, j \sim U[m], i \sim U[n]} \left| \ell(\mathcal{A}(\mathbf{S}^{(j)})(\mathcal{S}_j^{(i)}), z_j^i) - \ell(\mathcal{A}(\mathbf{S})(\mathcal{S}_j), z_j^i) \right|$$

$$\leq \bar{\beta} \qquad \qquad \text{(By definition of } \bar{\beta}\text{-on-average-replace-one-meta-stable)}$$

$\square$

# C  Missing Proofs of Section 4.1

**Lemma C.1** (Shalev-Shwartz and Ben-David [2014]). Given $\mathcal{S}$ and $\mathcal{S}^{(i)}$, for a fixed w, define $u(w, \mathcal{S})$ and $u(w, \mathcal{S}^{(i)})$ is achieved via Algo. 1 with Option 1 RERM. Then if $\ell$ is convex, $G$-Lipschitz, we have $\sup_{\mathcal{S}, i \in [n]} \|u(w, \mathcal{S}) - u(w, \mathcal{S}^{(i)})\| \leq \frac{4G}{\lambda n}$. If $\ell$ is convex and $H$-smooth ($H \leq \frac{\lambda n}{2}$), we have $\|u(w, \mathcal{S}) - u(w, \mathcal{S}^{(i)})\| \leq \frac{\sqrt{8H}}{\lambda n}(\sqrt{\ell(w, z_i)} + \sqrt{\ell(w, z')})$.

**Lemma 4.1.** Assume that the loss function $\ell$ is convex and $G$-Lipschitz loss. Let $\mathbf{S}, \mathbf{S}^{(j)}$ denote neighboring meta-samples and $\mathcal{S}, \mathcal{S}^{(i)}$ the neighboring samples on a test task. Then, the following holds for Algorithm 1 with RERM for task-specific learning (i.e., Option 1 for Algorithm 2) $\forall T \geq 1$,

$$\sup_{\mathbf{S}, \mathcal{S}, j \in [m], i \in [n]} \left\| \mathcal{A}(\mathbf{S})(\mathcal{S}) - \mathcal{A}(\mathbf{S}^{(j)})(\mathcal{S}^{(i)}) \right\| \leq \frac{G}{\lambda m} + \frac{2G}{\lambda n}.$$

Further, if $\ell$ is convex, $M$-bounded and $H$-smooth, then setting $\lambda \geq H, \gamma \leq \frac{1}{\lambda}$, we have $\forall T \geq 1$,

$$\sup_{\mathbf{S}, \mathcal{S}, j \in [m], i \in [n]} \left\| \mathcal{A}(\mathbf{S})(\mathcal{S}) - \mathcal{A}(\mathbf{S}^{(j)})(\mathcal{S}^{(i)}) \right\| \leq \frac{2\sqrt{2HM}}{2\lambda n - H} + \frac{n}{2\lambda n - H} \frac{4\sqrt{2HM}}{(m+1)}.$$

*Proof of Lemma 4.1.* We slightly abuse the notation, at iteration $t$, define $w_t = \mathcal{A}(\mathbf{S}), w'_t = \mathcal{A}(\mathbf{S}^{(j)})$. Given $w_{T+1}$, define $u(w_{T+1}, \mathcal{S}) = \mathcal{A}(\mathbf{S})(\mathcal{S}), u(w'_{T+1}, \mathcal{S}^{(i)}) = \mathcal{A}(\mathbf{S}^{(j)})(\mathcal{S}^{(i)})$.

We first consider the setting where the loss $\ell$ is convex, $G$-Lipschitz. Recall that $F_{\mathcal{S}}(u, w) = L(u, \mathcal{S}) + \frac{\lambda}{2} \|u - w\|^2$. If $\ell$ is convex, then $F_{\mathcal{S}}(u, w)$ is $\lambda$-strongly-convex w.r.t u. Define $u(w, \mathcal{S}) = \arg\min_{u \in \mathcal{W}} F_{\mathcal{S}}(u, w), u(w', \mathcal{S}) = \arg\min_{u \in \mathcal{W}} F_{\mathcal{S}}(u, w')$. We have the following:

$$F_{\mathcal{S}}(u(w', \mathcal{S}), w) - F_{\mathcal{S}}(u(w, \mathcal{S}), w) \geq \lambda \|u(w, \mathcal{S}) - u(w', \mathcal{S})\|^2$$

$$F_{\mathcal{S}}(u(w, \mathcal{S}), w') - F_{\mathcal{S}}(u(w', \mathcal{S}), w') \geq \lambda \|u(w, \mathcal{S}) - u(w', \mathcal{S})\|^2$$

Sum the above gives us that

$$
\begin{aligned}
& 2\lambda \left\| u(w, \mathcal{S}) - u(w', \mathcal{S}) \right\|^2 \\
& \leq F_{\mathcal{S}}(u(w', \mathcal{S}), w) - F_{\mathcal{S}}(u(w, \mathcal{S}), w) + F_{\mathcal{S}}(u(w, \mathcal{S}), w') - F_{\mathcal{S}}(u(w', \mathcal{S}), w') \\
& = \frac{\lambda}{2} \left( \left\| u(w', \mathcal{S}) - w \right\|^2 - \left\| u(w, \mathcal{S}) - w \right\|^2 + \left\| u(w, \mathcal{S}) - w' \right\|^2 - \left\| u(w', \mathcal{S}) - w' \right\|^2 \right) \\
& = \lambda \left\langle u(w, \mathcal{S}) - u(w', \mathcal{S}), w - w' \right\rangle \\
& \leq \lambda \left\| u(w, \mathcal{S}) - u(w', \mathcal{S}) \right\| \left\| w - w' \right\|
\end{aligned}
$$

This gives us that

$$
\left\| u(w, \mathcal{S}) - u(w', \mathcal{S}) \right\| \leq \frac{1}{2} \left\| w - w' \right\|. \tag{3}
$$

Similarly, define $u(w', \mathcal{S}') = \arg\min_{u \in \mathcal{W}} F_{\mathcal{S}'}(u, w')$, we have

$$
\begin{aligned}
F_{\mathcal{S}}(u(w', \mathcal{S}'), w) - F_{\mathcal{S}}(u(w, \mathcal{S}), w) &\geq \lambda \left\| u(w, \mathcal{S}) - u(w', \mathcal{S}') \right\|^2 \\
F_{\mathcal{S}'}(u(w, \mathcal{S}), w') - F_{\mathcal{S}'}(u(w', \mathcal{S}'), w') &\geq \lambda \left\| u(w, \mathcal{S}) - u(w', \mathcal{S}') \right\|^2
\end{aligned}
$$

Sum the above gives us that

$$
\begin{aligned}
& 2\lambda \left\| u(w, \mathcal{S}) - u(w', \mathcal{S}') \right\|^2 \\
& \leq F_{\mathcal{S}}(u(w', \mathcal{S}'), w) - F_{\mathcal{S}}(u(w, \mathcal{S}), w) + F_{\mathcal{S}'}(u(w, \mathcal{S}), w') - F_{\mathcal{S}'}(u(w', \mathcal{S}'), w') \\
& = L(u(w', \mathcal{S}'), \mathcal{S}) - L(u(w, \mathcal{S}), \mathcal{S}) + L(u(w, \mathcal{S}), \mathcal{S}') - L(u(w', \mathcal{S}'), \mathcal{S}') \\
& \quad + \frac{\lambda}{2} \left( \left\| u(w', \mathcal{S}') - w \right\|^2 - \left\| u(w, \mathcal{S}) - w \right\|^2 + \left\| u(w, \mathcal{S}) - w' \right\|^2 - \left\| u(w', \mathcal{S}') - w' \right\|^2 \right) \\
& \leq 2G \left\| u(w, \mathcal{S}) - u(w', \mathcal{S}') \right\| + \lambda \left\langle u(w, \mathcal{S}) - u(w', \mathcal{S}'), w - w' \right\rangle && (\ell \text{ is } G\text{-Lipschitz}) \\
& \leq 2G \left\| u(w, \mathcal{S}) - u(w', \mathcal{S}') \right\| + \lambda \left\| u(w, \mathcal{S}) - u(w', \mathcal{S}') \right\| \left\| w - w' \right\|
\end{aligned}
$$

This gives us that

$$
\left\| u(w, \mathcal{S}) - u(w', \mathcal{S}') \right\| \leq \frac{1}{2} \left\| w - w' \right\| + \frac{G}{\lambda} \tag{4}
$$

Finally, at iteration $t$, we have

$$
\left\| w_{t+1} - w'_{t+1} \right\| \leq \left\| w_t - \gamma\lambda \left( w_t - \frac{1}{m} \sum_{j=1}^{m} u(w_t, \mathcal{S}_j) \right) - w'_t + \gamma\lambda \left( w'_t - \frac{1}{m} \sum_{j=1}^{m} u(w'_t, \mathcal{S}'_j) \right) \right\|
$$

$$
\text{(Projection is non-expansive)}
$$

$$
= \left\| (1 - \gamma\lambda)(w_t - w'_t) + \gamma\lambda \frac{1}{m} \sum_{j=1}^{m} \left( u(w_t, \mathcal{S}_j) - u(w'_t, \mathcal{S}'_j) \right) \right\|
$$

$$
\leq (1 - \gamma\lambda) \left\| w_t - w'_t \right\| + \gamma\lambda \left( \frac{m-1}{m} \frac{1}{2} \left\| w_t - w'_t \right\| + \frac{1}{m} \left( \frac{1}{2} \left\| w_t - w'_t \right\| + \frac{G}{\lambda} \right) \right)
$$

$$
\text{(Equation (3), (4))}
$$

$$
= (1 - \frac{\gamma\lambda}{2}) \left\| w_t - w'_t \right\| + \frac{\gamma G}{m}
$$

Choose $\gamma \leq \frac{1}{\lambda}, \forall t$. Rearrange gives us that

$$
\frac{\left\| w_{t+1} - w'_{t+1} \right\|}{(1 - \gamma\lambda/2)^{t+1}} \leq \frac{\left\| w_t - w'_t \right\|}{(1 - \gamma\lambda/2)^t} + \frac{\gamma G}{m} \frac{1}{(1 - \gamma\lambda/2)^{t+1}}
$$

Note that at initialization when $t = 1$ we have $\left\| w_1 - w'_1 \right\| = 0$. Telescoping from $t = 1$ to $T + 1$

$$
\frac{\left\| w_{T+1} - w'_{T+1} \right\|}{(1 - \gamma\lambda/2)^T} \leq \frac{\gamma G}{m} \sum_{t=1}^{T-1} \frac{1}{(1 - \gamma\lambda/2)^{t+1}}
$$

Calculate gives us that

$$\left\|\mathbf{w}_{T+1} - \mathbf{w}'_{T+1}\right\| \leq \frac{2G}{\lambda m}$$

Similarly, define $\mathbf{u}(\mathbf{w}_{T+1}, \mathcal{S}) = \operatorname{argmin}_{\mathbf{u} \in \mathcal{W}} F_{\mathcal{S}}(\mathbf{u}, \mathbf{w}'_{T+1})$,
$\mathbf{u}(\mathbf{w}'_{T+1}, \mathcal{S}^{(i)}) = \operatorname{argmin}_{\mathbf{u} \in \mathcal{W}} F_{\mathcal{S}^{(i)}}(\mathbf{u}, \mathbf{w}'_T)$, we have

$$F_{\mathcal{S}}(\mathbf{u}(\mathbf{w}'_{T+1}, \mathcal{S}^{(i)}), \mathbf{w}_{T+1}) - F_{\mathcal{S}}(\mathbf{u}(\mathbf{w}_{T+1}, \mathcal{S}), \mathbf{w}_{T+1}) \geq \lambda \left\|\mathbf{u}(\mathbf{w}_{T+1}, \mathcal{S}) - \mathbf{u}(\mathbf{w}'_{T+1}, \mathcal{S}^{(i)})\right\|^2$$

$$F_{\mathcal{S}^{(i)}}(\mathbf{u}(\mathbf{w}_{T+1}, \mathcal{S}), \mathbf{w}'_{T+1}) - F_{\mathcal{S}^{(i)}}(\mathbf{u}(\mathbf{w}'_{T+1}, \mathcal{S}^{(i)}), \mathbf{w}'_{T+1}) \geq \lambda \left\|\mathbf{u}(\mathbf{w}_{T+1}, \mathcal{S}) - \mathbf{u}(\mathbf{w}'_{T+1}, \mathcal{S}^{(i)})\right\|^2$$

Sum the above gives us that

$$
\begin{aligned}
2\lambda &\left\|\mathbf{u}(\mathbf{w}_{T+1}, \mathcal{S}) - \mathbf{u}(\mathbf{w}'_{T+1}, \mathcal{S}^{(i)})\right\|^2 \\
&\leq F_{\mathcal{S}}(\mathbf{u}(\mathbf{w}'_{T+1}, \mathcal{S}^{(i)}), \mathbf{w}_{T+1}) - F_{\mathcal{S}}(\mathbf{u}(\mathbf{w}_{T+1}, \mathcal{S}), \mathbf{w}_{T+1}) \\
&\qquad + F_{\mathcal{S}^{(i)}}(\mathbf{u}(\mathbf{w}_{T+1}, \mathcal{S}), \mathbf{w}'_{T+1}) - F_{\mathcal{S}^{(i)}}(\mathbf{u}(\mathbf{w}'_{T+1}, \mathcal{S}^{(i)}), \mathbf{w}'_{T+1}) \\
&= L(\mathbf{u}(\mathbf{w}'_{T+1}, \mathcal{S}^{(i)}), \mathcal{S}) - L(\mathbf{u}(\mathbf{w}_{T+1}, \mathcal{S}), \mathcal{S}) + L(\mathbf{u}(\mathbf{w}_{T+1}, \mathcal{S}), \mathcal{S}^{(i)}) - L(\mathbf{u}(\mathbf{w}'_{T+1}, \mathcal{S}^{(i)}), \mathcal{S}^{(i)}) \\
&\qquad + \frac{\lambda}{2}\bigg(\left\|\mathbf{u}(\mathbf{w}'_{T+1}, \mathcal{S}^{(i)}) - \mathbf{w}_{T+1}\right\|^2 - \left\|\mathbf{u}(\mathbf{w}_{T+1}, \mathcal{S}) - \mathbf{w}_{T+1}\right\|^2 \\
&\qquad\qquad + \left\|\mathbf{u}(\mathbf{w}_{T+1}, \mathcal{S}) - \mathbf{w}'_{T+1}\right\|^2 - \left\|\mathbf{u}(\mathbf{w}'_{T+1}, \mathcal{S}^{(i)}) - \mathbf{w}'_{T+1}\right\|^2\bigg) \\
&\leq \frac{2G}{n} \left\|\mathbf{u}(\mathbf{w}_{T+1}, \mathcal{S}) - \mathbf{u}(\mathbf{w}'_{T+1}, \mathcal{S}^{(i)})\right\| + \lambda \left\langle \mathbf{u}(\mathbf{w}_{T+1}, \mathcal{S}^{(i)}) - \mathbf{u}(\mathbf{w}'_{T+1}, \mathcal{S}^{(i)}), \mathbf{w}_{T+1} - \mathbf{w}'_{T+1}\right\rangle \\
&\hspace{9cm} (\ell \text{ is } G\text{-Lipschitz}) \\
&\leq \frac{2G}{n} \left\|\mathbf{u}(\mathbf{w}_{T+1}, \mathcal{S}) - \mathbf{u}(\mathbf{w}'_{T+1}, \mathcal{S}^{(i)})\right\| + \lambda \left\|\mathbf{u}(\mathbf{w}_{T+1}, \mathcal{S}) - \mathbf{u}(\mathbf{w}'_{T+1}, \mathcal{S}^{(i)})\right\| \left\|\mathbf{w}_{T+1} - \mathbf{w}'_{T+1}\right\|
\end{aligned}
$$

This gives us that

$$\left\|\mathbf{u}(\mathbf{w}_{T+1}, \mathcal{S}) - \mathbf{u}(\mathbf{w}'_{T+1}, \mathcal{S}^{(i)})\right\| \leq \frac{1}{2}\left\|\mathbf{w}_{T+1} - \mathbf{w}'_{T+1}\right\| + \frac{2G}{\lambda n} \leq \frac{G}{\lambda m} + \frac{2G}{\lambda n} \tag{5}$$

We now consider the surrogate loss $\ell$ is convex, non-negative and $H$-smooth. Note that such loss is also self-bounded. From a similar argument, we have

$$\left\|\mathbf{u}(\mathbf{w}, \mathcal{S}) - \mathbf{u}(\mathbf{w}', \mathcal{S})\right\| \leq \frac{1}{2}\left\|\mathbf{w} - \mathbf{w}'\right\|.$$

Moreover,

$$
\begin{aligned}
2\lambda &\left\|\mathbf{u}(\mathbf{w}, \mathcal{S}) - \mathbf{u}(\mathbf{w}', \mathcal{S}')\right\|^2 \\
&\leq F_{\mathcal{S}}(\mathbf{u}(\mathbf{w}', \mathcal{S}'), \mathbf{w}) - F_{\mathcal{S}}(\mathbf{u}(\mathbf{w}, \mathcal{S}), \mathbf{w}) + F_{\mathcal{S}'}(\mathbf{u}(\mathbf{w}, \mathcal{S}), \mathbf{w}') - F_{\mathcal{S}'}(\mathbf{u}(\mathbf{w}', \mathcal{S}'), \mathbf{w}') \\
&= L(\mathbf{u}(\mathbf{w}', \mathcal{S}'), \mathcal{S}) - L(\mathbf{u}(\mathbf{w}, \mathcal{S}), \mathcal{S}) + L(\mathbf{u}(\mathbf{w}, \mathcal{S}), \mathcal{S}') - L(\mathbf{u}(\mathbf{w}', \mathcal{S}'), \mathcal{S}') \\
&\qquad + \frac{\lambda}{2}\left(\left\|\mathbf{u}(\mathbf{w}', \mathcal{S}') - \mathbf{w}\right\|^2 - \left\|\mathbf{u}(\mathbf{w}, \mathcal{S}) - \mathbf{w}\right\|^2 + \left\|\mathbf{u}(\mathbf{w}, \mathcal{S}) - \mathbf{w}'\right\|^2 - \left\|\mathbf{u}(\mathbf{w}', \mathcal{S}') - \mathbf{w}'\right\|^2\right) \\
&\leq (\left\|\nabla L(\mathbf{u}(\mathbf{w}, \mathcal{S}), \mathcal{S})\right\| + \left\|\nabla L(\mathbf{u}(\mathbf{w}', \mathcal{S}'), \mathcal{S}')\right\|) \left\|\mathbf{u}(\mathbf{w}', \mathcal{S}') - \mathbf{u}(\mathbf{w}, \mathcal{S})\right\| + H\left\|\mathbf{u}(\mathbf{w}', \mathcal{S}') - \mathbf{u}(\mathbf{w}, \mathcal{S})\right\|^2 \\
&\hspace{10cm} (\ell \text{ is } H\text{-smooth}) \\
&\qquad + \lambda\left\langle \mathbf{u}(\mathbf{w}, \mathcal{S}) - \mathbf{u}(\mathbf{w}', \mathcal{S}'), \mathbf{w} - \mathbf{w}'\right\rangle \\
&\leq \left(\sqrt{2HL(\mathbf{u}(\mathbf{w}, \mathcal{S}), \mathcal{S})} + \sqrt{2HL(\mathbf{u}(\mathbf{w}', \mathcal{S}'), \mathcal{S}')}\right)\left\|\mathbf{u}(\mathbf{w}', \mathcal{S}') - \mathbf{u}(\mathbf{w}, \mathcal{S})\right\| \\
&\qquad + H\left\|\mathbf{u}(\mathbf{w}', \mathcal{S}') - \mathbf{u}(\mathbf{w}, \mathcal{S})\right\|^2 + \lambda\left\|\mathbf{u}(\mathbf{w}, \mathcal{S}) - \mathbf{u}(\mathbf{w}', \mathcal{S}')\right\|\left\|\mathbf{w} - \mathbf{w}'\right\| \qquad (\ell \text{ is } H\text{-smooth})
\end{aligned}
$$

which is equivalent as

$$\|u(w, \mathcal{S}) - u(w', \mathcal{S}')\| \leq \frac{\sqrt{2HL(u(w,\mathcal{S}),\mathcal{S})} + \sqrt{2HL(u(w',\mathcal{S}'),\mathcal{S}')} + \lambda\|w - w'\|}{2\lambda - H} \quad (\lambda \geq H)$$

$$\leq \frac{\sqrt{2H}}{\lambda}\left(\sqrt{L(u(w,\mathcal{S}),\mathcal{S})} + \sqrt{L(u(w',\mathcal{S}'),\mathcal{S}')}\right) + \|w - w'\| \qquad (6)$$

Finally, at iteration $t$, we have

$$\|w_{t+1} - w'_{t+1}\|$$

$$\leq \left\|w_t - \gamma\lambda\left(w_t - \frac{1}{m}\sum_{j=1}^{m}u(w_t, \mathcal{S}_j)\right) - w'_t + \gamma\lambda\left(w'_t - \frac{1}{m}\sum_{j=1}^{m}u(w'_t, \mathcal{S}'_j)\right)\right\|$$
$$\text{(Projection is non-expansive)}$$

$$= \left\|(1 - \gamma\lambda)(w_t - w'_t) + \gamma\lambda\frac{1}{m}\sum_{j=1}^{m}\left(u(w_t, \mathcal{S}_j) - u(w'_t, \mathcal{S}'_j)\right)\right\|$$

$$\leq (1 - \gamma\lambda)\|w_t - w'_t\| + \gamma\lambda\left(\frac{m-1}{m}\frac{1}{2}\|w_t - w'_t\|\right.$$

$$\left. + \frac{1}{m}\left(\frac{\sqrt{2H}}{\lambda}\left(\sqrt{L(u(w_t,\mathcal{S}_j),\mathcal{S}_j)} + \sqrt{L(u(w'_t,\mathcal{S}'_j),\mathcal{S}'_j)}\right) + \|w_t - w'_t\|\right)\right)$$
$$\text{(Equation (3), (6))}$$

$$= \left(1 - \frac{m+1}{2m}\gamma\lambda\right)\|w_t - w'_t\| + \frac{\gamma\sqrt{2H}}{m}\left(\sqrt{L(u(w_t,\mathcal{S}_j),\mathcal{S}_j)} + \sqrt{L(u(w'_t,\mathcal{S}'_j),\mathcal{S}'_j)}\right)$$

Telescope gives us that

$$\|w_{T+1} - w'_{T+1}\| \leq \frac{\gamma\sqrt{2H\lambda}}{m}\sum_{t=1}^{T}\left(1 - \frac{m+1}{2m}\gamma\lambda\right)^{T-t}\left(\sqrt{L(u(w_t,\mathcal{S}_j),\mathcal{S}_j)} + \sqrt{L(u(w'_t,\mathcal{S}'_j),\mathcal{S}'_j)}\right)$$

$$\leq \frac{4\sqrt{2HM}}{\lambda(m+1)}, \qquad (7)$$

where the last line holds if we consider $M$-bounded loss. Otherwise, we have

$$\|w_{T+1} - w'_{T+1}\| \leq \frac{2\sqrt{2H}}{\lambda(m+1)}\left(\sqrt{\max_{t\in[T]}L(u(w_t,\mathcal{S}_j),\mathcal{S}_j)} + \sqrt{\max_{t\in[T]}L(u(w'_t,\mathcal{S}'_j),\mathcal{S}'_j)}\right). \quad (8)$$

Therefore, we have

$$2\lambda\left\|u(w, \mathcal{S}) - u(w', \mathcal{S}^{(i)})\right\|^2$$

$$\leq F_{\mathcal{S}}(u(w', \mathcal{S}^{(i)}), w) - F_{\mathcal{S}}(u(w, \mathcal{S}), w) + F_{\mathcal{S}^{(i)}}(u(w, \mathcal{S}), w') - F_{\mathcal{S}^{(i)}}(u(w', \mathcal{S}^{(i)}), w')$$

$$= L(u(w', \mathcal{S}^{(i)}), \mathcal{S}) - L(u(w, \mathcal{S}), \mathcal{S}) + L(u(w, \mathcal{S}), \mathcal{S}^{(i)}) - L(u(w', \mathcal{S}^{(i)}), \mathcal{S}^{(i)})$$

$$+ \frac{\lambda}{2}\left(\left\|u(w', \mathcal{S}^{(i)}) - w\right\|^2 - \|u(w, \mathcal{S}) - w\|^2 + \|u(w, \mathcal{S}) - w'\|^2 - \left\|u(w', \mathcal{S}^{(i)}) - w'\right\|^2\right)$$

$$\leq \frac{1}{n}\left(\ell(u(w', \mathcal{S}^{(i)}), z^i) - \ell(u(w', \mathcal{S}^{(i)}), z') + \ell(u(w', \mathcal{S}), z') - \ell(u(w', \mathcal{S}), z^i)\right)$$

$$+ \lambda\left\langle u(w, \mathcal{S}) - u(w', \mathcal{S}^{(i)}), w - w'\right\rangle$$

$$\leq \frac{1}{n}\left(\sqrt{2H\ell(u(w,\mathcal{S}),z^i)} + \sqrt{2H\ell(u(w',\mathcal{S}^{(i)}),z')}\right)\left\|u(w', \mathcal{S}^{(i)}) - u(w, \mathcal{S})\right\|$$

$$+ \frac{H}{n}\left\|u(w', \mathcal{S}^{(i)}) - u(w, \mathcal{S})\right\|^2 + \lambda\|u(w, \mathcal{S}) - u(w', \mathcal{S})\|\|w - w'\| \qquad (\ell \text{ is } H\text{-smooth})$$

Rearrange gives us,

$$\left\| \mathtt{u}(\mathtt{w}, \mathcal{S}) - \mathtt{u}(\mathtt{w}', \mathcal{S}^{(i)}) \right\| \leq \frac{1}{2\lambda n - H} \left( \sqrt{2H\ell(\mathtt{u}(\mathtt{w}, \mathcal{S}), \mathtt{z}^i)} + \sqrt{2H\ell(\mathtt{u}(\mathtt{w}', \mathcal{S}^{(i)}), \mathtt{z}')} \right)$$
$$+ \frac{\lambda n}{2\lambda n - H} \left\| \mathtt{w} - \mathtt{w}' \right\| \tag{9}$$

Plug in $\mathtt{w}_{T+1}$ and $\mathtt{w}'_{T+1}$ gives us that

$$\left\| \mathtt{u}(\mathtt{w}_{T+1}, \mathcal{S}) - \mathtt{u}(\mathtt{w}'_{T+1}, \mathcal{S}^{(i)}) \right\| \leq \frac{2\sqrt{2HM}}{2\lambda n - H} + \frac{n}{2\lambda n - H} \frac{4\sqrt{2HM}}{(m+1)}$$

$\square$

If we apply Lemma 4.1 and Lemma C.1 with Theorem 2.2, we have the following theorem.

**Theorem C.2.** The following holds for Algorithm 1 with step-size $\gamma \leq \frac{1}{\lambda}$ on a given meta-sample $\mathbf{S}$, and RERM for task-specific learning (i.e., Option 1 for Algorithm 2), for all $T \geq 1$:

1. For convex, $M$-bounded, and $G$-Lipschitz loss functions, with probability at least $1 - \delta$

$$L(\mathcal{A}(\mathbf{S}), \mu) \lesssim L(\mathcal{A}(\mathbf{S}), \mathbf{S}) + \frac{G^2}{\lambda m} \log(m) \log(1/\delta) + \frac{M}{\sqrt{m}} \sqrt{\log(1/\delta)} + \frac{G^2}{\lambda n}.$$

2. For convex, $M$-bounded, and $H$-smooth loss functions ($H \leq \lambda$), with probability at least $1 - \delta$

$$L(\mathcal{A}(\mathbf{S}), \mu) \lesssim L(\mathcal{A}(\mathbf{S}), \mathbf{S}) + \frac{HM}{(m+1)\lambda} \log(m) \log(1/\delta) + \frac{M}{\sqrt{m}} \sqrt{\log(1/\delta)} + \frac{HM}{\lambda n}.$$

*Proof of Theorem C.2.* We slightly abuse the notation, at iteration $t$, define $\mathtt{w}_t = \mathcal{A}(\mathbf{S})$, $\mathtt{w}'_t = \mathcal{A}(\mathbf{S}^{(j)})$, $\mathtt{u}(\mathtt{w}_{T+1}, \mathcal{S}) = \mathcal{A}(\mathbf{S})(\mathcal{S})$, $\mathtt{u}(\mathtt{w}'_{T+1}, \mathcal{S}^{(i)}) = \mathcal{A}(\mathbf{S}^{(j)})(\mathcal{S}^{(i)})$. Apply Lemma 4.1 gives us that

$$\left| L(\mathcal{A}(\mathbf{S})(\mathcal{S}), \mathcal{S}) - L(\mathcal{A}(\mathbf{S}^{(j)})(\mathcal{S}), \mathcal{S}) \right| = \left| L(\mathtt{u}(\mathtt{w}_{T+1}, \mathcal{S}), \mathcal{S}) - L(\mathtt{u}(\mathtt{w}'_{T+1}, \mathcal{S}), \mathcal{S}) \right|$$
$$\leq G \left\| \mathtt{u}(\mathtt{w}_{T+1}, \mathcal{S}) - \mathtt{u}(\mathtt{w}'_{T+1}, \mathcal{S}) \right\| \qquad (G\text{-Lipschitz})$$
$$\leq \frac{G}{2} \left\| \mathtt{w}_{T+1} - \mathtt{w}'_{T+1} \right\| \qquad (\text{Equation } (3))$$
$$\leq \frac{G^2}{\lambda m}$$

Apply Lemma C.1 gives us that

$$\left| \ell(\mathcal{A}(\mathbf{S})(\mathcal{S}), \mathtt{z}) - \ell(\mathcal{A}(\mathbf{S})(\mathcal{S}^{(i)}), \mathtt{z}) \right| = \left| \ell(\mathtt{u}(\mathtt{w}_{T+1}, \mathcal{S}), \mathtt{z}) - \ell(\mathtt{u}(\mathtt{w}_{T+1}, \mathcal{S}^{(i)}), \mathtt{z}) \right|$$
$$\leq G \left\| \mathtt{u}(\mathtt{w}_{T+1}, \mathcal{S}) - \mathtt{u}(\mathtt{w}_{T+1}, \mathcal{S}^{(i)}) \right\|$$
$$\leq \frac{4G^2}{\lambda n}$$

Apply Theorem 2.2 with $\beta' = \frac{G^2}{\lambda m}$, $\beta = \frac{4G^2}{\lambda n}$ achieves the results.

Similarly, if the loss is $M$-bounded, convex, non-negative and $H$-smooth, we have

$$\left| L(\mathcal{A}(\mathbf{S})(\mathcal{S}), \mathcal{S}) - L(\mathcal{A}(\mathbf{S}^{(j)})(\mathcal{S}), \mathcal{S}) \right|$$
$$= \left| L(\mathtt{u}(\mathtt{w}_{T+1}, \mathcal{S}), \mathcal{S}) - L(\mathtt{u}(\mathtt{w}'_{T+1}, \mathcal{S}), \mathcal{S}) \right|$$
$$\leq \sqrt{2HL(\mathtt{u}(\mathtt{w}_{T+1}, \mathcal{S}), \mathcal{S})} \left\| \mathtt{u}(\mathtt{w}_{T+1}, \mathcal{S}) - \mathtt{u}(\mathtt{w}'_{T+1}, \mathcal{S}) \right\| + \frac{H}{2} \left\| \mathtt{u}(\mathtt{w}_{T+1}, \mathcal{S}) - \mathtt{u}(\mathtt{w}'_{T+1}, \mathcal{S}) \right\|^2$$
$$(\ell \text{ is } H\text{-smooth})$$
$$\leq \sqrt{2HM} \frac{1}{2} \left\| \mathtt{w}_{T+1} - \mathtt{w}'_{T+1} \right\| + \frac{H}{8} \left\| \mathtt{w}_{T+1} - \mathtt{w}'_{T+1} \right\|^2 \qquad (\text{Equation } (7))$$
$$\leq \frac{4HM}{(m+1)\lambda} + \frac{4H^2M}{(m+1)^2\lambda^2}$$
$$\leq \frac{8HM}{(m+1)\lambda} \qquad (\lambda \geq H)$$

Apply Lemma C.1 gives us that

$$\left|\ell(\mathcal{A}(\mathbf{S})(\mathcal{S}), \mathbf{z}) - \ell(\mathcal{A}(\mathbf{S})(\mathcal{S}^{(i)}), \mathbf{z})\right|$$

$$= \left|\ell(\mathbf{u}(\mathbf{w}_{T+1}, \mathcal{S}), \mathbf{z}) - \ell(\mathbf{u}(\mathbf{w}_{T+1}, \mathcal{S}^{(i)}), \mathbf{z})\right|$$

$$\leq \sqrt{2H\ell(\mathbf{u}(\mathbf{w}_{T+1}, \mathcal{S}), \mathbf{z})} \left\|\mathbf{u}(\mathbf{w}_{T+1}, \mathcal{S}) - \mathbf{u}(\mathbf{w}_{T+1}, \mathcal{S}^{(i)})\right\| + \frac{H}{2} \left\|\mathbf{u}(\mathbf{w}_{T+1}, \mathcal{S}) - \mathbf{u}(\mathbf{w}_{T+1}, \mathcal{S}^{(i)})\right\|^2$$

$$\leq \frac{8HM}{\lambda n} + \frac{16H^2 M}{\lambda^2 n^2}$$

$$\leq \frac{24HM}{\lambda n} \qquad\qquad (\lambda \geq H)$$

Apply Theorem 2.2 with $\beta' = \frac{8HM}{(m+1)\lambda}, \beta = \frac{24HM}{\lambda n}$ achieves the results.

$\square$

**Theorem 4.2.** The following holds for Algorithm 1 with step-size $\gamma \leq \frac{1}{\lambda}$ on a given meta-sample $\mathbf{S}$, and RERM for task-specific learning (i.e., Option 1 for Algorithm 2), for all $T \geq 1$:

1. For convex, $M$-bounded, and $G$-Lipschitz loss functions, with probability at least $1 - \delta$

$$L(\mathcal{A}(\mathbf{S}), \mu) \lesssim L(\mathcal{A}(\mathbf{S}), \mathbf{S}) + \left(\frac{G^2}{\lambda n} + \frac{G^2}{\lambda m}\right) \log(mn) \log(1/\delta) + \frac{M\sqrt{\log(1/\delta)}}{\sqrt{mn}}.$$

2. For convex, $M$-bounded, and $H$-smooth loss functions ($H \leq \lambda$), with probability at least $1 - \delta$

$$L(\mathcal{A}(\mathbf{S}), \mu) \lesssim L(\mathcal{A}(\mathbf{S}), \mathbf{S}) + \left(\frac{HM}{(2n-1)\lambda} + \frac{HM}{(m+1)\lambda}\right) \log(mn) \log(1/\delta) + \frac{M\sqrt{\log(1/\delta)}}{\sqrt{mn}}.$$

*Proof of Theorem 4.2.* We slightly abuse the notation, at iteration $t$, define $\mathbf{w}_t = \mathcal{A}(\mathbf{S})$, $\mathbf{w}'_t = \mathcal{A}(\mathbf{S}^{(j)})$, $\mathbf{u}(\mathbf{w}_{T+1}, \mathcal{S}) = \mathcal{A}(\mathbf{S})(\mathcal{S}), \mathbf{u}(\mathbf{w}'_{T+1}, \mathcal{S}^{(i)}) = \mathcal{A}(\mathbf{S}^{(j)})(\mathcal{S}^{(i)})$. For $\ell$ to be convex and $G$-Lipschitz, applying Lemma 4.1 gives us that

$$\left|\ell(\mathcal{A}(\mathbf{S})(\mathcal{S}), \mathbf{z}) - \ell(\mathcal{A}(\mathbf{S}^{(j)})(\mathcal{S}^{(i)}), \mathbf{z})\right| = \left|\ell(\mathbf{u}(\mathbf{w}_{T+1}, \mathcal{S}), \mathbf{z}) - \ell(\mathbf{u}(\mathbf{w}'_{T+1}, \mathcal{S}^{(i)}), \mathbf{z})\right|$$

$$\leq G \left\|\mathbf{u}(\mathbf{w}_{T+1}, \mathcal{S}) - \mathbf{u}(\mathbf{w}'_{T+1}, \mathcal{S}^{(i)})\right\|$$

$$\leq \frac{G^2}{\lambda m} + \frac{2G^2}{\lambda n}.$$

Further apply Theorem 3.1 gives us the result. For $\ell$ to be convex, $M$-bounded and $H$-smooth, we have

$$\left|\ell(\mathcal{A}(\mathbf{S})(\mathcal{S}), \mathbf{z}) - \ell(\mathcal{A}(\mathbf{S}^{(j)})(\mathcal{S}^{(i)}), \mathbf{z})\right|$$

$$= \left|\ell(\mathbf{u}(\mathbf{w}_{T+1}, \mathcal{S}), \mathbf{z}) - \ell(\mathbf{u}(\mathbf{w}'_{T+1}, \mathcal{S}^{(i)}), \mathbf{z})\right|$$

$$\leq \sqrt{2H\ell(\mathbf{u}(\mathbf{w}_{T+1}, \mathcal{S}), \mathbf{z})} \left\|\mathbf{u}(\mathbf{w}_{T+1}, \mathcal{S}) - \mathbf{u}(\mathbf{w}'_{T+1}, \mathcal{S}^{(i)})\right\| + \frac{H}{2} \left\|\mathbf{u}(\mathbf{w}_{T+1}, \mathcal{S}) - \mathbf{u}(\mathbf{w}'_{T+1}, \mathcal{S}^{(i)})\right\|^2$$

$$\leq \sqrt{2HM}\left(\frac{2\sqrt{2HM}}{2\lambda n - H} + \frac{n}{2\lambda n - H}\frac{4\sqrt{2HM}}{(m+1)}\right) + \frac{H}{2}\left(\frac{2\sqrt{2HM}}{2\lambda n - H} + \frac{n}{2\lambda n - H}\frac{4\sqrt{2HM}}{(m+1)}\right)^2$$

$$\leq \frac{4HM}{(2n-1)\lambda} + \frac{8HM}{(m+1)\lambda} + \frac{8H^2 M}{(2n-1)^2\lambda^2} + \frac{16H^2 M}{(m+1)^2\lambda^2}$$

$$\leq \frac{12HM}{(2n-1)\lambda} + \frac{24HM}{(m+1)\lambda} \qquad\qquad (H \leq \lambda)$$

Apply Theorem 3.1 gives the results.

$\square$

**Lemma 4.3.** Assume that the loss function is convex, $G$-Lipschitz and $H$-smooth. Let $\mathbf{S}$, $\mathbf{S}^{(j)}$ denote neighboring meta-samples and $\mathcal{S}$, $\mathcal{S}^{(i)}$ the neighboring samples on a test task. Then the following holds for Algorithm 1 with GD for task-specific learning (i.e., Option 2 for Algorithm 2) with $\eta \leq \frac{2}{H+2\lambda}$, for all $T \geq 1$ as long as we set $\gamma \leq \frac{1}{\lambda T}$,

$$\sup_{\mathbf{S},\mathcal{S},j\in[m],i\in[n]} \left\| \mathcal{A}(\mathbf{S})(\mathcal{S}) - \mathcal{A}(\mathbf{S}^{(j)})(\mathcal{S}^{(i)}) \right\| \leq \frac{4eG}{\lambda m} + \frac{2G}{\lambda n}.$$

*Proof of Lemma 4.3.* We slightly abuse the notation, at iteration $t$, define $\mathbf{w}_t = \mathcal{A}(\mathbf{S})$, $\mathbf{w}'_t = \mathcal{A}(\mathbf{S}^{(j)})$, $\mathbf{u}(\mathbf{w}_{T+1}, \mathcal{S}) = \mathcal{A}(\mathbf{S})(\mathcal{S})$, $\mathbf{u}(\mathbf{w}'_{T+1}, \mathcal{S}^{(i)}) = \mathcal{A}(\mathbf{S}^{(j)})(\mathcal{S}^{(i)})$. Recall that $F_{\mathcal{S}}(\mathbf{u}, \mathbf{w}) = L(\mathbf{u}, \mathcal{S}) + \frac{\lambda}{2}\|\mathbf{u}-\mathbf{w}\|^2$. If $\ell$ is convex, then $F_{\mathcal{S}}(\mathbf{u}, \mathbf{w})$ is $\lambda$-strongly-convex w.r.t $\mathbf{u}$. If $\ell$ is $H$-smooth, then

$$\langle \nabla L(\mathbf{u}, \mathcal{S}) - \nabla L(\mathbf{v}, \mathcal{S}), \mathbf{u} - \mathbf{v} \rangle \geq \frac{1}{H} \|\nabla L(\mathbf{u}, \mathcal{S}) - \nabla L(\mathbf{v}, \mathcal{S})\|^2$$

Given $\mathcal{S}$, $\mathcal{S}'$, for any $\mathbf{w}$, $\mathbf{w}'$, we have

$$\left\| \mathbf{u}^{(k+1)}(\mathbf{w}, \mathcal{S}) - \mathbf{u}^{(k+1)}(\mathbf{w}', \mathcal{S}') \right\|$$

$$\leq \left\| \mathbf{u}^{(k)}(\mathbf{w}, \mathcal{S}) - \mathbf{u}^{(k)}(\mathbf{w}', \mathcal{S}') - \eta \Big( \nabla L(\mathbf{u}^{(k)}(\mathbf{w}, \mathcal{S}), \mathcal{S}) + \lambda \left( \mathbf{u}^{(k)}(\mathbf{w}, \mathcal{S}) - \mathbf{w} \right) \right.$$

$$\text{(Projection is non-expansive)}$$

$$\left. - \nabla L(\mathbf{u}^{(k)}(\mathbf{w}', \mathcal{S}'), \mathcal{S}') - \lambda \left( \mathbf{u}^{(k)}(\mathbf{w}', \mathcal{S}') - \mathbf{w}' \right) \Big) \right\|$$

$$\leq (1 - \eta\lambda) \left\| \mathbf{u}^{(k)}(\mathbf{w}, \mathcal{S}) - \mathbf{u}^{(k)}(\mathbf{w}', \mathcal{S}') \right\| + \eta\lambda \|\mathbf{w} - \mathbf{w}'\| + 2\eta G$$

Given $\mathcal{S}$, for any $\mathbf{w}$, $\mathbf{w}'$, we have

$$\left\| \mathbf{u}^{(k+1)}(\mathbf{w}, \mathcal{S}) - \mathbf{u}^{(k+1)}(\mathbf{w}', \mathcal{S}) \right\|$$

$$\leq \left\| \mathbf{u}^{(k)}(\mathbf{w}, \mathcal{S}) - \mathbf{u}^{(k)}(\mathbf{w}', \mathcal{S}) \right. \qquad \text{(Projection is non-expansive)}$$

$$\left. - \eta \left( \nabla L(\mathbf{u}^{(k)}(\mathbf{w}, \mathcal{S}), \mathcal{S}) + \lambda \left( \mathbf{u}^{(k)}(\mathbf{w}, \mathcal{S}) - \mathbf{w} \right) - \nabla L(\mathbf{u}^{(k)}(\mathbf{w}', \mathcal{S}), \mathcal{S}) - \lambda \left( \mathbf{u}^{(k)}(\mathbf{w}', \mathcal{S}) - \mathbf{w}' \right) \right) \right\|$$

$$= \left\| \mathbf{u}^{(k)}(\mathbf{w}, \mathcal{S}) - \mathbf{u}^{(k)}(\mathbf{w}', \mathcal{S}) + \lambda\eta(\mathbf{w} - \mathbf{w}') \right.$$

$$\left. - \eta \left( \nabla L(\mathbf{u}^{(k)}(\mathbf{w}, \mathcal{S}), \mathcal{S}) - \nabla L(\mathbf{u}^{(k)}(\mathbf{w}', \mathcal{S}), \mathcal{S}) + \lambda \left( \mathbf{u}^{(k)}(\mathbf{w}, \mathcal{S}) - \mathbf{u}^{(k)}(\mathbf{w}', \mathcal{S}) \right) \right) \right\|$$

$$\leq \lambda\eta \|\mathbf{w} - \mathbf{w}'\| + \left( \left\| \mathbf{u}^{(k)}(\mathbf{w}, \mathcal{S}) - \mathbf{u}^{(k)}(\mathbf{w}', \mathcal{S}) \right. \right.$$

$$\left. \left. - \eta \left( \nabla L(\mathbf{u}^{(k)}(\mathbf{w}, \mathcal{S}), \mathcal{S}) - \nabla L(\mathbf{u}^{(k)}(\mathbf{w}', \mathcal{S}), \mathcal{S}) + \lambda \left( \mathbf{u}^{(k)}(\mathbf{w}, \mathcal{S}) - \mathbf{u}^{(k)}(\mathbf{w}', \mathcal{S}) \right) \right) \right\|^2 \right)^{1/2}$$

$$\leq \lambda\eta \|\mathbf{w} - \mathbf{w}'\| + \left( (1 - \lambda\eta)^2 \left\| \mathbf{u}^{(k)}(\mathbf{w}, \mathcal{S}) - \mathbf{u}^{(k)}(\mathbf{w}', \mathcal{S}) \right\|^2 \right.$$

$$\left. + \left( \eta^2 - \frac{2\eta(1-\eta\lambda)}{H} \right) \left\| \nabla L(\mathbf{u}^{(k)}(\mathbf{w}, \mathcal{S}), \mathcal{S}) - \nabla L(\mathbf{u}^{(k)}(\mathbf{w}', \mathcal{S}), \mathcal{S}) \right\|^2 \right)^{1/2}$$

$$\text{($L$ is $H$-smooth, $\eta \leq \frac{2}{H+2\lambda}$)}$$

$$\leq (1-\lambda\eta) \left\| u^{(k)}(w, \mathcal{S}) - u^{(k)}(w', \mathcal{S}) \right\| + \lambda\eta \left\| w - w' \right\| \tag{10}$$

Combine the above two cases gives us that

$$\frac{1}{m} \sum_{j=1}^{m} \left\| u^{(k+1)}(w, \mathcal{S}_j) - u^{(k+1)}(w', \mathcal{S}_j') \right\|$$

$$\leq \frac{1}{m} \sum_{j\neq i}^{m} \left\| u^{(k)}(w, \mathcal{S}_j) - u^{(k)}(w, \mathcal{S}_j') \right\| + \frac{1}{m} \left\| u^{(k)}(w, \mathcal{S}_i) - u^{(k)}(w', \mathcal{S}_i') \right\|$$

$$\leq (1-\lambda\eta) \frac{1}{m} \sum_{j=1}^{m} \left\| u^{(k)}(w, \mathcal{S}_j) - u^{(k)}(w', \mathcal{S}_j') \right\| + \lambda\eta \left\| w - w' \right\| + \frac{2\eta G}{m}$$

Given $w_t, w_t'$, when $k = 1$, $u^{(1)}(w_t, \mathcal{S}_j') = w_t$, $u^{(1)}(w_t', \mathcal{S}_j) = u^{(1)}(w_t', \mathcal{S}_j') = w_t'$. Telescoping gives us that

$$\frac{1}{m} \sum_{j=1}^{m} \left\| u^{(k)}(w_t, \mathcal{S}_j) - u^{(k)}(w_t', \mathcal{S}_j') \right\| \leq \left(1 + (1-\lambda\eta)^{k-1}\right) \left\| w_t - w_t' \right\| + \frac{2G}{\lambda m}$$

$$\leq 2 \left\| w_t - w_t' \right\| + \frac{2G}{\lambda m} \tag{11}$$

Finally, at iteration $t$, we have

$$\left\| w_{t+1} - w_{t+1}' \right\|$$

$$\leq \left\| w_t - \gamma\lambda \left( w_t - \frac{1}{m}\sum_{j=1}^{m}\frac{1}{K}\sum_{k=1}^{K} u^{(k)}(w_t, \mathcal{S}_j) \right) - w_t' + \gamma\lambda \left( w_t' - \frac{1}{m}\sum_{j=1}^{m}\frac{1}{K}\sum_{k=1}^{K} u^{(k)}(w_t', \mathcal{S}_j') \right) \right\|$$

(Projection is non-expansive)

$$= \left\| (1-\gamma\lambda)(w_t - w_t') + \gamma\lambda\frac{1}{m}\sum_{j=1}^{m}\frac{1}{K}\sum_{k=1}^{K} \left( u^{(k)}(w_t, \mathcal{S}_j) - u^{(k)}(w_t', \mathcal{S}_j') \right) \right\|$$

$$\leq (1-\gamma\lambda) \left\| w_t - w_t' \right\| + \gamma\lambda \left( 2 \left\| w_t - w_t' \right\| + \frac{2G}{\lambda m} \right) \qquad \text{(Equation (11))}$$

$$= (1+\gamma\lambda) \left\| w_t - w_t' \right\| + \frac{2\gamma G}{m}$$

Note that $w_1 = w_1' = 0$. Choosing $\gamma \leq \frac{1}{\lambda T}$ and telescoping gives us that

$$\left\| w_{T+1} - w_{T+1}' \right\| \leq \left(1 + \frac{1}{T}\right) \left\| w_t - w_t' \right\| + \frac{2G}{m\lambda T} \leq \frac{2eG}{\lambda m} \tag{12}$$

where the inequality holds because $(1 + \frac{1}{T})^T \leq e$. Similarly, we have

$$\left\| u^{(k+1)}(w, \mathcal{S}) - u^{(k+1)}(w', \mathcal{S}^{(i)}) \right\|$$

$$\leq \left\| u^{(k)}(w, \mathcal{S}) - u^{(k)}(w', \mathcal{S}^{(i)}) - \eta\Big( \nabla L(u^{(k)}(w, \mathcal{S}), \mathcal{S}) + \lambda \left( u^{(k)}(w, \mathcal{S}) - w \right) \right.$$

(Projection is non-expansive)

$$\left. - \nabla L(u^{(k)}(w', \mathcal{S}^{(i)}), \mathcal{S}^{(i)}) - \lambda \left( u^{(k)}(w', \mathcal{S}^{(i)}) - w' \right) \Big) \right\|$$

$$\leq \eta\lambda \left\| w - w' \right\| + \left\| (1-\eta\lambda) \left( u^{(k)}(w, \mathcal{S}) - u^{(k)}(w', \mathcal{S}^{(i)}) \right) \right.$$

$$\left. + \eta \left( \nabla L(u^{(k)}(w, \mathcal{S}), \mathcal{S}) - \nabla L(u^{(k)}(w', \mathcal{S}^{(i)}), \mathcal{S}) \right) \right.$$

$$+ \eta \left( \nabla L(\mathbf{u}^{(k)}(\mathbf{w}', \mathcal{S}^{(i)}), \mathcal{S}) - \nabla L(\mathbf{u}^{(k)}(\mathbf{w}', \mathcal{S}^{(i)}), \mathcal{S}^{(i)}) \right) \Big\|$$

$$\leq \eta\lambda \left\| \mathbf{w} - \mathbf{w}' \right\| + \frac{2G\eta}{n} + \left( \left\| (1 - \eta\lambda) \left( \mathbf{u}^{(k)}(\mathbf{w}, \mathcal{S}) - \mathbf{u}^{(k)}(\mathbf{w}', \mathcal{S}^{(i)}) \right) \right. \right.$$

$$+ \eta \left( \nabla L(\mathbf{u}^{(k)}(\mathbf{w}, \mathcal{S}), \mathcal{S}) - \nabla L(\mathbf{u}^{(k)}(\mathbf{w}', \mathcal{S}^{(i)}), \mathcal{S}) \right) \Big\|^2 \Big)^{1/2}$$

$$\leq \eta\lambda \left\| \mathbf{w} - \mathbf{w}' \right\| + \frac{2G\eta}{n} + \left( (1 - \lambda\eta)^2 \left\| \mathbf{u}^{(k)}(\mathbf{w}, \mathcal{S}) - \mathbf{u}^{(k)}(\mathbf{w}', \mathcal{S}^{(i)}) \right\|^2 \right.$$

$$+ \left( \eta^2 - \frac{2\eta(1 - \eta\lambda)}{H} \right) \left\| \nabla L(\mathbf{u}^{(k)}(\mathbf{w}, \mathcal{S}), \mathcal{S}) - \nabla L(\mathbf{u}^{(k)}(\mathbf{w}', \mathcal{S}), \mathcal{S}^{(i)}) \right\|^2 \Big)^{1/2}$$

$$\text{($L$ is $H$-smooth, $\eta \leq \frac{2}{H+2\lambda}$)}$$

$$\leq (1 - \lambda\eta) \left\| \mathbf{u}^{(k)}(\mathbf{w}, \mathcal{S}) - \mathbf{u}^{(k)}(\mathbf{w}', \mathcal{S}^{(i)}) \right\| + \eta\lambda \left\| \mathbf{w} - \mathbf{w}' \right\| + \frac{2G\eta}{n} \tag{13}$$

Therefore we have $\forall k \in [K - 1]$,

$$\left\| \mathbf{u}^{(k+1)}(\mathbf{w}_{T+1}, \mathcal{S}) - \mathbf{u}^{(k+1)}(\mathbf{w}'_{T+1}, \mathcal{S}^{(i)}) \right\| \leq \left( 1 + (1 - \lambda\eta)^{k-1} \right) \left\| \mathbf{w}_{T+1} - \mathbf{w}'_{T+1} \right\| + \frac{2G}{\lambda n}.$$

As $\mathbf{u}^{(1)}(\mathbf{w}_{T+1}, \mathcal{S}) = \mathbf{w}_{T+1}$, $\mathbf{u}^{(1)}(\mathbf{w}'_{T+1}, \mathcal{S}) = \mathbf{w}'_{T+1}$, plug in Equation (12), we have

$$\left\| \frac{1}{K} \sum_{k=1}^{K} \mathbf{u}^{(k)}(\mathbf{w}_{T+1}, \mathcal{S}) - \frac{1}{K} \sum_{k=1}^{K} \mathbf{u}^{(k)}(\mathbf{w}'_{T+1}, \mathcal{S}^{(i)}) \right\|$$

$$\leq \min \left( 2, 1 + \frac{1}{\lambda\eta K} \right) \left\| \mathbf{w}_{T+1} - \mathbf{w}'_{T+1} \right\| + \frac{2G}{\lambda n}$$

$$\leq \min \left( 2, 1 + \frac{1}{\lambda\eta K} \right) \frac{2eG}{\lambda m} + \frac{2G}{\lambda n}$$

$$\leq \frac{4eG}{\lambda m} + \frac{2G}{\lambda n}$$

$\square$

The following theorem can be derived via Theorem 2.2 and Lemma 4.3.

**Theorem C.3.** Consider a meta-learning problem with convex, $M$-bounded, $G$-Lipschitz and $H$-smooth loss function. Then, after $T$ iterations of Algorithm 1 with $\gamma \leq \frac{1}{\lambda T}$ on a given meta-sample **S**, and GD for task-specific learning (i.e., Option 2 for Algorithm 2) with $\eta \leq \frac{2}{H+2\lambda}$, we have with probability at least $1 - \delta$,

$$L(\mathcal{A}(\mathbf{S}), \mu) \lesssim L(\mathcal{A}(\mathbf{S}), \mathbf{S}) + \frac{G^2}{\lambda m} \log(m) \log(1/\delta) + \frac{M}{\sqrt{m}} \sqrt{\log(1/\delta)} + \frac{G^2}{\lambda n}.$$

*Proof of Theorem C.3.* We slightly abuse the notation, at iteration $t$, define $\mathbf{w}_t = \mathcal{A}(\mathbf{S})$, $\mathbf{w}'_t = \mathcal{A}(\mathbf{S}^{(j)})$, $\mathbf{u}^{(K)}(\mathbf{w}_{T+1}, \mathcal{S}) = \mathcal{A}(\mathbf{S})(\mathcal{S})$, $\mathbf{u}^{(K)}(\mathbf{w}'_{T+1}, \mathcal{S}^{(i)}) = \mathcal{A}(\mathbf{S}^{(j)})(\mathcal{S}^{(i)})$. If the loss is $M$-bounded,

convex and $G$-Lipschitz, apply Equation (10) gives us

$$\left| L(\mathcal{A}(\mathbf{S})(\mathcal{S}), \mathcal{S}) - L(\mathcal{A}(\mathbf{S}^{(j)})(\mathcal{S}), \mathcal{S}) \right|$$

$$= \left| L(\mathbf{u}^{(K)}(\mathbf{w}_{T+1}, \mathcal{S}), \mathcal{S}) - L(\mathbf{u}^{(K)}(\mathbf{w}'_{T+1}, \mathcal{S}), \mathcal{S}) \right|$$

$$\leq G \left\| \mathbf{u}^{(K)}(\mathbf{w}_{T+1}, \mathcal{S}) - \mathbf{u}^{(K)}(\mathbf{w}'_{T+1}, \mathcal{S}) \right\| \qquad \text{(}G\text{-Lipschitz)}$$

$$\leq G \left( (1 - \lambda\eta) \left\| \mathbf{u}^{(K)}(\mathbf{w}_{T+1}, \mathcal{S}) - \mathbf{u}^{(K)}(\mathbf{w}'_{T+1}, \mathcal{S}) \right\| + \lambda\eta \left\| \mathbf{w}_{T+1} - \mathbf{w}'_{T+1} \right\| \right) \quad \text{(Equation (10))}$$

$$\leq G \left\| \mathbf{w}_{T+1} - \mathbf{w}'_{T+1} \right\|$$

$$\leq \frac{2eG^2}{\lambda m} \qquad \text{(Equation (12))}$$

Given $\mathcal{S}, \mathcal{S}^{(i)}$. For any w, by Equation (13), for all $k \in [K-1]$, we have

$$\left\| \mathbf{u}^{(k+1)}(\mathbf{w}, \mathcal{S}) - \mathbf{u}^{(k+1)}(\mathbf{w}, \mathcal{S}^{(i)}) \right\| \leq \frac{2G\eta}{n} + (1 - \lambda\eta) \left\| \mathbf{u}^{(k)}(\mathbf{w}, \mathcal{S}) - \mathbf{u}^{(k)}(\mathbf{w}, \mathcal{S}^{(i)}) \right\| \leq \frac{2G}{\lambda n}$$
$$\text{(Telescope)}$$

Therefore, we have

$$\left| \ell(\mathcal{A}(\mathbf{S})(\mathcal{S}), \mathbf{z}) - \ell(\mathcal{A}(\mathbf{S})(\mathcal{S}^{(i)}), \mathbf{z}) \right|$$

$$= \left| \ell \left( \frac{1}{K} \sum_{k=1}^{K} \mathbf{u}^{(k)}(\mathbf{w}_{T+1}, \mathcal{S}), \mathbf{z} \right) - \ell \left( \frac{1}{K} \sum_{k=1}^{K} \mathbf{u}^{(k)}(\mathbf{w}_{T+1}, \mathcal{S}^{(i)}), \mathbf{z} \right) \right|$$

$$\leq G \left\| \frac{1}{K} \sum_{k=1}^{K} \mathbf{u}^{(k)}(\mathbf{w}_{T+1}, \mathcal{S}) - \frac{1}{K} \sum_{k=1}^{K} \mathbf{u}^{(k)}(\mathbf{w}_{T+1}, \mathcal{S}^{(i)}) \right\|$$

$$\leq \frac{2G^2}{\lambda n}$$

Apply Theorem 2.2 with $\beta' = \frac{2eG^2}{\lambda m}$, with $\beta = \frac{2G^2}{\lambda n}$ gives us the result.

$\square$

**Theorem 4.4.** Assume that the loss function is convex, $M$-bounded, $G$-Lipschitz and $H$-smooth. Suppose we run Algorithm 1 for $T$ iterations with $\gamma \leq \frac{1}{\lambda T}$ on a given meta-sample $\mathbf{S}$, and GD for task-specific learning (Option 2, Algorithm 2) with $\eta \leq \frac{2}{H+2\lambda}$. Then, with probability at least $1 - \delta$,

$$L(\mathcal{A}(\mathbf{S}), \mu) \lesssim L(\mathcal{A}(\mathbf{S}), \mathbf{S}) + \left( \frac{G^2}{\lambda m} + \frac{G^2}{\lambda n} \right) \log(mn) \log(1/\delta) + \frac{M\sqrt{\log(1/\delta)}}{\sqrt{mn}}.$$

*Proof of Theorem 4.4.* We denote $\mathbf{w}_t = \mathcal{A}(\mathbf{S})$, $\mathbf{w}'_t = \mathcal{A}(\mathbf{S}^{(j)})$, $\mathbf{u}(\mathbf{w}_{T+1}, \mathcal{S}) = \mathcal{A}(\mathbf{S})(\mathcal{S})$, $\mathbf{u}(\mathbf{w}_{T+1}, \mathcal{S}^{(i)}) = \mathcal{A}(\mathbf{S})(\mathcal{S}^{(i)})$. For $\ell$ to be convex and $G$-Lipschitz, applying Lemma 4.3 gives us that

$$\left| \ell(\mathcal{A}(\mathbf{S})(\mathcal{S}), \mathbf{z}) - \ell(\mathcal{A}(\mathbf{S}^{(j)})(\mathcal{S}^{(i)}), \mathbf{z}) \right|$$

$$= \left| \ell \left( \frac{1}{K} \sum_{k=1}^{K} \mathbf{u}^{(k)}(\mathbf{w}_{T+1}, \mathcal{S}), \mathbf{z} \right) - \ell \left( \frac{1}{K} \sum_{k=1}^{K} \mathbf{u}(\mathbf{w}'_{T+1}, \mathcal{S}^{(i)}), \mathbf{z} \right) \right|$$

$$\leq G \left\| \frac{1}{K} \sum_{k=1}^{K} \mathbf{u}^{(k)}(\mathbf{w}_{T+1}, \mathcal{S}), \mathbf{z}) - \frac{1}{K} \sum_{k=1}^{K} \mathbf{u}^{(k)}(\mathbf{w}'_{T+1}, \mathcal{S}^{(i)}) \right\|$$

$$\leq \min \left( 2, 1 + \frac{1}{\lambda\eta K} \right) \frac{2eG^2}{\lambda m} + \frac{2G^2}{\lambda n}.$$

Further apply Theorem 3.1 gives us the result. $\square$

# D  Missing Proofs of Section 4.2

We start with a proposition that provide some equivalent characterizations of weak convexity.

**Proposition D.1** (Proposition 2.1 in Davis and Grimmer [2019]). Suppose $f : \mathbb{R}^d \to \mathbb{R} \cup \{\infty\}$ is a closed function and $\rho > 0$, then the following are equivalent:

1. For any $w_1 \in \mathbb{R}^d$, $f(\cdot) + \frac{\rho}{2} \|\cdot - w_1\|$ is convex.

2. For any $w_1, w_2 \in \mathbb{R}^d$ with $g(w_1) \in \partial f(w_1)$, we have

$$f(w_2) \geq f(w_1) + \langle g(w_1), w_2 - w_1 \rangle - \frac{\rho}{2} \|w_2 - w_1\|^2.$$

3. For any $w_1, w_2 \in \mathbb{R}^d$ and $\lambda > 0$,

$$f(\lambda w_1 + (1 - \lambda)w_2) \leq \lambda f(w_1) + (1 - \lambda)f(w_2) + \frac{\rho \lambda (1 - \lambda)}{2} \|w_1 - w_2\|^2.$$

**Lemma D.2.** [Bassily et al. [2020]] Given $\mathcal{S}$ and $\mathcal{S}^{(i)}$, for a fixed $w$, consider $u(w, \mathcal{S})$ and $u(w, \mathcal{S}^{(i)})$ are achieved via Algo. 1 with gradient descent for $K$ iterations. Then if $\ell$ is convex and $G$-Lipschitz, we have $\sup_{i \in [n]} \left\| \frac{1}{K} \sum_{k=1}^{K} u^{(k)}(w, \mathcal{S}) - \frac{1}{K} \sum_{k=1}^{K} u^{(k)}(w, \mathcal{S}^{(i)}) \right\| \leq \frac{4GK\eta}{n} + 4G\eta\sqrt{K}$.

Below we provide our key Lemma D.3 for the stability analysis.

**Lemma D.3.** Consider a meta-learning problem with $\rho$-weakly convex and $G$-Lipschitz loss function. Let $\mathbf{S}, \mathbf{S}^{(j)}$ denote neighboring meta-samples and $\mathcal{S}, \mathcal{S}^{(i)}$ the neighboring samples on a test task. Then, after $T$ iterations of Algorithm 1 with $\gamma \leq \frac{1}{\lambda T}, \lambda \geq 2\rho$, and GD for task-specific learning (i.e., Option 2 for Algorithm 2) with $\eta \leq \frac{1}{\lambda}$,

$$\sup_{\mathbf{S}, j \in [m]} \|w_{T+1} - w'_{T+1}\| \leq 2eG\sqrt{\frac{\eta}{\lambda}} + \frac{2eG}{\lambda m}.$$

*Proof of Lemma D.3.* If $\ell$ is $\rho$-weakly convex, then from Proposition D.1 we have that

$$\langle \nabla \ell(u) - \nabla \ell(v), u - v \rangle \geq -\rho \|u - v\|^2, \forall u, v \in \mathbb{R}^d$$

Given $\mathcal{S}$ and $\mathcal{S}'$, for any $w$ and $w'$, we have

$$\forall k \in [K - 1], \quad \left\| u^{(k+1)}(w, \mathcal{S}) - u^{(k+1)}(w', \mathcal{S}') \right\|$$

$$\leq \left\| u^{(k)}(w, \mathcal{S}) - u^{(k)}(w', \mathcal{S}') - \eta \bigg( \nabla L(u^{(k)}(w, \mathcal{S}), \mathcal{S}) + \lambda \left( u^{(k)}(w, \mathcal{S}) - w \right) \right.$$

(Projection is non-expansive)

$$\left. - \nabla L(u^{(k)}(w', \mathcal{S}'), \mathcal{S}) - \lambda \left( u^{(k)}(w', \mathcal{S}') - w' \right) \bigg) \right\|$$

$$= (1 - \eta\lambda) \left\| u^{(k)}(w, \mathcal{S}) - u^{(k)}(w', \mathcal{S}') \right\| + \eta\lambda \|w - w'\| + 2\eta G$$

$$\leq \left( 1 + (1 - \eta\lambda)^k \right) \|w - w'\| + \frac{2G}{\lambda}$$

(Telescope, $u^{(1)}(w, \mathcal{S}) = w, u^{(1)}(w', \mathcal{S}) = w'$.)

And therefore

$$\left\| \frac{1}{K} \sum_{k=1}^{K} u^{(k)}(w, \mathcal{S}) - \frac{1}{K} \sum_{k=1}^{K} u^{(k)}(w', \mathcal{S}') \right\| \leq \min \left( 2, 1 + \frac{1}{\lambda\eta K} \right) \|w - w'\| + \frac{2G}{\lambda}$$

We now focus on the situation where we give $\mathcal{S}$ and $\mathcal{S}'$ with a fix $w$. For simplicity, we define $\delta_k = \left\| u^{(k)}(w_t, \mathcal{S}) - u^{(k)}(w'_t, \mathcal{S}) \right\|$. Note that $\delta_1 = \|w_t - w'_t\|$. We have

$$\delta_{k+1} = \left\| u^{(k+1)}(w_t, \mathcal{S}) - u^{(k+1)}(w'_t, \mathcal{S}) \right\|$$

$$\leq \left\| \mathbf{u}^{(k)}(\mathbf{w}_t, \mathcal{S}) - \mathbf{u}^{(k)}(\mathbf{w}'_t, \mathcal{S}) - \eta \left( \nabla L(\mathbf{u}^{(k)}(\mathbf{w}_t, \mathcal{S}), \mathcal{S}) + \lambda \left( \mathbf{u}^{(k)}(\mathbf{w}_t, \mathcal{S}) - \mathbf{w}_t \right) \right. \right.$$

(Projection is non-expansive)

$$\left. \left. - \nabla L(\mathbf{u}^{(k)}(\mathbf{w}'_t, \mathcal{S}), \mathcal{S}) - \lambda \left( \mathbf{u}^{(k)}(\mathbf{w}'_t, \mathcal{S}) - \mathbf{w}'_t \right) \right) \right\|$$

$$\leq \lambda \eta \, \|\mathbf{w}_t - \mathbf{w}'_t\| + \left\| (1 - \eta\lambda) \left( \mathbf{u}^{(k)}(\mathbf{w}_t, \mathcal{S}) - \mathbf{u}^{(k)}(\mathbf{w}'_t, \mathcal{S}) \right) \right.$$

$$\left. - \eta \left( \nabla L(\mathbf{u}^{(k)}(\mathbf{w}_t, \mathcal{S}), \mathcal{S}) - \nabla L(\mathbf{u}^{(k)}(\mathbf{w}'_t, \mathcal{S}), \mathcal{S}) \right) \right\|$$

$$= \lambda \eta \, \|\mathbf{w}_t - \mathbf{w}'_t\| + \Delta_k$$

where we define

$$\Delta_k = \left\| (1 - \eta\lambda) \left( \mathbf{u}^{(k)}(\mathbf{w}_t, \mathcal{S}) - \mathbf{u}^{(k)}(\mathbf{w}'_t, \mathcal{S}) \right) - \eta \left( \nabla L(\mathbf{u}^{(k)}(\mathbf{w}_t, \mathcal{S}), \mathcal{S}) - \nabla L(\mathbf{u}^{(k)}(\mathbf{w}'_t, \mathcal{S}), \mathcal{S}) \right) \right\|.$$

We have that

$$\Delta_k^2 = (1 - \eta\lambda)^2 \delta_k^2 + 4\eta^2 G^2$$
$$- 2\eta(1 - \eta\lambda) \left\langle \mathbf{u}^{(k)}(\mathbf{w}_t, \mathcal{S}) - \mathbf{u}^{(k)}(\mathbf{w}'_t, \mathcal{S}), \nabla L(\mathbf{u}^{(k)}(\mathbf{w}_t, \mathcal{S}), \mathcal{S}) - \nabla L(\mathbf{u}^{(k)}(\mathbf{w}'_t, \mathcal{S}), \mathcal{S}) \right\rangle$$
$$\leq (1 - \eta\lambda)^2 \delta_k^2 + 4\eta^2 G^2 + 2\eta(1 - \eta\lambda)\rho\delta_k^2 \qquad\qquad (\ell \text{ is } \rho\text{-weakly convex})$$
$$\leq (1 - \eta\lambda)\delta_k^2 + 4\eta^2 G^2 \qquad\qquad (\eta \leq \tfrac{1}{\lambda}, \lambda \geq 2\rho)$$

Therefore, we have

$$\delta_{k+1}^2 + \lambda^2\eta^2 \, \|\mathbf{w}_t - \mathbf{w}'_t\|^2 - 2\lambda\eta\delta_{k+1} \, \|\mathbf{w}_t - \mathbf{w}'_t\| \leq \Delta_k^2 \leq (1 - \eta\lambda)\delta_k^2 + 4\eta^2 G^2 \qquad (14)$$

Rearrange it gives us that

$$\frac{\delta_{k+1}^2}{(1 - \eta\lambda)^{k+1}} + \frac{\lambda^2\eta^2 \, \|\mathbf{w}_t - \mathbf{w}'_t\|^2}{(1 - \eta\lambda)^{k+1}} - \frac{2\lambda\eta \, \|\mathbf{w}_t - \mathbf{w}'_t\| \, \delta_{k+1}}{(1 - \eta\lambda)^{k+1}} \leq \frac{\delta_k^2}{(1 - \eta\lambda)^k} + \frac{4\eta^2 G^2}{(1 - \eta\lambda)^{k+1}}$$

Telescoping from $k = 1$ to $K$ gives us that

$$\frac{\delta_{K+1}^2}{(1 - \eta\lambda)^{K+1}} + \sum_{k=1}^{K} \frac{\lambda^2\eta^2 \, \|\mathbf{w}_t - \mathbf{w}'_t\|^2}{(1 - \eta\lambda)^{k+1}} \leq \sum_{k=1}^{K} \frac{2\lambda\eta \, \|\mathbf{w}_t - \mathbf{w}'_t\| \, \delta_{k+1}}{(1 - \eta\lambda)^{k+1}} + \sum_{k=1}^{K} \frac{4\eta^2 G^2}{(1 - \eta\lambda)^{k+1}}$$

Thus

$$\delta_{K+1}^2 + \lambda^2\eta^2 \, \|\mathbf{w}_t - \mathbf{w}'_t\|^2 \sum_{k=1}^{K} (1 - \eta\lambda)^{K-k} - 2\lambda\eta \, \|\mathbf{w}_t - \mathbf{w}'_t\| \, \delta_{K+1}$$

$$\leq \frac{4\eta G^2}{\lambda} + 2\lambda\eta \, \|\mathbf{w}_t - \mathbf{w}'_t\| \sum_{k=1}^{K-1} \delta_{k+1}(1 - \eta\lambda)^{K-k}$$

$$\leq \frac{4\eta G^2}{\lambda} + 2\lambda\eta(1 - \eta\lambda) \, \|\mathbf{w}_t - \mathbf{w}'_t\| \sum_{k=1}^{K} \delta_k(1 - \eta\lambda)^{K-k} \qquad (15)$$

Now we start proving the following bound by induction:

$$\delta_K \leq 2 \, \|\mathbf{w}_t - \mathbf{w}'_t\| + 2G\sqrt{\frac{\eta}{\lambda}}.$$

This claim holds when $k = 1$. For the inductive step, we assume it holds for some $k \in [K]$ and prove the result for $k + 1$. We consider the following two cases. If $\delta_{k+1} \leq \max_{s \in [k]} \delta_k$, induction automatically holds. Otherwise, $\delta_{k+1} > \max_{s \in [k]} \delta_s$. Applying Equation (15) gives us that

$$\delta_{k+1}^2 + \lambda^2 \eta^2 \left\| \mathbf{w}_t - \mathbf{w}_t' \right\|^2 \sum_{j=1}^{k} (1 - \eta\lambda)^{k-j} - 2\lambda\eta \left\| \mathbf{w}_t - \mathbf{w}_t' \right\| \delta_{k+1}$$

$$\leq \frac{4\eta G^2}{\lambda} + 2\lambda\eta(1 - \eta\lambda) \left\| \mathbf{w}_t - \mathbf{w}_t' \right\| \sum_{j=1}^{k} \delta_k (1 - \eta\lambda)^{k-j}$$

$$\leq \frac{4\eta G^2}{\lambda} + 2\lambda\eta(1 - \eta\lambda) \left\| \mathbf{w}_t - \mathbf{w}_t' \right\| \delta_{k+1} \sum_{j=1}^{k} (1 - \eta\lambda)^{k-j}$$

which is equivalent to

$$\delta_{k+1}^2 + \lambda^2 \eta^2 \left\| \mathbf{w}_t - \mathbf{w}_t' \right\|^2 \sum_{j=1}^{k} (1 - \eta\lambda)^{k-j}$$

$$\leq \frac{4\eta G^2}{\lambda} + 2\lambda\eta \left\| \mathbf{w}_t - \mathbf{w}_t' \right\| \delta_{k+1} \left( 1 + (1 - \eta\lambda) \sum_{j=1}^{k} (1 - \eta\lambda)^{k-j} \right)$$

Rearrange gives us that

$$\left( \delta_{k+1} - \lambda\eta \left\| \mathbf{w}_t - \mathbf{w}_t' \right\| \left( 1 + (1 - \eta\lambda) \sum_{j=1}^{k} (1 - \eta\lambda)^{k-j} \right) \right)^2$$

$$\leq \left( \lambda\eta \left\| \mathbf{w}_t - \mathbf{w}_t' \right\| \left( 1 + (1 - \eta\lambda) \sum_{j=1}^{k} (1 - \eta\lambda)^{k-j} \right) \right)^2 + \frac{4\eta G^2}{\lambda}$$

$$- \lambda\eta \left\| \mathbf{w}_t - \mathbf{w}_t' \right\|^2 \left( 1 - (1 - \eta\lambda)^{k+1} \right)$$

Therefore, we have

$$\forall k \in [K-1] \quad \left\| \mathbf{u}^{(k+1)}(\mathbf{w}_t, \mathcal{S}) - \mathbf{u}^{(k+1)}(\mathbf{w}_t', \mathcal{S}) \right\|$$

$$\leq 2G\sqrt{\frac{\eta}{\lambda}} + 2 \left( \lambda\eta \left\| \mathbf{w}_t - \mathbf{w}_t' \right\| \left( 1 + (1 - \eta\lambda) \sum_{j=1}^{k} (1 - \eta\lambda)^{k-j} \right) \right)$$

$$\leq 2 \left\| \mathbf{w}_t - \mathbf{w}_t' \right\| + 2G\sqrt{\frac{\eta}{\lambda}}$$

And therefore

$$\left\| \frac{1}{K} \sum_{k=1}^{K} \mathbf{u}^{(k)}(\mathbf{w}_t, \mathcal{S}) - \frac{1}{K} \sum_{k=1}^{K} \mathbf{u}^{(k)}(\mathbf{w}_t', \mathcal{S}) \right\| \leq 2 \left\| \mathbf{w}_t - \mathbf{w}_t' \right\| + 2G\sqrt{\frac{\eta}{\lambda}} \tag{16}$$

As a result,

$$\left\| \mathbf{w}_{t+1} - \mathbf{w}_{t+1}' \right\|$$

$$\leq \left\| \mathbf{w}_t - \gamma\lambda \left( \mathbf{w}_t - \frac{1}{m} \sum_{j=1}^{m} \frac{1}{K} \sum_{k=1}^{K} \mathbf{u}^{(k)}(\mathbf{w}_t, \mathcal{S}_j) \right) - \mathbf{w}_t' + \gamma\lambda \left( \mathbf{w}_t' - \frac{1}{m} \sum_{j=1}^{m} \frac{1}{K} \sum_{k=1}^{K} \mathbf{u}^{(k)}(\mathbf{w}_t', \mathcal{S}_j') \right) \right\|$$

$$\text{(Projection is non-expansive)}$$

$$= (1 - \gamma\lambda) \left\| \mathbf{w}_t - \mathbf{w}_t' \right\| + \gamma\lambda \left\| \frac{1}{m} \sum_{j=1}^{m} \frac{1}{K} \sum_{k=1}^{K} \mathbf{u}^{(k)}(\mathbf{w}_t, \mathcal{S}_j) - \frac{1}{m} \sum_{j=1}^{m} \frac{1}{K} \sum_{k=1}^{K} \mathbf{u}^{(k)}(\mathbf{w}_t', \mathcal{S}_j') \right\|$$

$$\leq (1 - \gamma\lambda) \|\mathbf{w}_t - \mathbf{w}'_t\| + \frac{m-1}{m}\gamma\lambda\left(2\|\mathbf{w}_t - \mathbf{w}'_t\| + 2G\sqrt{\frac{\eta}{\lambda}}\right) + \frac{\gamma\lambda}{m}\left(2\|\mathbf{w}_t - \mathbf{w}'_t\| + \frac{2G}{\lambda}\right)$$

$$\leq (1 + \gamma\lambda)\|\mathbf{w}_t - \mathbf{w}'_t\| + 2G\gamma\sqrt{\eta\lambda} + \frac{2G\gamma}{m}$$

Telescoping gives us that

$$\left\|\mathbf{w}_{T+1} - \mathbf{w}'_{T+1}\right\| \leq (1 + \gamma\lambda)^T\left(2G\sqrt{\frac{\eta}{\lambda}} + \frac{2G}{\lambda m}\right)$$

Choosing $\gamma \leq \frac{1}{\lambda T}$ gives us that

$$\left\|\mathbf{w}_{T+1} - \mathbf{w}'_{T+1}\right\| \leq (1 + \frac{1}{T})^T\left(2G\sqrt{\frac{\eta}{\lambda}} + \frac{2G}{\lambda m}\right) \leq 2eG\sqrt{\frac{\eta}{\lambda}} + \frac{2eG}{\lambda m}$$

We remark that if we consider convex and non-smooth loss function by setting $\rho = 0$, then follow a similar argument, Equation (14) can be replaced by

$$\delta_{k+1}^2 + \lambda^2\eta^2\|\mathbf{w}_t - \mathbf{w}'_t\|^2 - 2\lambda\eta\delta_{k+1}\|\mathbf{w}_t - \mathbf{w}'_t\| \leq \Delta_k^2 \leq (1 - \eta\lambda)^2\delta_k^2 + 4\eta^2 G^2$$

And therefore Equation (16) can be replaced by

$$\left\|\mathbf{u}^{(k+1)}(\mathbf{w}_t, \mathcal{S}) - \mathbf{u}^{(k+1)}(\mathbf{w}'_t, \mathcal{S})\right\|$$

$$\leq 2G\sqrt{\frac{\eta}{\lambda}} + 2\left(\lambda\eta\|\mathbf{w}_t - \mathbf{w}'_t\|\left(1 + (1 - \eta\lambda)^2\sum_{j=1}^{k}(1 - \eta\lambda)^{2k-2j}\right)\right)$$

$$\leq \frac{2}{2 - \eta\lambda}\|\mathbf{w}_t - \mathbf{w}'_t\| + 2G\sqrt{\frac{\eta}{\lambda}}$$

$$\leq 2\|\mathbf{w}_t - \mathbf{w}'_t\| + 2G\sqrt{\frac{\eta}{\lambda}} \qquad\qquad (\eta\lambda \leq 1)$$

and the rest follows.

$\square$

**Theorem D.4.** Consider a meta-learning problem with $\rho$-weakly convex, $M$-bounded, $G$-Lipschitz loss function. Then, after $T$ iterations of Algorithm 1 with $\gamma \leq \frac{1}{\lambda T}, \lambda \geq 2\rho$, and GD for task-specific learning (i.e., Option 2 for Algorithm 2) with $\eta \leq \frac{1}{\lambda}$, we have with probability at least $1 - \delta$,

$$L_{(}\mathcal{A}(\mathbf{S}), \mu) \lesssim L(\mathcal{A}(\mathbf{S}), \mathbf{S}) + \left(G^2\sqrt{\frac{\eta}{\lambda}} + \frac{G^2}{\lambda m}\right)\log(m)\log(1/\delta) + \frac{M}{\sqrt{m}}\sqrt{\log(1/\delta)} + \frac{G^2}{\lambda n} + G^2\eta\sqrt{K}.$$

By setting $\eta = \frac{1}{\lambda K}$, we have

$$L_{(}\mathcal{A}(\mathbf{S}), \mu) \lesssim L(\mathcal{A}(\mathbf{S}), \mathbf{S}) + \left(\frac{G^2}{\lambda\sqrt{K}} + \frac{G^2}{\lambda m}\right)\log(m)\log(1/\delta) + \frac{M}{\sqrt{m}}\sqrt{\log(1/\delta)} + \frac{G^2}{\lambda n} + \frac{G^2}{\lambda\sqrt{K}}.$$

*Proof of Theorem D.4.* If the loss is $M$-bounded and $G$-Lipschitz, apply Lemma D.3 gives us

$$\left|L(\mathcal{A}(\mathbf{S})(\mathcal{S}), \mathcal{S}) - L(\mathcal{A}(\mathbf{S}^{(j)})(\mathcal{S}), \mathcal{S})\right|$$

$$= \left|L\left(\frac{1}{K}\sum_{k=1}^{K}\mathbf{u}^{(k)}(\mathbf{w}_{T+1}, \mathcal{S}), \mathcal{S}\right) - L\left(\frac{1}{K}\sum_{k=1}^{K}\mathbf{u}^{(k)}(\mathbf{w}'_{T+1}, \mathcal{S}), \mathcal{S}\right)\right|$$

$$\leq G\left\|\frac{1}{K}\sum_{k=1}^{K}\mathbf{u}^{(k)}(\mathbf{w}_{T+1}, \mathcal{S}) - \frac{1}{K}\sum_{k=1}^{K}\mathbf{u}^{(k)}(\mathbf{w}'_{T+1}, \mathcal{S})\right\| \qquad (G\text{-Lipschitz})$$

$$\leq 2G\left\|\mathbf{w}_{T+1} - \mathbf{w}'_{T+1}\right\| + 2G^2\sqrt{\frac{\eta}{\lambda}} \qquad (\text{Equation (16)})$$

$$\leq \left(4eG^2 + 2G^2\right)\sqrt{\frac{\eta}{\lambda}} + \frac{4eG^2}{\lambda m} \qquad (\text{Lemma D.3})$$

$$\leq \left(4eG^2 + 2G^2\right)\frac{1}{\lambda\sqrt{K}} + \frac{4eG^2}{\lambda m} \qquad (\text{Set } \eta \leq \frac{1}{\lambda K})$$

On the other hand, applying Lemma D.2 gives us that

$$
\begin{aligned}
\left|\ell(\mathcal{A}(\mathbf{S})(\mathcal{S}), \mathbf{z}) - \ell(\mathcal{A}(\mathbf{S})(\mathcal{S}^{(i)}), \mathbf{z})\right| &= \left|\ell(\mathbf{u}^{(K+1)}(\mathbf{w}_{T+1}, \mathcal{S}), \mathbf{z}) - \ell(\mathbf{u}^{(K+1)}(\mathbf{w}_{T+1}, \mathcal{S}^{(i)}), \mathbf{z})\right| \\
&\leq G \left\|\mathbf{u}^{(K+1)}(\mathbf{w}_T, \mathcal{S}) - \mathbf{u}^{(K+1)}(\mathbf{w}_T, \mathcal{S}^{(i)})\right\| \\
&\leq \frac{4G^2 K\eta}{n} + 4G^2\eta\sqrt{K} \\
&\leq \frac{4G^2}{\lambda n} + \frac{4G^2}{\lambda\sqrt{K}} \qquad\qquad\qquad \text{(Set } \eta \leq \tfrac{1}{\lambda K}\text{)}
\end{aligned}
$$

Plug back into Theorem 2.2 gives the result. $\qquad\square$

**Lemma 4.5.** Assume that the loss function is $\rho$-weakly convex and $G$-Lipschitz. Let $\mathbf{S}, \mathbf{S}^{(j)}$ denote neighboring meta-samples and $\mathcal{S}, \mathcal{S}^{(i)}$ the neighboring samples on a test task. Then the following holds for Algorithm 1 with $\lambda \geq 2\rho$, and GD for task-specific learning (i.e., Option 2 for Algorithm 2) with $\eta \leq \frac{1}{\lambda}$, for all $T \geq 1$ as long as we set $\gamma \leq \frac{1}{\lambda T}$,

$$
\sup_{\mathbf{S},\mathcal{S},j\in[m],i\in[n]} \left\|\mathcal{A}(\mathbf{S})(\mathcal{S}) - \mathcal{A}(\mathbf{S}^{(j)})(\mathcal{S}^{(i)})\right\| \leq (8eG + 2G)\sqrt{\frac{\eta}{\lambda}} + \frac{8eG}{\lambda m} + \frac{8G}{\lambda n}.
$$

*Proof of Lemma 4.5.* We slightly abuse the notation, at outer iteration $t$, define $\mathbf{w}_t = \mathcal{A}(\mathbf{S})$, $\mathbf{w}'_t = \mathcal{A}(\mathbf{S}^{(j)})$. Given $\mathbf{w}_t$, at inner iteration $k$, define $\mathbf{u}^{(k)}(\mathbf{w}_t, \mathcal{S}) = \mathcal{A}(\mathbf{S})(\mathcal{S})$, $\mathbf{u}^{(k)}(\mathbf{w}'_t, \mathcal{S}^{(i)}) = \mathcal{A}(\mathbf{S}^{(j)})(\mathcal{S}^{(i)})$. We now provide the upper bound on $\left\|\frac{1}{K}\sum_{k=1}^{K} \mathbf{u}^{(k)}(\mathbf{w}_{T+1}, \mathcal{S}) - \frac{1}{K}\sum_{k=1}^{K} \mathbf{u}^{(k)}(\mathbf{w}'_{T+1}, \mathcal{S}^{(i)})\right\|$. Recall that if $\ell$ is $\rho$-weakly convex, then we have

$$
\langle \nabla\ell(\mathbf{u}) - \nabla\ell(\mathbf{v}), \mathbf{u} - \mathbf{v}\rangle \geq -\rho\|\mathbf{u} - \mathbf{v}\|^2
$$

We apply a similar procedure as Lemma D.3. For simplicity, we define $\delta_k = \left\|\mathbf{u}^{(k)}(\mathbf{w}_t, \mathcal{S}) - \mathbf{u}^{(k)}(\mathbf{w}'_t, \mathcal{S}^{(i)})\right\|$. Note that $\delta_1 = \|\mathbf{w}_t - \mathbf{w}'_t\|$. We have

$$
\begin{aligned}
\delta_{k+1} &= \left\|\mathbf{u}^{(k+1)}(\mathbf{w}_t, \mathcal{S}) - \mathbf{u}^{(k+1)}(\mathbf{w}'_t, \mathcal{S}^{(i)})\right\| \\
&= \left\|\mathbf{u}^{(k)}(\mathbf{w}_t, \mathcal{S}) - \mathbf{u}^{(k)}(\mathbf{w}'_t, \mathcal{S}^{(i)}) - \eta\Big(\nabla L(\mathbf{u}^{(k)}(\mathbf{w}_t, \mathcal{S}), \mathcal{S}) + \lambda\big(\mathbf{u}^{(k)}(\mathbf{w}_t, \mathcal{S}) - \mathbf{w}_t\big)\right. \\
&\qquad\left. - \nabla L(\mathbf{u}^{(k)}(\mathbf{w}'_t, \mathcal{S}^{(i)}), \mathcal{S}^{(i)}) - \lambda\big(\mathbf{u}^{(k)}(\mathbf{w}'_t, \mathcal{S}^{(i)}) - \mathbf{w}'_t\big)\Big)\right\| \\
&\leq \lambda\eta\|\mathbf{w}_t - \mathbf{w}'_t\| + \left\|(1 - \eta\lambda)\big(\mathbf{u}^{(k)}(\mathbf{w}_t, \mathcal{S}) - \mathbf{u}^{(k)}(\mathbf{w}'_t, \mathcal{S}^{(i)})\big)\right. \\
&\qquad\left. - \eta\big(\nabla L(\mathbf{u}^{(k)}(\mathbf{w}_t, \mathcal{S}), \mathcal{S}) - \nabla L(\mathbf{u}^{(k)}(\mathbf{w}'_t, \mathcal{S}^{(i)}), \mathcal{S}^{(i)})\big)\right\| \\
&= \lambda\eta\|\mathbf{w}_t - \mathbf{w}'_t\| + \Delta_k
\end{aligned}
$$

where we define

$$
\Delta_k = \left\|(1 - \eta\lambda)\big(\mathbf{u}^{(k)}(\mathbf{w}_t, \mathcal{S}) - \mathbf{u}^{(k)}(\mathbf{w}'_t, \mathcal{S}^{(i)})\big)\right.
$$
$$
\left. - \eta\big(\nabla L(\mathbf{u}^{(k)}(\mathbf{w}_t, \mathcal{S}), \mathcal{S}) - \nabla L(\mathbf{u}^{(k)}(\mathbf{w}'_t, \mathcal{S}^{(i)}), \mathcal{S}^{(i)})\big)\right\|.
$$

We have that

$$
\Delta_k^2 = (1 - \eta\lambda)^2\delta_k^2 + 4\eta^2 G^2
$$
$$
- 2\eta(1 - \eta\lambda)\Big\langle \mathbf{u}^{(k)}(\mathbf{w}_{T+1}, \mathcal{S}) - \mathbf{u}^{(k)}(\mathbf{w}'_{T+1}, \mathcal{S}^{(i)}), \nabla L(\mathbf{u}^{(k)}(\mathbf{w}_{T+1}, \mathcal{S}), \mathcal{S}) - \nabla L(\mathbf{u}^{(k)}(\mathbf{w}'_{T+1}, \mathcal{S}^{(i)}), \mathcal{S}^{(i)})\Big\rangle
$$

$$= (1 - \eta\lambda)^2 \delta_k^2 + 4\eta^2 G^2$$
$$- 2\eta(1 - \eta\lambda) \Big\langle \mathbf{u}^{(k)}(\mathbf{w}_{T+1}, \mathcal{S}) - \mathbf{u}^{(k)}(\mathbf{w}'_{T+1}, \mathcal{S}^{(i)}), \nabla L(\mathbf{u}^{(k)}(\mathbf{w}_{T+1}, \mathcal{S}), \mathcal{S}) - \nabla L(\mathbf{u}^{(k)}(\mathbf{w}'_{T+1}, \mathcal{S}^{(i)}), \mathcal{S}) \Big\rangle$$
$$- \frac{2\eta(1 - \eta\lambda)}{n} \Big\langle \mathbf{u}^{(k)}(\mathbf{w}_{T+1}, \mathcal{S}) - \mathbf{u}^{(k)}(\mathbf{w}'_{T+1}, \mathcal{S}^{(i)}), \nabla \ell(\mathbf{u}^{(k)}(\mathbf{w}'_{T+1}, \mathcal{S}^{(i)}), \mathbf{z}^i) - \nabla \ell(\mathbf{u}^{(k)}(\mathbf{w}'_{T+1}, \mathcal{S}^{(i)}), \mathbf{z}') \Big\rangle$$
$$\leq (1 - \eta\lambda)^2 \delta_k^2 + 4\eta^2 G^2 + 2\eta(1 - \eta\lambda)\rho\delta_k^2 + \frac{4G\eta(1 - \eta\lambda)}{n}\delta_k$$
$$\leq (1 - \eta\lambda)\delta_k^2 + 4\eta^2 G^2 + \frac{4G\eta(1 - \eta\lambda)}{n}\delta_k \qquad\qquad (\eta \leq \tfrac{1}{\lambda}, \lambda \geq 2\rho)$$

Therefore, we have

$$\delta_{k+1}^2 + \lambda^2\eta^2 \|\mathbf{w}_t - \mathbf{w}'_t\|^2 - 2\lambda\eta\delta_{k+1} \|\mathbf{w}_t - \mathbf{w}'_t\| \leq \Delta_k^2 \leq (1 - \eta\lambda)\delta_k^2 + 4\eta^2 G^2 + \frac{4G\eta(1 - \eta\lambda)}{n}\delta_k$$

Rearrange it gives us that

$$\frac{\delta_{k+1}^2}{(1 - \eta\lambda)^{k+1}} + \frac{\lambda^2\eta^2 \|\mathbf{w}_t - \mathbf{w}'_t\|^2}{(1 - \eta\lambda)^{k+1}} - \frac{2\lambda\eta \|\mathbf{w}_t - \mathbf{w}'_t\| \delta_{k+1}}{(1 - \eta\lambda)^{k+1}}$$
$$\leq \frac{\delta_k^2}{(1 - \eta\lambda)^k} + \frac{4\eta^2 G^2}{(1 - \eta\lambda)^{k+1}} + \frac{4G\eta(1 - \eta\lambda)\delta_k}{n(1 - \eta\lambda)^{k+1}}$$

Telescoping from $k = 1$ to $K$ gives us that

$$\frac{\delta_{K+1}^2}{(1 - \eta\lambda)^{K+1}} + \sum_{k=1}^{K} \frac{\lambda^2\eta^2 \|\mathbf{w}_t - \mathbf{w}'_t\|^2}{(1 - \eta\lambda)^{k+1}}$$
$$\leq \sum_{k=1}^{K} \frac{2\lambda\eta \|\mathbf{w}_t - \mathbf{w}'_t\| \delta_{k+1}}{(1 - \eta\lambda)^{k+1}} + \sum_{k=1}^{K} \frac{4\eta^2 G^2}{(1 - \eta\lambda)^{k+1}} + \sum_{k=1}^{K} \frac{4G\eta(1 - \eta\lambda)\delta_k}{n(1 - \eta\lambda)^{k+1}}$$

Thus

$$\delta_{K+1}^2 + \lambda^2\eta^2 \|\mathbf{w}_t - \mathbf{w}'_t\|^2 \sum_{k=1}^{K} (1 - \eta\lambda)^{K-k} - 2\lambda\eta \|\mathbf{w}_t - \mathbf{w}'_t\| \delta_{K+1}$$
$$\leq \frac{4\eta G^2}{\lambda} + 2\lambda\eta \|\mathbf{w}_t - \mathbf{w}'_t\| \sum_{k=1}^{K-1} \delta_{k+1}(1 - \eta\lambda)^{K-k} + \frac{4G\eta(1 - \eta\lambda)}{n} \sum_{k=1}^{K} \delta_k(1 - \eta\lambda)^{K-k}$$
$$\leq \frac{4\eta G^2}{\lambda} + \left( 2\lambda\eta(1 - \eta\lambda) \|\mathbf{w}_t - \mathbf{w}'_t\| + \frac{4G\eta(1 - \eta\lambda)}{n} \right) \sum_{k=1}^{K} \delta_k(1 - \eta\lambda)^{K-k} \qquad (17)$$

Now we start proving the following bound by induction:

$$\delta_K \leq 2G\sqrt{\frac{\eta}{\lambda}} + 4 \|\mathbf{w}_t - \mathbf{w}'_t\| + \frac{8G(1 - \eta\lambda)}{\lambda n} \qquad (18)$$

This claim holds when $k = 1$. For the inductive step, we assume it holds for some $k \in [K]$ and prove the result for $k + 1$. We consider the following two cases. If $\delta_{k+1} \leq \max_{s \in [k]} \delta_k$, induction automatically holds. Otherwise, $\delta_{k+1} > \max_{s \in [k]} \delta_s$. Applying Equation (17) gives us that

$$\delta_{k+1}^2 + \lambda^2\eta^2 \|\mathbf{w}_t - \mathbf{w}'_t\|^2 \sum_{j=1}^{k} (1 - \eta\lambda)^{k-j} - 2\lambda\eta \|\mathbf{w}_t - \mathbf{w}'_t\| \delta_{k+1}$$
$$\leq \frac{4\eta G^2}{\lambda} + \left( 2\lambda\eta(1 - \eta\lambda) \|\mathbf{w}_t - \mathbf{w}'_t\| + \frac{4G\eta(1 - \eta\lambda)}{n} \right) \delta_{k+1} \sum_{j=1}^{k} (1 - \eta\lambda)^{k-j}$$

which is equivalent to

$$\left(\delta_{k+1}-\left(2\lambda\eta\|\mathbf{w}_t-\mathbf{w}'_t\|+2\lambda\eta(1-\lambda\eta)\|\mathbf{w}_t-\mathbf{w}'_t\|\sum_{j=1}^{k}(1-\eta\lambda)^{k-j}+\frac{4G\eta(1-\eta\lambda)}{n}\sum_{j=1}^{k}(1-\lambda\eta)^{k-j}\right)\right)^2$$

$$\leq\frac{4\eta G^2}{\lambda}+\left(2\lambda\eta\|\mathbf{w}_t-\mathbf{w}'_t\|+2\lambda\eta(1-\lambda\eta)\|\mathbf{w}_t-\mathbf{w}'_t\|\sum_{j=1}^{k}(1-\eta\lambda)^{k-j}+\frac{4G\eta(1-\eta\lambda)}{n}\sum_{j=1}^{k}(1-\lambda\eta)^{k-j}\right)^2$$

Therefore, we have

$$\delta_{k+1}\leq 2G\sqrt{\frac{\eta}{\lambda}}+2\left(2\lambda\eta\|\mathbf{w}_t-\mathbf{w}'_t\|+2\lambda\eta(1-\lambda\eta)\|\mathbf{w}_t-\mathbf{w}'_t\|\sum_{j=1}^{k}(1-\eta\lambda)^{k-j}\right.$$

$$\left.+\frac{4G\eta(1-\eta\lambda)}{n}\sum_{j=1}^{k}(1-\lambda\eta)^{k-j}\right)$$

$$\leq 2G\sqrt{\frac{\eta}{\lambda}}+4\|\mathbf{w}_t-\mathbf{w}'_t\|+\frac{8G(1-\eta\lambda)}{\lambda n}$$

Plug in Lemma D.3 gives us that

$$\delta_{k+1}\leq(8eG+2G)\sqrt{\frac{\eta}{\lambda}}+\frac{8eG}{\lambda m}+\frac{8G}{\lambda n}$$

Therefore we have

$$\left\|\frac{1}{K}\sum_{k=1}^{K}\mathbf{u}^{(k)}(\mathbf{w}_{T+1},\mathcal{S})-\frac{1}{K}\sum_{k=1}^{K}\mathbf{u}^{(k)}(\mathbf{w}'_{T+1},\mathcal{S}^{(i)})\right\|\leq(8eG+2G)\sqrt{\frac{\eta}{\lambda}}+\frac{8eG}{\lambda m}+\frac{8G}{\lambda n}$$

Moreover, setting $\eta=\frac{1}{\lambda K}$ gives us that

$$\left\|\frac{1}{K}\sum_{k=1}^{K}\mathbf{u}^{(k)}(\mathbf{w}_{T+1},\mathcal{S})-\frac{1}{K}\sum_{k=1}^{K}\mathbf{u}^{(k)}(\mathbf{w}'_{T+1},\mathcal{S}^{(i)})\right\|\leq\frac{8eG+2G}{\lambda\sqrt{K}}+\frac{8eG}{\lambda m}+\frac{8G}{\lambda n}$$

$\square$

**Theorem 4.6.** Assume that the loss function is $\rho$-weakly convex, $M$-bounded, and $G$-Lipschitz. Suppose we run Algorithm 1 for $T$ iterations with $\gamma\leq\frac{1}{\lambda T}$, $\lambda\geq 2\rho$ on a meta-sample $\mathbf{S}$, and GD for task-specific learning (Option 2, Algorithm 2) with $\eta\leq\frac{1}{\lambda}$, Then, with probability at least $1-\delta$,

$$L(\mathcal{A}(\mathbf{S}),\mu)\lesssim L(\mathcal{A}(\mathbf{S}),\mathbf{S})+\left(G^2\sqrt{\frac{\eta}{\lambda}}+\frac{G^2}{\lambda m}+\frac{G^2}{\lambda n}\right)\log{(mn)}\log{(1/\delta)}+\frac{M\sqrt{\log{(1/\delta)}}}{\sqrt{mn}}.$$

*Proof of Theorem 4.6.* For $\ell$ to be $G$-Lipschitz, applying Lemma 4.1 gives us that

$$\left|\ell(\mathcal{A}(\mathbf{S})(\mathcal{S}),\mathbf{z})-\ell(\mathcal{A}(\mathbf{S}^{(j)})(\mathcal{S}^{(i)}),\mathbf{z})\right|$$

$$=\left|\ell\left(\frac{1}{K}\sum_{k=1}^{K}\mathbf{u}^{(k)}(\mathbf{w}_{T+1},\mathcal{S}),\mathbf{z}\right)-\ell\left(\frac{1}{K}\sum_{k=1}^{K}\mathbf{u}^{(k)}(\mathbf{w}'_{T+1},\mathcal{S}^{(i)}),\mathbf{z}\right)\right|$$

$$\leq G\left\|\frac{1}{K}\sum_{k=1}^{K}\mathbf{u}^{(k)}(\mathbf{w}_{T+1},\mathcal{S})-\frac{1}{K}\sum_{k=1}^{K}\mathbf{u}^{(k)}(\mathbf{w}'_{T+1},\mathcal{S}^{(i)})\right\|$$

$$\leq(8eG^2+2G^2)\sqrt{\frac{\eta}{\lambda}}+\frac{8eG^2}{\lambda m}+\frac{8G^2}{\lambda n}.$$

Plug it back into Theorem 3.1 gives the result. $\square$

**Theorem D.5** (Restatement of Theorem 4.7). Assume the loss $\ell$ is convex and $G$-Lipschitz. Define $u_j^* = \operatorname{argmin}_u L(u, \mathcal{S}_j), \forall j \in [m]$. Suppose we run Algorithm 1 with GD for task-specific learning with $\gamma = \frac{1}{\lambda T}$ to find an algorithm $\mathcal{A}(\mathbf{S}) = \mathcal{A}_{\text{task}}(w_{T+1}, \cdot)$ which is then run on $\mathcal{S}_j$ for $K$ iterations with step-size $\eta \leq \frac{1}{\lambda}$. Then, we have that

$$L(\mathcal{A}(\mathbf{S})(\mathcal{S}_j), \mathcal{S}_j) - \inf_u L(u, \mathcal{S}_j) \leq \frac{D^2}{2\eta(1-\eta\lambda)K} + \frac{G^2\eta}{2(1-\eta\lambda)} + \frac{GD\eta\lambda}{1-\eta\lambda} + \frac{\lambda\|w_{T+1} - \widehat{w}\|^2 + \lambda\sigma^2}{(1-\eta\lambda)(2-\eta\lambda)}$$

where $\sigma^2 := \frac{1}{K}\sum_{j=1}^K \|\widehat{w} - u_j^*\|^2$, with $\widehat{w}$ as defined in Equation (1). $\|w_{T+1} - \widehat{w}\|^2 \leq \frac{1}{T}\left(8D^2 + \frac{4D^2}{\eta\lambda K} + \frac{\eta(G+2\lambda D)^2}{\lambda}\right) + \frac{2D^2}{\eta\lambda K} + \frac{\eta(G+2\lambda D)^2}{2\lambda}$.

*Proof of Theorem D.5.* Recall the definition $\widehat{w} = \operatorname{argmin}_{w \in \mathcal{W}} \frac{1}{m}\sum_{j=1}^m \min_u \left[L(u; \mathcal{S}_j) + \frac{\lambda}{2}\|u - w\|^2\right]$, $u^*(w, \mathcal{S}) = \operatorname{argmin}_{u \in \mathcal{W}}\left[L(u; \mathcal{S}) + \frac{\lambda}{2}\|u - w\|^2\right]$, $u_j^* = \operatorname{argmin}_{u \in \mathcal{W}} L(u, \mathcal{S}_j), \forall j \in [m]$. We slightly abuse the notation by defining $u^{(k)}(w_t, \mathcal{S}_j) = \mathcal{A}(\mathbf{S})(\mathcal{S}_j)$ at inner iteration $k$ for given $w_t$. Then we have

$$\left\|u^{(k+1)}(w_t, \mathcal{S}_j) - u_j^*\right\|^2$$

$$= \left\|\Pi_{\mathcal{W}}\left(u^{(k)}(w_t, \mathcal{S}_j) - \eta\left(\nabla L(u^{(k)}(w_t, \mathcal{S}_j), \mathcal{S}_j) + \lambda(u^{(k)}(w_t, \mathcal{S}_j) - w_t))\right)\right) - u_j^*\right\|^2$$

$$\leq \left\|(1-\eta\lambda)\left(u^{(k)}(w_t, \mathcal{S}_j) - u_j^*\right) + \eta\lambda\left(w_t - u_j^*\right) - \eta\nabla L(u^{(k)}(w_t, \mathcal{S}_j), \mathcal{S}_j)\right\|^2$$

$$\leq (1-\eta\lambda)^2\left\|u^{(k)}(w_t, \mathcal{S}_j) - u_j^*\right\|^2 + \eta^2\left\|\nabla L(u^{(k)}(w_t, \mathcal{S}_j), \mathcal{S}_j)\right\|^2 + \eta^2\lambda^2\left\|w_t - u_j^*\right\|^2$$

$$\quad + 2\eta\lambda(1-\eta\lambda)\left\|u^{(k)}(w_t, \mathcal{S}_j) - u_j^*\right\|\left\|w_t - u_j^*\right\| + \eta^2\lambda G\left\|w_t - u_j^*\right\|$$

$$\quad - 2\eta(1-\eta\lambda)\left\langle\nabla L(u^{(k)}(w_t, \mathcal{S}_j), \mathcal{S}_j), u^{(k)}(w_t, \mathcal{S}_j) - u_j^*\right\rangle$$

$$= \left((1-\eta\lambda)\left\|u^{(k)}(w_t, \mathcal{S}_j) - u_j^*\right\| + \eta\lambda\left\|w_t - u_j^*\right\|\right)^2 + \eta^2\left\|\nabla L(u^{(k)}(w_t, \mathcal{S}_j), \mathcal{S}_j)\right\|^2$$

$$\quad + \eta^2\lambda G\left\|w_t - u_j^*\right\| - 2\eta(1-\eta\lambda)\left\langle\nabla L(u^{(k)}(w_t, \mathcal{S}_j), \mathcal{S}_j), u^{(k)}(w_t, \mathcal{S}_j) - u_j^*\right\rangle$$

$$\leq \left\|u^{(k)}(w_t, \mathcal{S}_j) - u_j^*\right\|^2 + \frac{\eta\lambda}{2-\eta\lambda}\left\|w_t - u_j^*\right\|^2 + \eta^2 G^2 + \eta^2\lambda G\left\|w_t - u_j^*\right\|$$

$$\hspace{4cm} ((a+b)^2 \leq (1+p)a^2 + (1+1/p)b^2 \text{ with } p = \frac{(2-\eta\lambda)\eta\lambda}{(1-\eta\lambda)^2})$$

$$\quad - 2\eta(1-\eta\lambda)\left\langle\nabla L(u^{(k)}(w_t, \mathcal{S}_j), \mathcal{S}_j), u^{(k)}(w_t, \mathcal{S}_j) - u_j^*\right\rangle$$

Rearrange it and telescope it gives us that

$$L\left(\frac{1}{K}\sum_{k=1}^K u^{(k)}(w_t, \mathcal{S}_j), \mathcal{S}_j\right) - L(u_j^*, \mathcal{S}_j)$$

$$\leq \frac{1}{K}\sum_{k=1}^K L(u^{(k)}(w_t, \mathcal{S}_j), \mathcal{S}_j) - L(u_j^*, \mathcal{S}_j) \hspace{2cm} \text{(Jensen's inequality)}$$

$$\leq \frac{1}{K}\sum_{k=1}^K \left\langle\nabla L(u^{(k)}(w_t, \mathcal{S}_j), \mathcal{S}_j), u^{(k)}(w_t, \mathcal{S}_j) - u_j^*\right\rangle \hspace{1.5cm} \text{(Convexity)}$$

$$\leq \frac{\left\|u_j^*\right\|^2 + \eta^2 G^2 K + \eta^2\lambda G\sum_{j=1}^K \left\|w_t - u_j^*\right\| + \frac{\eta\lambda}{2-\eta\lambda}\sum_{j=1}^K \left\|w_t - u_j^*\right\|^2}{2\eta(1-\eta\lambda)K}$$

$$\leq \frac{D^2}{2\eta(1-\eta\lambda)K} + \frac{G^2\eta}{2(1-\eta\lambda)} + \frac{2GD\eta\lambda}{1-\eta\lambda} + \frac{\lambda\|w_t - \widehat{w}\|^2 + \lambda\sigma^2}{(1-\eta\lambda)(2-\eta\lambda)} \quad (\sigma^2 = \frac{1}{K}\sum_{k=1}^K \|u_j^* - \widehat{w}\|^2)$$

Follow from [Zhou et al., 2019, Theorem 1], we now control $\|\mathbf{w}_{t+1} - \widehat{\mathbf{w}}\|^2$. Define $\mathbf{u}^*(\mathbf{w}_t, \mathcal{S}_j) = \arg\min_{\mathbf{u}} F_{\mathcal{S}_j}(\mathbf{u}, \mathbf{w}_t) = \arg\min_{\mathbf{u}} L(\mathbf{u}, \mathcal{S}_j) + \frac{\lambda}{2} \|\mathbf{u} - \mathbf{w}_t\|^2$. We start with the following:

$$
\begin{aligned}
\|\mathbf{w}_{t+1} - \widehat{\mathbf{w}}\|^2 &= \left\| \Pi_{\mathcal{W}} \left( \mathbf{w}_t - \gamma\lambda \left( \mathbf{w}_t - \frac{1}{m} \sum_{j=1}^{m} \frac{1}{K} \sum_{k=1}^{K} \mathbf{u}^{(k)}(\mathbf{w}_t, \mathcal{S}_j) \right) \right) - \widehat{\mathbf{w}} \right\|^2 \\
&\leq \left\| \mathbf{w}_t - \widehat{\mathbf{w}} - \gamma\lambda \left( \mathbf{w}_t - \frac{1}{m} \sum_{j=1}^{m} \frac{1}{K} \sum_{k=1}^{K} \mathbf{u}^{(k)}(\mathbf{w}_t, \mathcal{S}_j) \right) \right\|^2 \\
&\leq \|\mathbf{w}_t - \widehat{\mathbf{w}}\|^2 - 2\gamma\lambda \left\langle \mathbf{w}_t - \widehat{\mathbf{w}}, \mathbf{w}_t - \frac{1}{m} \sum_{j=1}^{m} \frac{1}{K} \sum_{k=1}^{K} \mathbf{u}^{(k)}(\mathbf{w}_t, \mathcal{S}_j) \right\rangle \\
&\quad + \gamma^2\lambda^2 \left\| \mathbf{w}_t - \frac{1}{m} \sum_{j=1}^{m} \frac{1}{K} \sum_{k=1}^{K} \mathbf{u}^{(k)}(\mathbf{w}_t, \mathcal{S}_j) \right\|^2 \qquad (19)
\end{aligned}
$$

We now bound the latter two terms separately as follows:

$$
\begin{aligned}
& \left\| \mathbf{w}_t - \frac{1}{m} \sum_{j=1}^{m} \frac{1}{K} \sum_{k=1}^{K} \mathbf{u}^{(k)}(\mathbf{w}_t, \mathcal{S}_j) \right\|^2 \\
&= \left\| \mathbf{w}_t - \frac{1}{m} \sum_{j=1}^{m} \frac{1}{K} \sum_{k=1}^{K} \mathbf{u}^*(\mathbf{w}_t, \mathcal{S}_j) + \frac{1}{m} \sum_{j=1}^{m} \frac{1}{K} \sum_{k=1}^{K} \left( \mathbf{u}^*(\mathbf{w}_t, \mathcal{S}_j) - \mathbf{u}^{(k)}(\mathbf{w}_t, \mathcal{S}_j) \right) \right\|^2 \\
&\leq 2 \left\| \mathbf{w}_t - \frac{1}{m} \sum_{j=1}^{m} \frac{1}{K} \sum_{k=1}^{K} \mathbf{u}^*(\mathbf{w}_t, \mathcal{S}_j) \right\|^2 + \frac{2}{m} \sum_{j=1}^{m} \frac{1}{K} \sum_{k=1}^{K} \left\| \left( \mathbf{u}^{(k)}(\mathbf{w}_t, \mathcal{S}_j) - \mathbf{u}^*(\mathbf{w}_t, \mathcal{S}_j) \right) \right\|^2 \\
&\leq 8D^2 + \frac{2}{m} \sum_{j=1}^{m} \frac{1}{K} \sum_{k=1}^{K} \left\| \left( \mathbf{u}^{(k)}(\mathbf{w}_t, \mathcal{S}_j) - \mathbf{u}^*(\mathbf{w}_t, \mathcal{S}_j) \right) \right\|^2
\end{aligned}
$$

as well as

$$
\begin{aligned}
& \left\langle \mathbf{w}_t - \widehat{\mathbf{w}}, \mathbf{w}_t - \frac{1}{m} \sum_{j=1}^{m} \frac{1}{K} \sum_{k=1}^{K} \mathbf{u}^{(k)}(\mathbf{w}_t, \mathcal{S}_j) \right\rangle \\
&= \frac{1}{m} \sum_{j=1}^{m} \frac{1}{K} \sum_{k=1}^{K} \left\langle \mathbf{w}_t - \widehat{\mathbf{w}}, \mathbf{w}_t - \mathbf{u}^*(\mathbf{w}_t, \mathcal{S}_j) \right\rangle \\
&\quad - \frac{1}{m} \sum_{j=1}^{m} \frac{1}{K} \sum_{k=1}^{K} \left\langle \mathbf{w}_t - \widehat{\mathbf{w}}, \mathbf{u}^{(k)}(\mathbf{w}_t, \mathcal{S}_j) - \mathbf{u}^*(\mathbf{w}_t, \mathcal{S}_j) \right\rangle \\
&\geq \frac{1}{m} \sum_{j=1}^{m} \frac{1}{K} \sum_{k=1}^{K} \left\langle \mathbf{w}_t - \widehat{\mathbf{w}}, \frac{1}{\lambda} \nabla F_{\mathcal{S}_j}(\mathbf{u}^{(k)}, \mathbf{w}_t) \right\rangle - \frac{1}{2} \|\mathbf{w}_t - \widehat{\mathbf{w}}\|^2 \\
&\quad - \frac{1}{2m} \sum_{j=1}^{m} \frac{1}{K} \sum_{k=1}^{K} \left\| \mathbf{u}^{(k)}(\mathbf{w}_t, \mathcal{S}_j) - \mathbf{u}^*(\mathbf{w}_t, \mathcal{S}_j) \right\|^2 \\
&\geq \|\mathbf{w}_t - \widehat{\mathbf{w}}\|^2 - \frac{1}{2} \|\mathbf{w}_t - \widehat{\mathbf{w}}\|^2 - \frac{1}{2m} \sum_{j=1}^{m} \frac{1}{K} \sum_{k=1}^{K} \left\| \mathbf{u}^{(k)}(\mathbf{w}_t, \mathcal{S}_j) - \mathbf{u}^*(\mathbf{w}_t, \mathcal{S}_j) \right\|^2 \\
&\qquad\qquad\qquad\qquad\qquad\qquad (\tfrac{1}{m} \sum_{j=1}^{m} F_{\mathcal{S}_j}(\mathbf{u}, \mathbf{w}) \text{ is } \lambda\text{-strongly convex w.r.t. } \mathbf{w}) \\
&= \frac{1}{2} \|\mathbf{w}_t - \widehat{\mathbf{w}}\|^2 - \frac{1}{2m} \sum_{j=1}^{m} \frac{1}{K} \sum_{k=1}^{K} \left\| \mathbf{u}^{(k)}(\mathbf{w}_t, \mathcal{S}_j) - \mathbf{u}^*(\mathbf{w}_t, \mathcal{S}_j) \right\|^2
\end{aligned}
$$

where the common term $\left\|\mathbf{u}^{(k)}(\mathbf{w}_t, \mathcal{S}_j) - \mathbf{u}^*(\mathbf{w}_t, \mathcal{S}_j)\right\|^2$ can be controlled as follows:

$$\left\|\mathbf{u}^{(k+1)}(\mathbf{w}_t, \mathcal{S}_j) - \mathbf{u}^*(\mathbf{w}_t, \mathcal{S}_j)\right\|^2$$

$$\leq \left\|\mathbf{u}^{(k)}(\mathbf{w}_t, \mathcal{S}_j) - \eta\nabla F_{\mathcal{S}_j}(\mathbf{u}^{(k)}(\mathbf{w}_t, \mathcal{S}_j), \mathbf{w}_t) - \mathbf{u}^*(\mathbf{w}_t, \mathcal{S}_j)\right\|^2$$

$$\leq \left\|\mathbf{u}^{(k+1)}(\mathbf{w}_t, \mathcal{S}_j) - \mathbf{u}^*(\mathbf{w}_t, \mathcal{S}_j)\right\|^2 - 2\eta\left\langle\nabla F_{\mathcal{S}_j}(\mathbf{u}^{(k)}(\mathbf{w}_t, \mathcal{S}_j), \mathbf{w}_t), \mathbf{u}^{(k+1)}(\mathbf{w}_t, \mathcal{S}_j) - \mathbf{u}^*(\mathbf{w}_t, \mathcal{S}_j)\right\rangle$$
$$+ \eta^2\left\|\nabla F_{\mathcal{S}_j}(\mathbf{u}^{(k)}(\mathbf{w}_t, \mathcal{S}_j), \mathbf{w}_t)\right\|^2$$

$$\leq (1-2\eta\lambda)\left\|\mathbf{u}^{(k+1)}(\mathbf{w}_t, \mathcal{S}_j) - \mathbf{u}^*(\mathbf{w}_t, \mathcal{S}_j)\right\|^2 + \eta^2\left\|\nabla L(\mathbf{u}^{(k)}(\mathbf{w}_t, \mathcal{S}_j), \mathcal{S}_j) + \lambda(\mathbf{u}^{(k)}(\mathbf{w}_t, \mathcal{S}_j) - \mathbf{w}_t)\right\|^2$$
$$(F_{\mathcal{S}_j}(\mathbf{u}, \mathbf{w}) \text{ is } \lambda\text{-strongly convex w.r.t. } \mathbf{u})$$

$$\leq (1-2\eta\lambda)\left\|\mathbf{u}^{(k+1)}(\mathbf{w}_t, \mathcal{S}_j) - \mathbf{u}^*(\mathbf{w}_t, \mathcal{S}_j)\right\|^2 + \eta^2(G + 2\lambda D)^2$$

$$\leq 4(1-2\eta\lambda)^k D^2 + \frac{\eta(G + 2\lambda D)^2}{2\lambda} \qquad\qquad \text{(Telescoping)}$$

Plug back into Equation (19) gives us that

$$\|\mathbf{w}_{t+1} - \widehat{\mathbf{w}}\|^2$$

$$\leq \|\mathbf{w}_t - \widehat{\mathbf{w}}\|^2 - 2\gamma\lambda\left(\frac{1}{2}\|\mathbf{w}_t - \widehat{\mathbf{w}}\|^2 - \frac{1}{2m}\sum_{j=1}^m\frac{1}{K}\sum_{k=1}^K\left\|\mathbf{u}^{(k)}(\mathbf{w}_t, \mathcal{S}_j) - \mathbf{u}^*(\mathbf{w}_t, \mathcal{S}_j)\right\|^2\right)$$

$$+ \gamma^2\lambda^2\left(8D^2 + \frac{2}{m}\sum_{j=1}^m\frac{1}{K}\sum_{k=1}^K\left\|\left(\mathbf{u}^{(k)}(\mathbf{w}_t, \mathcal{S}_j) - \mathbf{u}^*(\mathbf{w}_t, \mathcal{S}_j)\right)\right\|^2\right)$$

$$\leq (1 - \gamma\lambda)\|\mathbf{w}_t - \widehat{\mathbf{w}}\|^2 + \gamma\lambda\left(\frac{1}{K}\sum_{k=1}^K 4(1-2\eta\lambda)^k D^2 + \frac{\eta(G + 2\lambda D)^2}{2\lambda}\right)$$

$$+ \gamma^2\lambda^2\left(8D^2 + \frac{1}{K}\sum_{k=1}^K 8(1-2\eta\lambda)^k D^2 + \frac{\eta(G + 2\lambda D)^2}{\lambda}\right)$$

$$\leq (1 - \gamma\lambda)\|\mathbf{w}_t - \widehat{\mathbf{w}}\|^2 + \gamma\lambda\left(\frac{2D^2}{\eta\lambda K} + \frac{\eta(G + 2\lambda D)^2}{2\lambda}\right) + \gamma^2\lambda^2\left(8D^2 + \frac{4D^2}{\eta\lambda K} + \frac{\eta(G + 2\lambda D)^2}{\lambda}\right)$$

Choosing $\gamma = \frac{1}{\lambda T}$ gives us that

$$\|\mathbf{w}_{t+1} - \widehat{\mathbf{w}}\|^2$$

$$\leq (1 - \frac{1}{T})\|\mathbf{w}_t - \widehat{\mathbf{w}}\|^2 + \frac{1}{T}\left(\frac{2D^2}{\eta\lambda K} + \frac{\eta(G + 2\lambda D)^2}{2\lambda}\right) + \frac{1}{T^2}\left(8D^2 + \frac{4D^2}{\eta\lambda K} + \frac{\eta(G + 2\lambda D)^2}{\lambda}\right)$$

$$\leq \frac{\left(8D^2 + \frac{4D^2}{\eta\lambda K} + \frac{\eta(G + 2\lambda D)^2}{\lambda}\right)}{T} + \frac{2D^2}{\eta\lambda K} + \frac{\eta(G + 2\lambda D)^2}{2\lambda} \qquad\qquad \text{(Telescope)}$$

$$\square$$

# E   Missing Proofs in Section 5

**Theorem E.1** (Bennett's inequality). Let $x_1, \ldots, x_n$ be independent r.v. with finite variance. Further assume $|x_i - \mathbb{E}x_i| \leq a$ a.s. for all $i$. Define $S_n = \sum_{i=1}^n [x_i - \mathbb{E}[x_i]]$ and $\sigma^2 = \sum_{i=1}^n \mathbb{E}(x_i - \mathbb{E}[x_i])^2$. Then for any $t \geq 0$,

$$\mathrm{P}(S_n > t) \leq \exp\left(-\frac{\sigma^2}{a^2}h\left(\frac{at}{\sigma^2}\right)\right),$$

where $h(u) = (1 + u)\log(1 + u) - u$.

**Lemma 5.1.** Assume that the loss function is $\rho$-weakly convex and $G$-Lipschitz. Let $\mathbf{S}, \mathbf{S}^{(j)}$ denote neighboring meta-samples and $\mathcal{S}, \mathcal{S}^{(i)}$ the neighboring samples on a test task. Then, with probability at least $1 - \exp\left(-T^2 e^2/m^2\right)$, the following holds for Algorithm 3 with $\lambda \geq 2\rho$, and GD for task-specific learning (i.e., Option 2 for Algorithm 2) with $\eta \leq \frac{1}{\lambda}$, for all $T \geq 1$ as long as we set $\gamma \leq \frac{1}{\lambda T}$,

$$\sup_{\mathbf{S}, \mathcal{S}, i \in [n], j \in [m]} \left\| \mathcal{A}(\mathbf{S})(\mathcal{S}) - \mathcal{A}(\mathbf{S}^{(j)})(\mathcal{S}^{(i)}) \right\| \leq (8eG + 2G)\sqrt{\frac{\eta}{\lambda}} + \frac{8eG}{\lambda m} + \frac{8G}{\lambda n}.$$

*Proof of Lemma 5.1.* We slightly abuse the notation, at outer iteration $t$, define $\mathbf{w}_t = \mathcal{A}(\mathbf{S})$, $\mathbf{w}'_t = \mathcal{A}(\mathbf{S}^{(j)})$. Given $\mathbf{w}_t$, at inner iteration $k$, define $\mathbf{u}^{(k)}(\mathbf{w}_t, \mathcal{S}) = \mathcal{A}(\mathbf{S})(\mathcal{S})$, $\mathbf{u}^{(k)}(\mathbf{w}'_t, \mathcal{S}^{(i)}) = \mathcal{A}(\mathbf{S}^{(j)})(\mathcal{S}^{(i)})$. From a similar argument as Lemma D.3, $\forall k \in [K-1]$, we have

$$\left\| \mathbf{u}^{(k+1)}(\mathbf{w}_t, \mathcal{S}) - \mathbf{u}^{(k+1)}(\mathbf{w}'_t, \mathcal{S}') \right\| \leq 2\|\mathbf{w}_t - \mathbf{w}'_t\| + \frac{2G}{\lambda}$$

$$\left\| \mathbf{u}^{(k+1)}(\mathbf{w}_t, \mathcal{S}) - \mathbf{u}^{(k+1)}(\mathbf{w}'_t, \mathcal{S}) \right\| \leq 2\|\mathbf{w}_t - \mathbf{w}'_t\| + 2G\sqrt{\frac{\eta}{\lambda}}.$$

Let us define $r_t = \mathbb{1}(\mathcal{S}_{j_t} \neq \mathcal{S}_{j_t})$. Note that at every step $t$, $\mathbb{E}_{\mathcal{A}}(r_t) = \frac{1}{m}$. Moreover, note that $\{r_t : t \in [T]\}$ is an independent sequence of Bernoulli random variables. As a result,

$$\left\|\mathbf{w}_{t+1} - \mathbf{w}'_{t+1}\right\|$$

$$\leq \left\| \mathbf{w}_t - \gamma\lambda\left(\mathbf{w}_t - \frac{1}{K}\sum_{k=1}^{K} \mathbf{u}^{(k)}(\mathbf{w}_t, \mathcal{S}_{j_t})\right) - \mathbf{w}'_t + \gamma\lambda\left(\mathbf{w}'_t - \frac{1}{K}\sum_{k=1}^{K} \mathbf{u}^{(k)}(\mathbf{w}'_t, \mathcal{S}'_{j_t})\right) \right\|$$

$$\text{(Projection is non-expansive)}$$

$$= (1 - \gamma\lambda)\|\mathbf{w}_t - \mathbf{w}'_t\| + \gamma\lambda \left\| \frac{1}{K}\sum_{k=1}^{K} \mathbf{u}^{(k)}(\mathbf{w}_t, \mathcal{S}_{j_t}) - \frac{1}{K}\sum_{k=1}^{K} \mathbf{u}^{(k)}(\mathbf{w}'_t, \mathcal{S}'_{j_t}) \right\|$$

$$\leq (1 - \gamma\lambda)\|\mathbf{w}_t - \mathbf{w}'_t\| + \gamma\lambda(1 - r_t)\left(2\|\mathbf{w}_t - \mathbf{w}'_t\| + 2G\sqrt{\frac{\eta}{\lambda}}\right) + \gamma\lambda r_t\left(\|\mathbf{w}_t - \mathbf{w}'_t\| + \frac{2G}{\lambda}\right)$$

$$\leq (1 + (1 - r_t)\gamma\lambda)\|\mathbf{w}_t - \mathbf{w}'_t\| + 2G\gamma\sqrt{\eta\lambda} + 2G\gamma r_t$$

$$\leq (1 + \gamma\lambda)\|\mathbf{w}_t - \mathbf{w}'_t\| + 2G\gamma\sqrt{\eta\lambda} + 2G\gamma r_t$$

Telescoping gives us that

$$\left\|\mathbf{w}_{T+1} - \mathbf{w}'_{T+1}\right\| \leq 2G\sqrt{\frac{\eta}{\lambda}}(1 + \gamma\lambda)^T + 2G\gamma\sum_{t=1}^{T}(1 + \gamma\lambda)^{t-1} r_t$$

Further taking expectation w.r.t the randomness of the algorithm and gives us that

$$\mathbb{E}_{\mathcal{A}}\left\|\mathbf{w}_{T+1} - \mathbf{w}'_{T+1}\right\| \leq (1 + \gamma\lambda)^T \left(2G\sqrt{\frac{\eta}{\lambda}} + \frac{2G}{\lambda m}\right)$$

Choosing $\gamma \leq \frac{1}{\lambda T}$ gives us that

$$\mathbb{E}_{\mathcal{A}}\left\|\mathbf{w}_{T+1} - \mathbf{w}'_{T+1}\right\| \leq (1 + \frac{1}{T})^T \left(2G\sqrt{\frac{\eta}{\lambda}} + \frac{2G}{\lambda m}\right) \leq 2eG\sqrt{\frac{\eta}{\lambda}} + \frac{2eG}{\lambda m}$$

Plug this back into Equation (18) gives us that

$$\mathbb{E}_{\mathcal{A}}\left\| \mathbf{u}^{(K+1)}(\mathbf{w}_{T+1}, \mathcal{S}) - \mathbf{u}^{(K+1)}(\mathbf{w}'_{T+1}, \mathcal{S}^{(i)}) \right\| \leq 2G\sqrt{\frac{\eta}{\lambda}} + 4\mathbb{E}_{\mathcal{A}}\|\mathbf{w}_t - \mathbf{w}'_t\| + \frac{8G(1 - \eta\lambda)}{\lambda n}$$

$$\leq (8eG + 2G)\sqrt{\frac{\eta}{\lambda}} + \frac{8eG}{\lambda m} + \frac{8G}{\lambda n}$$

Setting $\gamma \leq \frac{1}{\lambda T}$. We note that for each $r_t$ has variance smaller than $\frac{1}{m}$. Define random variable $x_t := (1 + \frac{1}{T})^{t-1} r_t$. We have

$$|x_t - \mathbb{E}[x_t]| = (1 + \frac{1}{T})^{t-1} (r_t - \mathbb{E}[r_t]) \leq e \, |x_t - \mathbb{E}[x_t]| \leq (1 + \frac{1}{T})^{t-1} \left(1 - \frac{1}{m}\right) \leq e$$

$$\sum_{t=1}^{T} \mathbb{E}\,(x_i - \mathbb{E}[x_t])^2 \leq \sum_{t=1}^{T} (1 + \frac{1}{T})^{2t-2} \frac{1}{m} \left(1 - \frac{1}{m}\right) < \frac{Te^2}{m}$$

Hence by Bennett's inequality Theorem E.1, we have

$$\mathrm{P}\left[\sum_{t=1}^{T} (1 + \frac{1}{T})^{t-1} r_t \geq \frac{1}{m} \sum_{t=1}^{T} (1 + \frac{1}{T})^{t-1}\right] \leq \exp\left(-\frac{T^2 e^2}{m^2}\right).$$

Therefore, with probability at least $1 - \exp\left(-T^2 e^2 / m^2\right)$, we have

$$\left\|\mathrm{w}_{T+1} - \mathrm{w}'_{T+1}\right\| \leq 2G\sqrt{\frac{\eta}{\lambda}}(1 + \frac{1}{T})^T + \frac{2G}{\lambda m T}\sum_{t=1}^{T}\left(1 + \frac{1}{T}\right)^{t-1} \leq 2eG\sqrt{\frac{\eta}{\lambda}} + \frac{2eG}{\lambda m}$$

and therefore with probability at least $1 - \exp\left(-T^2 e^2 / m^2\right)$, we have

$$\forall k \in [K-1], \left\|\mathrm{u}^{(K+1)}(\mathrm{w}_{T+1}, \mathcal{S}) - \mathrm{u}^{(K+1)}(\mathrm{w}'_{T+1}, \mathcal{S}^{(i)})\right\| \leq 2G\sqrt{\frac{\eta}{\lambda}} + 4\left\|\mathrm{w}_t - \mathrm{w}'_t\right\| + \frac{8G(1 - \eta\lambda)}{\lambda n}$$

$$\leq (8eG + 2G)\sqrt{\frac{\eta}{\lambda}} + \frac{8eG}{\lambda m} + \frac{8G}{\lambda n}$$

By triangle inequality, we have with probability at least $1 - \exp\left(-T^2 e^2 / m^2\right)$,

$$\left\|\frac{1}{K}\sum_{k=1}^{K} \mathrm{u}^{(k)}(\mathrm{w}_{T+1}, \mathcal{S}) - \frac{1}{K}\sum_{k=1}^{K} \mathrm{u}^{(k)}(\mathrm{w}'_{T+1}, \mathcal{S}^{(i)})\right\| \leq (8eG + 2G)\sqrt{\frac{\eta}{\lambda}} + \frac{8eG}{\lambda m} + \frac{8G}{\lambda n}$$

$\square$

**Proposition 5.2.** Given a loss function $\ell(\cdot, \mathrm{z})$ and its adversarial counterpart $\tilde{\ell}(\cdot, \mathrm{z})$, the following holds: (1) If $\ell$ is $G$-Lipschitz (in its first argument), then $\tilde{\ell}$ is $G$-Lipschitz. (2) $\tilde{\ell}$ is **not** $H$-smooth even if $\ell$ is $H$-smooth. (3) If $\ell$ is $H$-smooth in $\mathrm{w}$, then $\tilde{\ell}$ is $H$-weakly convex in $\mathrm{w}$.

*Proof of Proposition 5.2.* Given $\mathrm{w}_1, \mathrm{w}_2$, define

$$\tilde{\mathrm{z}}_1 \in \underset{\tilde{\mathrm{z}} \in \mathcal{B}(\mathrm{z})}{\operatorname{argmax}} \ell(\mathrm{w}_1, \tilde{\mathrm{z}})$$

$$\tilde{\mathrm{z}}_2 \in \underset{\tilde{\mathrm{z}} \in \mathcal{B}(\mathrm{z})}{\operatorname{argmax}} \ell(\mathrm{w}_2, \tilde{\mathrm{z}}).$$

For the first item, it holds as

$$\left\|\tilde{\ell}(\mathrm{w}_1, \mathrm{z}) - \tilde{\ell}(\mathrm{w}_2, \mathrm{z})\right\| = \|\ell(\mathrm{w}_1, \tilde{\mathrm{z}}_1) - \ell(\mathrm{w}_2, \tilde{\mathrm{z}}_2)\|$$

$$= \max\left\{|\ell(\mathrm{w}_1, \tilde{\mathrm{z}}_1) - \ell(\mathrm{w}_2, \tilde{\mathrm{z}}_1)|, |\ell(\mathrm{w}_1, \tilde{\mathrm{z}}_2) - \ell(\mathrm{w}_2, \tilde{\mathrm{z}}_2)|\right\}$$

$$\leq G\|\mathrm{w}_1 - \mathrm{w}_2\|.$$

For the second item, the non-smoothness of the adversarial loss has been verified in Xing et al. [2021], Xiao et al. [2022]. For the third item, $\ell(\mathrm{w}, \mathrm{z})$ is $H$-smooth implies that $\ell(\mathrm{w}, \mathrm{z})$ is $H$-weakly convex, and further derive that $\tilde{\ell}(\mathrm{w}, \mathrm{z})$ is $H$-weakly convex because

$$\tilde{\ell}(\mathrm{w}_2, \mathrm{z}) = \ell(\mathrm{w}_2, \tilde{\mathrm{z}}_2)$$

$$\geq \ell(\mathrm{w}_2, \tilde{\mathrm{z}}_1) \qquad\qquad \text{(By definition of } \tilde{\mathrm{z}}_1, \tilde{\mathrm{z}}_2)$$

$$\geq \ell(\mathrm{w}_1, \tilde{\mathrm{z}}_1) + \langle g(\mathrm{w}_1, \tilde{\mathrm{z}}_1), \mathrm{w}_2 - \mathrm{w}_1 \rangle - \frac{\rho}{2}\|\mathrm{w}_2 - \mathrm{w}_1\|^2$$

$$\qquad\qquad (g(\mathrm{w}_1, \tilde{\mathrm{z}}_1) \in \partial\ell(\mathrm{w}_2, \tilde{\mathrm{z}}_1), \text{ apply Proposition D.1})$$

$$= \tilde{\ell}(\mathrm{w}_1, \mathrm{z}) + \langle \tilde{g}(\mathrm{w}_1, \mathrm{z}), \mathrm{w}_2 - \mathrm{w}_1 \rangle - \frac{\rho}{2}\|\mathrm{w}_2 - \mathrm{w}_1\|^2 \quad \text{(Redefine } \tilde{g}(\mathrm{w}_1, \mathrm{z}) \in \partial\tilde{\ell}(\mathrm{w}_1, \mathrm{z}))$$

$\square$

