# OpenReview forum: "On the Stability and Generalization of Meta-Learning"
_NeurIPS.cc/2024/Conference — NeurIPS 2024 poster_

### Official Review · Reviewer_zAce · 2024-06-15

**Soundness:** 2
**Presentation:** 3
**Contribution:** 3
**Rating:** 4
**Confidence:** 4

**Summary:**

The paper presents stability analysis for meta-learning. The paper first introduces a uniform meta-stability, where there is both a change of the task in the meta-sample and also a change of an example for the task at test time. For this uniform meta-stability, the paper gives high-probability bounds of the order of $\beta+O(1/\sqrt{mn})$, where $\beta$ is the stability parameter, m is the number of tasks and n is the number of examples per task. The paper then considers a prox meta-learning algorithm with two task-specific algorithms. For these algorithms, the paper establishes stability bounds for different problem settings: convex & Lipschitz, convex & smooth and weakly-convex problems. Applications to proximal-meta learning with stochastic optimization and robust adversarial proximal meta-learning are also given.

**Strengths:**

The paper introduces a new stability probability measure for meta-learning, and give both high-probability bounds and bounds in expectation for algorithms satisfying this uniform meat-stability.

The paper also studies two algorithms and establishes stability bounds under different problem settings. These bounds better than existing bounds in the sense of involving $1/\sqrt{mn}$, which shows the benefit of considering several tasks together.

The paper is clearly written and the main contributions are clearly stated.

**Weaknesses:**

In Theorem 4.4, the paper establishes risk bounds of the order of $\frac{1}{\lambda m}+\frac{1}{\lambda n}+\sqrt{1/(mn)}$. However, $\lambda$ should be often small in practice. For example, Theorem 4.7 gives bounds depending on $\lambda\sigma^2$. To get meaningful bounds, $\lambda$ should be of the order $o(1)$. At the end of Section 4.3, the paper suggests $\lambda=O(1/\sqrt{n})$. In this case, the excess risk bounds in Theorem 4.4 are of the order of $\sqrt{n}/m+1/\sqrt{n}$. Then, the term $\sqrt{1/(mn)}$ does not play an important role in the generalization. Also, Theorem 4.1 requires $\lambda\geq H$ in the smooth case, which requires a large regularization parameter.

In Theorem 4.6, the risk bounds involve $\sqrt{\eta/\lambda}$. Therefore, the bounds would be nonvacuous only if $\eta$ is small and $\lambda$ is large. However, a small $\eta$ would affect the convergence of the algorithm. Therefore, it may be difficult to find a balance between stability and optimization.

In the proof of Theorem B.3, the paper uses the identity (I omit the index $j,i$ for brevity)
$$
g^{0,0}-g^{r,k}=\sum_{g=0}^{r-1}\sum_{l=0}^{k-1}(g^{q,l}-g^{q+1,l+1}).
$$
However, it seems that this identity does not hold. For example, let us consider the simple case $r=2,k=2$. Then the identity becomes
$$
g^{0,0} - g^{2,2} = g^{0,0} - g^{1,1} + g^{0,1}- g^{1,2} + g^{1,0} - g^{2,1} + g^{1,1} - g^{2,2}.
$$
It is clear that this identity does not hold. Therefore, it seems that the proof of Theorem B.3 is not rigorous.

**Questions:**

Is the proof of Theorem B.3 correct?

In Theorem 4.7, the paper uses $\sigma^2=\frac{1}{K}\sum_j\|\hat{w}-u_j^*\|$. It seems that $K$ should be $m$?

Typos:
- Below Theorem 4.4: "should not surprising"

**Limitations:**

No potential negative societal impacts.

---

> ### Author Rebuttal · Authors · 2024-08-05
>
> **Choice of $\lambda$.**
>
> $\lambda=O(1/\sqrt{n})$ is just one choice of $\lambda$ that leads to non-vacuous excess risk and there are other options. For example, if choosing $\lambda=O(1/n^{\frac14})$, then under the setting where $\sqrt{n}\leq m \leq n^{3/2}$, the generalization gap from Theorem 4.4 is of rate $O(\frac{1}{\lambda m}+\frac{1}{\lambda n} + \frac{1}{\sqrt{mn}})=O(\frac{n^{\frac14}}{m}+\frac{1}{n^{\frac34}}+\frac{1}{\sqrt{mn}})=O(\max(\frac{n^{\frac14}}{m},\frac{1}{n^{\frac34}}))$. This is tighter than the generalization gap that derived from Theorem 2.2, which is of rate $O(\frac{1}{\lambda m}+\frac{1}{\lambda n} + \frac{1}{\sqrt{m}})=O(\frac{n^{\frac14}}{m}+\frac{1}{n^{\frac34}}+\frac{1}{\sqrt{m}})=O(\frac{1}{\sqrt{m}})$.
>
> Theorem 4.1 requires $\lambda>H$ for smooth loss under Option 1 for Algorithm 2. However, Theorem 4.7 only considers Option 2 for Algorithm 2. The corresponding theorem for the generalization gap in this case is Theorem 4.4, where we only assume $\lambda>0$.
>
> **Balance between stability and optimization.**
>
> We discussed the proper choice of $\eta=O(\frac{1}{\lambda K^{2/3}})$ for convex and Lipschitz losses to achieve a decaying excess risk, as described in lines 320-323. With sufficiently large $K, T$, by appropriately choosing $\lambda=o(1)$, the excess risk remains non-vacuous.
>
> **Proof of Theorem B.3.**
>
> We thank the reviewer for carefully checking our proof. For given $r=k=2$, the right decomposition would be $g^{0,0}-g^{2,2}=g^{0,0}-g^{1,0}+g^{1,0}-g^{2,0}+g^{2,0}-g^{2,1}+g^{2,1}-g^{2,2}$.
>
> We now fix the equation the reviewer point it out as well as some other typos as follows. We note that proof of Theorem B.3 is an extension of  [Bousquet et al., 2020, Theorem 4], and the following fix would not change our main result.
>
> Continue with line 689 in the paper, we have the following
>
> $g_{j,i}-\mathbb{E}[g_{j,i}|Z_j,z_i]=\sum_{q=0}^{r-1}g_{j,i}^{q,0} - g_{j,i}^{q+1,0}+\sum_{l=0}^{k-1} g_{j,i}^{r,l} - g_{j,i}^{r,l+1},$
>
> and the total sum of interest satisfies by the triangle inequality
> $||\sum_{j=1}^m\sum_{i=1}^n g_{j,i}||\leq ||\sum_{j=1}^m\sum_{i=1}^n \mathbb{E}[g_{j,i}|Z_j,z_i]||+\sum_{q=0}^{r-1}||\sum_{j=1}^m\sum_{i=1}^n g_{j,i}^{q,0}-g_{j,i}^{q+1,0}||
> +\sum_{l=0}^{k-1}||\sum_{j=1}^m\sum_{i=1}^n g_{j,i}^{r,l}-g_{j,i}^{r,l+1}||.$
>
> Apply McDiarmid’s inequality gives us that
> $||g_{j,i}^{q,0}-g_{j,i}^{q+1,0}||(Z_j,Z_{[m]\backslash E^{q+1}(j)},z_i,z_{[n]\backslash C^{0}(i)})\leq 2\sqrt{2^{q+1}}\bar\beta$,
>
> $||g_{j,i}^{r,l}-g_{j,i}^{r,l+1}||(Z_j,Z_{[m]\backslash E^{r}(j)},z_i,z_{[n]\backslash C^{l+1}(i)})\leq 2\sqrt{2^{l+1}}\bar\beta$.
>
>
> Since $g_{j,i}^{q,0}-g_{j,i}^{q+1,0}$ for $j\in E^q, i\in C^0$ depends on $Z_j$, $Z_{[m]\backslash E^{q+1}(j)}$, $z_i$, $z_{[n]\backslash C^0(i)}$,
> the terms are independent and zero mean conditioned on $Z_{[m]\backslash E^{q+1}(j)}$.
>
> Applying Theorem B.2, we have
> \begin{align*}
> ||\sum_{j\in E^q}\sum_{i\in C^0} g_{j,i}^{q,0} - g_{j,i}^{q+1,0}||^2(Z_{[m]\backslash E^q})
> \leq 36\cdot 2^{q} \frac{1}{2^q} \sum_{j\in E^q}\sum_{i\in C^0}||g_{j,i}^{q,0}-g_{j,i}^{q+1,0}||^2(Z_{[m]\backslash E^q})
> \end{align*}
>
>
> Integrating with respect to $(Z_{[m]\backslash E^q})$ and using $||g_{j,i}^{q,0} - g_{j,i}^{q+1,0}||\leq 2\sqrt{2^{q+1}}\bar\beta$, we have
> \begin{align*}
> ||\sum_{j\in E^q}\sum_{i\in C^0} g_{j,i}^{q,0}-g_{j,i}^{q+1,0}||
> \leq 6\sqrt{2^{q}}\times 2\sqrt{ 2^{q+1}}\bar\beta= 12\sqrt{2}\cdot2^{q}\bar\beta.
> \end{align*}
>
>
> Applying triangle inequality over all sets $C^0\in\mathcal{C_0},E^q\in\mathcal{E_q}$ gives us that
>
> \begin{align*}
> ||\sum_{j\in[m]}\sum_{i\in[n]} g_{j,i}^{q,0}-g_{j,i}^{q+1,0}||
> \leq \sum_{E^q\in\mathcal{E_q},C^0\in\mathcal{C_0}}||\sum_{j\in E^q, i\in C^0}g_{j,i}^{q,0} - g_{j,i}^{q+1,0}||
> \leq12\sqrt{2}\cdot 2^{r+k}\bar\beta.
> \end{align*}
>
> Similarly, applying triangle inequality over all sets $C^l\in\mathcal{C}_l,E^r\in\mathcal{E}_r$ gives us that
>
> \begin{align*}
> ||\sum_{j\in[m]}\sum_{i\in[n]} g_{j,i}^{r,l}-g_{j,i}^{r,l+1}||
> \leq \sum_{E^r\in\mathcal{E_r},C^l\in\mathcal{C_l}}||\sum_{j\in E^r, i\in C^l}g_{j,i}^{r,l} - g_{j,i}^{r,l+1}||\leq12\sqrt{2}\cdot 2^{r+k}\bar\beta.
> \end{align*}
>
> Recall that $2^k < 2n, 2^r<2m$ due to the possible extension of the sample. Therefore we have
> \begin{align*}
> \sum_{q=0}^{r-1}||\sum_{j=1}^m\sum_{i=1}^n g_{j,i}^{q,0}-g_{j,i}^{q+1,0}||
> +\sum_{l=0}^{k-1}||\sum_{j=1}^m\sum_{i=1}^n g_{j,i}^{r,l}-g_{j,i}^{r,l+1}||
> \lesssim mn\bar\beta \log(mn).
> \end{align*}
>
> [Bousquet et al., 2020] Bousquet, Olivier, Yegor Klochkov, and Nikita Zhivotovskiy. "Sharper bounds for uniformly stable algorithms." Conference on Learning Theory. PMLR, 2020.

---

> > ### Comment · Reviewer_zAce · 2024-08-12
> > **Thank you for the response**
> >
> > I thank the authors for their responses. My concerns on the proof of Theorem B.3 are addressed. However, I still think the choice of $\lambda$ is a bit restrictive. As stated in the paper, a typical choice of $\lambda$ is $\lambda=O(1/\sqrt{n})$, which would be small. Lemma 4.1 and Corollary 5.3 require $\lambda\geq H$, Lemma 4.5, Lemma 4.6 and Lemma 5.1 require $\lambda\geq 2\rho$. In this case, both $H$ and $\rho$ should be very small, which are restrictive.
> >
> > Furthermore, the analysis requires very large $K$ to get good bounds. Since the total complexity is $TmK$, a large $K$ would make the algorithm computationally expensive.

---

> ### Author Response · Authors · 2024-08-12
>
> Thanks for the follow up discussion.
>
> ### Choice of $\lambda$
> Choice of $\lambda$ as $1/\sqrt{n}$ or any inverse proportion to the sample size is standard in statistical learning theory. This is because as the sample size increases, it is necessary to decrease the penalty/regularization term accordingly. Intuitively, if you have infinite data, you do not need any prior knowledge and minimizing only the empirical risk is good enough for learning.
>
> In the discussion of expected excess risk (lines 320-329), we have mentioned that Theorem 4.7 only considers Option 2 for Algorithm 2 (GD) in our previous response. Therefore, for convex, Lipschitz, and smooth losses, the corresponding theorem for the generalization gap in this case is Theorem 4.4, where we only assume $\lambda>0$. Similarly, for convex and non-smooth losses, we apply Theorem 4.6 (where $\rho=0$ for convex losses), and we also only assume $\lambda>0$ in that setting. The assumptions that the reviewers point it out such as $\lambda>H$ and $\lambda>2\rho$ with $\rho>0$ are used in deriving generalization gap in various setting, they **do not appear in the discussion of the expected excess risk**.
>
> ### Size of K
> Large K makes sense in many settings such as distributed learning and federated learning. Also, we do not have the luxury to tweak theory as we please. It is what it is. The point of theory is to inform practice. So, the takeaway here is that **IN THE WORST CASE**, if nothing else is helping in practice, then K should be increased. This is why such results are important.
>
> It is also not a bad idea to choose $K=n$. We have such a discussion with reviewer S84Y (please see https://openreview.net/forum?id=J8rOw29df2&noteId=PyRch0218g) where we compare our result with an existing work [1] under reasonable choice of parameters. We encourage the reviewer to take a look at our discussion.
>
> Please let us know if you have more questions.
>
> [1] Giulia Denevi, Carlo Ciliberto, Riccardo Grazzi, and Massimiliano Pontil. Learning-to-learn stochastic gradient descent with biased regularization. In International Conference on Machine Learning, pages 1566–1575. PMLR, 2019a.

---

> > ### Author Response · Authors · 2024-08-13
> > **Open to discussion**
> >
> > Dear reviewer zAce,
> >
> > We thank the reviewer for their careful reading of the paper and have discussion with us. As the end of discussion period is approaching, please let us know whether we address your concerns. We are happy to have further discussion if you have any remaining/follow-up questions.
> >
> > Best,
> > Authors

---

### Official Review · Reviewer_VSfZ · 2024-07-07

**Soundness:** 3
**Presentation:** 3
**Contribution:** 2
**Rating:** 5
**Confidence:** 3

**Summary:**

This paper introduces the notion of "uniform meta-stability" to bound the generalization error of $\ell_2$-regularized meta-learning problem. Theoretical guarantees are respectively established for smooth loss functions as well as weakly convex losses that are not necessarily smooth. Variants of the algorithm on stochastic and adversarially robust meta-learning problems are investigated.

**Strengths:**

1. This paper provides a solid theoretical analysis regarding the stability and generalization of meta-learning.
2. Application of the theoretical results to stochastic and adversarially robust meta-learning is both interesting and promising.

However, I have several concerns on the applicable scope and technical assumptions of this paper.

**Weaknesses:**

Major concerns:

1. The scope of this paper can be quite limited. In particular, this paper focuses exclusively on the specialized meta-learning formulation where i) an explicit $\ell_2$-regularization is leveraged to encode the prior, and ii) the inner-level and outer-level rely on the same dataset. These two particular structures are intentionally designed in [1] to simplify the computation of meta-gradient. In contrast, most widely-used meta-learning algorithms like MAML employ i) GD-based implicit regularization [2] or a generic explicit regularizer beyond $\ell_2$, and ii) the support/query (S/Q) training setup. Unfortunately, it seems that the results of this paper are merely applicable to the problem addressed in [1].
2. Some assumptions made in this paper are rather strong. For instance, it is assumed in line 127 that $\mathcal{W}$ is closed with a finite radius (i.e., compact). This compactness premise is uncommon in meta-learning. It is not made in not only the most related work [3], but also [1] which this paper builds upon. In addition, while [3] provided an analysis for Holder smooth loss, this paper requires it to be weakly convex, which is a stronger assumption.

Minor comments:
1. In line 141, the symbol $\mathcal{H}$ comes out of the blue without definition, which I guess might be the hypothesis space.
2. In line 242, "this should not surprising" should be corrected to "this should not be surprising".

[1] P. Zhou, X. Yuan, H. Xu, S. Yan, and J. Feng, "Efficient Meta-Learning via Minibatch Proximal Update," NeurIPS 2019.
[2] E. Grant, C. Finn, S. Levine, T. Darrell, and T. Griffiths, "Recasting Gradient-Based Meta-Learning as hierarchical Bayes," ICLR 2018.
[3] J. Guan, Y. Liu, and Z. Lu, "Fine-grained analysis of stability and generalization for modern meta learning algorithms,", NeurIPS 2022.

**Questions:**

See Weaknesses.

**Limitations:**

There is no explicit discussion regarding the limitations of this work. Social impact is not applicable.

---

> ### Author Rebuttal · Authors · 2024-08-05
>
> **Limited Scope.** Apply l2 regularization is common in meta-learning literature [1,4,5,6,7,8] and transfer learning literature in general [9,10].
> Extending our idea to MAML and S/Q learning could be interesting future direction.
>
> **Compact radius assumption.** The compact radius assumption is only necessary for obtaining the excess transfer risk, as discussed in Theorem 4.7 in Section 4.3. This is a standard assumption in the optimization literature and has been utilized in previous work that provided excess transfer risk results; see, for example, Assumption 1 and Algorithm 1 in [4]. We emphasize that the generalization gap based on the stability argument (Sections 4.1 and 4.2) does not rely on such assumptions, which is consistent with prior work [1,3] as they do not consider excess transfer risk.
>
> **Weakly-convexity assumption.** A weakly convex function is essentially non-convex and non-smooth but with bounded lower curvature. This is a milder assumption than requiring the loss to be smooth and is achievable in practical scenarios, such as training neural networks using gradient descent [11]. In contrast, [3] considers Hölder smooth loss, which is another way of relaxing the smoothness assumption, and it appears to be limited to linear classifiers based on the examples provided in [3]. Moreover, we have discussed the limitations of [3] as it assumes that the loss function after gradient update $\hat R(\cdot, S)$ is convex or Hölder smooth (see lines 284-288).
>
>
>
> [1] P. Zhou, X. Yuan, H. Xu, S. Yan, and J. Feng, "Efficient Meta-Learning via Minibatch Proximal Update," NeurIPS 2019.
>
> [2] E. Grant, C. Finn, S. Levine, T. Darrell, and T. Griffiths, "Recasting Gradient-Based Meta-Learning as hierarchical Bayes," ICLR 2018.
>
> [3] J. Guan, Y. Liu, and Z. Lu, "Fine-grained analysis of stability and generalization for modern meta learning algorithms,", NeurIPS 2022.
>
> [4] Giulia Denevi, Carlo Ciliberto, Riccardo Grazzi, and Massimiliano Pontil. "Learning-to-learn stochastic gradient descent with biased regularization." In International Conference on Machine Learning, 2019.
>
> [5] Zhou, Xinyu, and Raef Bassily. "Task-level differentially private meta learning." Advances in Neural Information Processing Systems, 2022.
>
> [6] Jiang, Weisen, James Kwok, and Yu Zhang. "Effective meta-regularization by kernelized proximal regularization." Advances in Neural Information Processing Systems, 2021.
>
> [7] Balcan, Maria-Florina, Mikhail Khodak, and Ameet Talwalkar. "Provable guarantees for gradient-based meta-learning." International Conference on Machine Learning, 2019.
>
> [8] Rajeswaran, Aravind, et al. "Meta-learning with implicit gradients." Advances in neural information processing systems, 2019.
>
> [9] Kuzborskij, Ilja, and Francesco Orabona. "Stability and hypothesis transfer learning." International Conference on Machine Learning, 2013.
>
> [10] Kuzborskij, Ilja, and Francesco Orabona. "Fast rates by transferring from auxiliary hypotheses." Machine Learning, 2017.
>
> [11] Richards, Dominic, and Ilja Kuzborskij. "Stability & generalisation of gradient descent for shallow neural networks without the neural tangent kernel." Advances in neural information processing systems, 2021.

---

> > ### Comment · Reviewer_VSfZ · 2024-08-11
> >
> > Thank you for your response. My concerns regarding the theoretical aspects of this work have been addressed. However, I still believe the scope of the paper is rather limited due to the use of non-S/Q setups. This setup is uncommon in meta-learning where the goal is to achieve better and faster generalization to new tasks. Could you please provide examples beyond [1] that adopt Eq. (1) (i.e., the same dataset for two levels, plus $\ell_2$ regularization) as their meta-learning objectives?

---

> > > ### Author Response · Authors · 2024-08-12
> > >
> > > Thanks for the follow up discussion.
> > >
> > > We would like to clarify that our goal is indeed to achieve strong generalization guarantees on **new, unseen tasks, rather than the seen tasks**. This is based on the assumption that all tasks are drawn from an unknown task distribution, which is a common assumption in meta-learning literature [2,3,4,5,6,7].
> > >
> > > In our previous response, we have listed a number of references that considered $\ell_2$ regularization, which is also commonly studied in meta-learning as **meta-regularization**. If the reviewer is interested in additional work beyond [1], please refer to [8] (the online version of [1]) for excess risk analysis in the meta-learning setting for linear classifiers; [9] also examined a similar online algorithm with a focus on establishing connections between meta initialization and meta regularization. Furthermore, [10] considered the same meta-learning setting as [1] with a focus on obtaining differential privacy guarantees, while [11] investigated the same formulation as eq (1) with the application on federated learning.
> > >
> > > We hope we have provided sufficient evidence to demonstrate that the setting our work considered is indeed studied by others and worth investigating in different scenarios. Please let us know if you have more questions.
> > >
> > > [1] P. Zhou, X. Yuan, H. Xu, S. Yan, and J. Feng, "Efficient Meta-Learning via Minibatch Proximal Update," NeurIPS 2019.
> > >
> > > [2] Baxter, Jonathan. "A model of inductive bias learning." Journal of artificial intelligence research 12 (2000): 149-198.
> > >
> > > [3] Andreas Maurer. 'Algorithmic stability and meta-learning'. Journal of Machine Learning Research, 2005.
> > >
> > > [4] Chen, Jiaxin, et al. "A closer look at the training strategy for modern meta-learning." Advances in neural information processing systems 33 (2020): 396-406.
> > >
> > > [5] Pentina, Anastasia, and Christoph Lampert. "A PAC-Bayesian bound for lifelong learning." International Conference on Machine Learning. PMLR, 2014.
> > >
> > > [6] Guan, Jiechao, Yong Liu, and Zhiwu Lu. "Fine-grained analysis of stability and generalization for modern meta learning algorithms." Advances in Neural Information Processing Systems 35 (2022): 18487-18500.
> > >
> > > [7] Finn, Chelsea, et al. "Online meta-learning." International conference on machine learning. PMLR, 2019.
> > >
> > > [8] Giulia Denevi, Carlo Ciliberto, Riccardo Grazzi, and Massimiliano Pontil. "Learning-to-learn stochastic gradient descent with biased regularization." In International Conference on Machine Learning, 2019.
> > >
> > > [9] Balcan, Maria-Florina, Mikhail Khodak, and Ameet Talwalkar. "Provable guarantees for gradient-based meta-learning." International Conference on Machine Learning, 2019.
> > >
> > > [10] Zhou, Xinyu, and Raef Bassily. "Task-level differentially private meta learning." Advances in Neural Information Processing Systems, 2022.
> > >
> > > [11] T Dinh, Canh, Nguyen Tran, and Josh Nguyen. "Personalized federated learning with moreau envelopes." Advances in neural information processing systems 33 (2020): 21394-21405.

---

> > > > ### Comment · Reviewer_VSfZ · 2024-08-12
> > > >
> > > > Thanks for the further elaborations. I have no further questions and I will raise my score.

---

### Official Review · Reviewer_SWCq · 2024-07-12

**Soundness:** 3
**Presentation:** 3
**Contribution:** 3
**Rating:** 6
**Confidence:** 3

**Summary:**

This submission introduces a new bound on the transfer risk of meta-learning based on a modified form of algorithmic stability. Several examples of how the new bound can be used to analyze the transfer risk of meta-learning algorithms built on gradient-based optimisation are provided. A comparison with the existing bound of Maurer (2005) is given, where it is shown that the new bound is generally better.

**Strengths:**

* The submission introduces a new bound on the transfer risk of meta-learning based on a modified form of algorithmic stability. The new stability definition seems intuitive, and enables the bound to have a much better dependence on the sampling error compared to the conventional meta-learning bound based on algorithmic stability.
* They provide several examples of how the new bound can be used to analyze the transfer risk of meta-learning algorithms built on gradient-based optimisation. Several combinations of loss function properties (convex, weakly convex, Lipschitz, smooth, etc) are considered.
* The submission also considers some small extensions to stochastic and adversarial settings.

**Weaknesses:**

* The proofs can be a bit hard to follow at times. For example, the construction of $\mathcal{C}_l$ is the proof of Theorem B.3 is somewhat underspecified, and I could not really follow what was going on here.
* When analysing specific algorithms with the new framework, it would have been good if the submission also contained a comparison with papers that also focus on, e.g., convex smooth meta-learning settings. The work of Giulia Denevi, Massimiliamo Pontil, and others could be relevant here.
* The submission could be improved by providing a more detailed discussion of the implications of the new bound, and the fundamental high-level reasons for the differences. For example, the main difference that I can see is improved dependence on the sampling error compared with Maurer (2005) [Note: Tommi Jaakkola is erroneously listed as a co-author in this submission]. A high-level overview of the proof strategy and any other benefits that I missed would be good.

**Questions:**

N/A

**Limitations:**

There is no substantial discussion of the limitations, but this might be because there is no downside over the existing bound. The authors do provide some examples of what is still generally missing from meta-learning theory.

---

> ### Author Rebuttal · Authors · 2024-08-05
>
> **Proof idea.** Our proof leverages the same sample-splitting approaches as described in [1,2]. The recursive structure is based on the specific telescoping sum. The design of the sequence of partitions $\mathcal{C_0},\ldots,\mathcal{C_k}$ relates to the analysis of the terms $g_{i,j}^{q,0}-g_{i,j}^{q+1,0}$, and the sequence of partitions $\mathcal{E_0},\ldots,\mathcal{E_r}$ relates to the analysis of the terms $g_{i,j}^{r,l}-g_{i,j}^{r,l+1}$. Please refer to the answer to Reviewer VSfZ for a detailed discussion on the proof.
>
> **Comparison with literature.**
> We discuss some of the related work in page 7 line 276-300. We now provide a detailed comparison between our result and [3].
> - **Different Function Classes Considered.** The function classes considered in [A] are limited to compositions of linear hypothesis classes with convex and closed losses. In contrast, our work considers a broader range of functions, encompassing not only convex, Lipschitz, and smooth functions but also weakly-convex and non-smooth functions.
>
> - **Different Analysis Techniques.** [A] employs a primal-dual formulation of bias regulation ERM due to the simplicity of linear classifier that discussed above. Given our broader range of function classes, we introduce a new definition of stability and use stability arguments to derive generalization bounds.
>
> - **The expected excess risk is different.** The expected excess risk from Theorem 4.7 in our paper is of the form $O(\frac{1}{\lambda\sqrt{K}}+\frac{1}{\lambda m}+\frac{1}{\lambda n}+\frac{\lambda}{T}+\lambda\sigma^2),$
> where $m$ is the number of tasks, $n$ is the number of samples per task, $K$ is the number of iterations for task-specific algorithm 2 and $T$ is the number of iterations for meta-learning algorithm 1,
> $\sigma$ is the approximate error that captures the average distance between the optimal task-specific parameters $\mathrm{u}_j$'s and the optimal estimated meta-parameter $\mathrm{\hat w}$.
> Moreover, choosing $K=O(n^2)$ and $\lambda=O(\frac{1}{\sqrt{n}})$, the expected excess risk is
> $O(\frac{\sqrt{n}}{m}+\frac{1}{\sqrt{n}}+\frac{1}{T\sqrt{n}}+\frac{\sigma^2}{\sqrt{n}}),$
> which depends on the number of tasks $m$.
> On the other hand, Algorithm 2 in [A] assumes solving the within-task problem approximately, so there's no within task iteration $K$ involved in the bound. The expected excess risk in [A] is of the form $O(\frac{Var_m}{\sqrt{n}}+\frac{1}{\sqrt{T}}),$ where $Var_m$ captures the relatedness among the tasks sampled from the task environment. Note that this bound is gained based on a specific choise of $\lambda$ that depends on $Var_m$, which is unlikely to know ahead in practice.
> Moreover, assuming $Var_m$ is a constant, then this bound is independent of the number of tasks $m$. Therefore, our bound is tighter when $n\lesssim m$.
>
> **Comparison with [Maurer 2005].**
> We present the main result from [B] as Theorem 2.1 in Sec 2. The advantage of [B] is that it directly leverages the uniform stability argument from single-task learning to meta-learning. [B] considers uniform stability definitions at both the task-sample and meta-sample levels. By applying new tools from [r3], we provide Theorem 2.2, which improves upon Theorem 2.1 under the same definition of uniform stability.
> On the other hand, our paper introduces a new notion of uniform stability that accounts for changes at both the task-sample and meta-sample levels, specifically designed for meta-learning, and provides the corresponding generalization gap, as shown in Theorem 3.1. In each specific setting (convex+Lipschitz, convex+smooth, weakly-convex+non-smooth), we primarily compare our results (derived from Thm 3.1) with those derived from Thm 2.2. For further details, see lines 235-238 and 269-271.
>
> [1] Feldman, Vitaly, and Jan Vondrak. "Generalization bounds for uniformly stable algorithms." Advances in Neural Information Processing Systems, 2018.
>
> [2] Bousquet, Olivier, Yegor Klochkov, and Nikita Zhivotovskiy. "Sharper bounds for uniformly stable algorithms." Conference on Learning Theory, 2020.
>
> [3] Giulia Denevi, Carlo Ciliberto, Riccardo Grazzi, and Massimiliano Pontil. Learning-to-learn stochastic gradient descent with biased regularization. In International Conference on Machine Learning, 2019.

---

> > ### Author Response · Authors · 2024-08-13
> > **Open to discussion**
> >
> > Dear reviewer SWCq,
> >
> > We thank the reviewer for their careful reading of the paper. As the end of discussion period is approaching, please let us know whether we address your concerns. We are happy to have further discussion if you have any remaining/follow-up questions.
> >
> > Best,
> > Authors

---

### Official Review · Reviewer_S84Y · 2024-07-12

**Soundness:** 3
**Presentation:** 3
**Contribution:** 2
**Rating:** 6
**Confidence:** 3

**Summary:**

The authors introduce a new notion of stability for meta-learning algorithms and they show how it is possible to bound their generalization gap by their stability property. The new definition of stability measures the sensitivity of the learning algorithm as one replaces both a task in the meta-sample as well as a single training example available for the task at test time. They consider two variants of within-task algorithms – based on regularized empirical risk minimization (RERM) and gradient descent (GD). For meta-learning they employ a gradient descent method. They apply their stability-based analysis to these variants to learning problems with convex, smooth losses and weakly convex, non-smooth losses. Finally, they adapt their results to stochastic and robust variants to inference-time adversarial attacks of the proposed meta-learning algorithms.

**Strengths:**

1. The introduction of a new meta-learning stability is innovative and alternative.

2. The effort of providing a theoretical analysis on the topic is valuable.

3. The topic is interesting for the venue.

**Weaknesses:**

1. The method is not new. It coincides with that in [A]

2. The comparison with the literature should be clearer. In particular, I would like to have more details and about the advantages and disadvantages w.r.t. the reference below [B].

[A] Giulia Denevi, Carlo Ciliberto, Riccardo Grazzi, and Massimiliano Pontil. Learning-to-learn stochastic gradient descent with biased regularization. In International Conference on Machine Learning, pages 1566–1575. PMLR, 2019a.

[B] Andreas Maurer and Tommi Jaakkola. 'Algorithmic stability and meta-learning'. Journal of Machine Learning Research, 6(6), 2005.

**Questions:**

1. The underline should be removed in the paper.

2. It would be nice to relate Thm. 3.1 to the question about the possibility to get meta-learning rates going as $1/(nm)$, with $n$ the number of within-task points and $m$ the number of tasks.

3. 'Indeed, we show that Theorem 2.2 yields a rate of $O( 1/m + 1/n + \sqrt{m})$, which is worse for all $n > m$.' Why? Moreover this is the setting less frequent in Meta-learning (where usually one assumes $n << m$).

4. In Thm. 3.2 I expected a term of the form $1/(\eta m) + 1/(\lambda n)$, instead of $1/(\lambda m) + 1/(\lambda n)$.

5. Please compare more in detail the bound in Thm. 4.7 on the excess risk with those in the reference [A] above.

**Limitations:**

I do not see any potential negative societal impact of this work.

---

> ### Author Rebuttal · Authors · 2024-08-05
>
> W1 / Q5: We never claim we provide a new meta-learning algorithm. Our algorithm is indeed of the same nature as in [r2] and [A]. We now provide a detailed comparison between our result and [A].
>
> - **Different Function Classes Considered.** The function classes considered in [A] are limited to compositions of linear hypothesis classes with convex and closed losses. In contrast, our work considers a broader range of functions, encompassing not only convex, Lipschitz, and smooth functions but also weakly-convex and non-smooth functions.
>
> - **Different Analysis Techniques.** [A] employs a primal-dual formulation of bias regulation ERM due to the simplicity of linear classifier that discussed above. Given our broader range of function classes, we introduce a new definition of stability and use stability arguments to derive generalization bounds.
>
> - **The expected excess risk is different.** The expected excess risk from Theorem 4.7 in our paper is of the form $O(\frac{1}{\lambda\sqrt{K}}+\frac{1}{\lambda m}+\frac{1}{\lambda n}+\frac{\lambda}{T}+\lambda\sigma^2),$
> where $m$ is the number of tasks, $n$ is the number of samples per task, $K$ is the number of iterations for task-specific algorithm 2 and $T$ is the number of iterations for meta-learning algorithm 1,
> $\sigma$ is the approximate error that captures the average distance between the optimal task-specific parameters $\mathrm{u}_j$'s and the optimal estimated meta-parameter $\mathrm{\hat w}$.
> Moreover, choosing $K=O(n^2)$ and $\lambda=O(\frac{1}{\sqrt{n}})$, the expected excess risk is
> $O(\frac{\sqrt{n}}{m}+\frac{1}{\sqrt{n}}+\frac{1}{T\sqrt{n}}+\frac{\sigma^2}{\sqrt{n}}),$
> which depends on the number of tasks $m$.
> On the other hand, Algorithm 2 in [A] assumes solving the within-task problem approximately, so there's no within task iteration $K$ involved in the bound. The expected excess risk in [A] is of the form $O(\frac{Var_m}{\sqrt{n}}+\frac{1}{\sqrt{T}}),$ where $Var_m$ captures the relatedness among the tasks sampled from the task environment. Note that this bound is gained based on a specific choise of $\lambda$ that depends on $Var_m$, which is unlikely to know ahead in practice.
> Moreover, assuming $Var_m$ is a constant, then this bound is independent of the number of tasks $m$. Therefore, our bound is tighter when $n\lesssim m$.
>
> W2: We present the main result from [B] as Theorem 2.1 in Sec 2. The advantage of [B] is that it directly leverages the uniform stability argument from single-task learning to meta-learning. [B] considers uniform stability definitions at both the task-sample and meta-sample levels. By applying new tools from [r3], we provide Theorem 2.2, which improves upon Theorem 2.1 under the same definition of uniform stability.
> On the other hand, our paper introduces a new notion of uniform stability that accounts for changes at both the task-sample and meta-sample levels, specifically designed for meta-learning, and provides the corresponding generalization gap, as shown in Theorem 3.1. In each specific setting (convex+Lipschitz, convex+smooth, weakly-convex+non-smooth), we primarily compare our results (derived from Thm 3.1) with those derived from Thm 2.2. For further details, see lines 235-238 and 269-271.
>
> Q2: To achieve a rate of $1/mn$, stronger assumptions may be necessary. Prior work has demonstrated this: [r1] obtained a rate of $1/mn$ for strongly convex functions with Lipschitz continuous Hessians, while [r4] presented a generalization bound of $O(\sqrt{\frac{C}{mn}})$ under the task-relatedness assumption, where $C$ accounts for the logarithm of the covering number of the hypothesis class. More recently, [r5] provided $O(\frac{1}{m}+\frac1n)$ fast rate generalization bounds under an additional extended Bernstein's condition. Exploring the possibility of obtaining faster rates by considering stronger conditions in our setting could be an interesting direction for future work.
>
> Q3: When $n>m$, the generalization gap derived from Theorem 3.1 is $O(\frac1m+\frac1n+\frac{1}{\sqrt{mn}})=O(\frac1m)$. The generalization gap derived from Theorem 2.2 is $O(\frac1m+\frac1n+\frac{1}{\sqrt{m}})=O(\frac{1}{\sqrt{m}})$, which is worse that the former.
> We note that when $n<m\leq n^2$, the generalization gap derived from Theorem 3.1 can be simplified as $O(\frac1n)$. The generalization gap derived from Theorem 2.2 can be simplified as $O(\frac{1}{\sqrt{m}})$, which is still worse than the former.
>
> Q4: There is no similar form term in Theorem 3.2. We guess the reviewer refers to Lemma 4.2.
> Nevertheless, we don't understand what's the intuition of having term of form $1/\eta m +1/\lambda n$.
> $\lambda$ is the regularization term, but $\eta$ is the step size for task-specific algorithm.
>
> [A] Giulia Denevi, Carlo Ciliberto, Riccardo Grazzi, and Massimiliano Pontil. Learning-to-learn stochastic gradient descent with biased regularization. In International Conference on Machine Learning, pages 1566–1575. PMLR, 2019a.
>
> [B] Andreas Maurer. 'Algorithmic stability and meta-learning'. Journal of Machine Learning Research, 6(6), 2005.
>
> [r1] Fallah, Alireza, Aryan Mokhtari, and Asuman Ozdaglar. "Generalization of model-agnostic meta-learning algorithms: Recurring and unseen tasks." Advances in Neural Information Processing Systems 34 (2021): 5469-5480.
>
> [r2] Zhou, Pan, et al. "Efficient meta learning via minibatch proximal update." Advances in Neural Information Processing Systems 32 (2019).
>
> [r3] Bousquet, Olivier, Yegor Klochkov, and Nikita Zhivotovskiy. "Sharper bounds for uniformly stable algorithms." Conference on Learning Theory. PMLR, 2020.
>
> [r4] Guan, J, and Lu Z. "Task relatedness-based generalization bounds for meta learning." In International Conference on Learning Representations, 2022.
>
> [r5] Riou, Charles, Pierre Alquier, and Badr-Eddine Chérief-Abdellatif. "Bayes meets bernstein at the meta level: an analysis of fast rates in meta-learning with pac-bayes." arXiv preprint arXiv:2302.11709 (2023).

---

> > ### Comment · Reviewer_S84Y · 2024-08-10
> > **Further Clarification**
> >
> > Thank you for the reply.
> >
> > Regarding the comparison with the excess risk in [A]: the comparison should be done by imposing the same computational resources, in [A] both the within-task and the meta algorithm perform one pass on the data, so the comparison should be done by setting K=n and T=m. Could you compare the rates in such a setting? Take also in account the meaningful constants in the comparison, please: Var could bring for instance a significant advantage in comparison to others (such as a simple 1) when tasks are related in appropriate way.

---

> ### Author Response · Authors · 2024-08-11
>
> Thanks for the follow up discussion.
>
> We acknowledge that in [A], the within-task and meta algorithms perform a single pass on the data. However, in our main discussion (Algorithms 1 and 2), we leverage all samples and meta-samples for every iteration. To ensure that the computational resources are comparable to those used in [A], we would need to provide a stochastic version of the meta-learning algorithm (which we provided in Algorithm 3) and a stochastic version of the task-specific algorithm (which we did not discuss in the paper).
> As our theorem can be extended to stochastic version algorithms, below we set $K=n$ and $T=m$ to make a comparison between our result and the one presented in [A] as suggested by the reviewer.
>
> The expected excess risk in [A] is of the form $O(\frac{Var_m}{\sqrt{n}}+\frac{1}{\sqrt{m}})$. This bound is gained based on a specific choise of $\lambda$ that depends on $Var_m$, which is unlikely to know ahead in practice.
>
> In our work, to compare with the result in [A], we apply Theorem 4.4 with Theorem 4.7,
> $$
> \textrm{expected excess risk}\lesssim \frac{G^2}{\lambda m} + \frac{G^2}{\lambda n} + \frac{D^2}{\eta K} + G^2\eta+GD\eta\lambda + \frac{\lambda D^2}{T} + \frac{D^2}{T\eta K} + \frac{\eta (G+2\lambda D)^2}{T}+\lambda\sigma^2.
> $$
> Setting $K=n$, $T=m$, $\eta=O(\frac{1}{\sqrt{n}})$, $\lambda=O(\frac{1}{\sqrt{n}})$ gives us that
> $$
> \textrm{expected excess risk}\lesssim \frac{\sqrt{n}}{m} + \frac{1}{\sqrt{n}} +\frac1n+ \frac{1}{m\sqrt{n}} + \frac{\sigma^2}{\sqrt{n}}=O(\frac{\sqrt{n}}{m}+\frac{\max(1,\sigma^2)}{\sqrt{n}}).
> $$
>
> Consider both $Var_m$ and $\sigma$ to be some constants (e.g. 1), then our result is tighter than [A] when $n\lesssim m$.
>
> Please let us know if you have more questions.

---

> > ### Author Response · Authors · 2024-08-13
> > **Open to discussion**
> >
> > Dear reviewer S84Y,
> >
> > We thank the reviewer for their careful reading of the paper and have discussion with us. As the end of discussion period is approaching, please let us know whether we address your concerns. We are happy to have further discussion if you have any remaining/follow-up questions.
> >
> > Best,
> > Authors

---

### Comment · Area_Chair_NXkn · 2024-08-12
**Reminder: End of Discussion Period Approaching**

Dear Reviewers,

Thank you for taking the time to review this work. As the discussion period between reviewers and authors is approaching its end, please ensure that you have read the authors’ responses and participated in the discussions, especially if you have not done so already. If you have no further questions, kindly acknowledge the authors' response.

I look forward to continuing our discussions in the next phase of reviewing.

Please feel free to reach out to me publicly or privately if you have any questions.

Best regards,

AC

---

### Decision · Program_Chairs · 2024-09-25

**Decision:**

Accept (poster)

**Comment:**

This paper investigates the quantification of stability and generalization performance in meta-learning algorithms, addressing both convex and weakly convex problems with non-smooth losses. A newly proposed stability measure, adapted from existing ones, is tailored for this specific setting. While the two algorithms considered are fairly standard, some mismatches exist, such as in the projection operation. Under the strong assumptions made, other properties, like G-Lipschitz continuity and M-boundedness, are straightforward, though they may not hold in practice. The paper provides insights into how the meta-learning structure, the number of tasks or samples, and the regularization parameter influence stability and generalization performance. However, the numerical results are limited, as only toy examples are presented. Despite this, most reviewers generally acknowledge the contribution of this work, and the authors have addressed the majority of the concerns raised.